# AMStraMGRAM: Adaptive Multi-cutoff Strategy Modification for ANaGRAM

## Abstract

Recent works have shown that natural gradient methods can significantly out-perform standard optimizers when training physics-informed neural networks (PINNs). In this paper, we analyze the training dynamics of PINNs optimized with ANaGRAM, a natural-gradient-inspired approach employing singular value decomposition with cutoff regularization. Building on this analysis, we propose a multi-cutoff adaptation strategy that further enhances ANaGRAM's performance. Experiments on benchmark PDEs validate the effectiveness of our method, which allows to reach machine precision on some experiments. To provide theoretical grounding, we develop a framework based on spectral theory that explains the necessity of regularization and extend previous shown connections with Green's functions theory.

## 1 Introduction

Physics-informed neural networks (PINNs) have recently emerged as a promising alternative for the numerical solution of partial differential equations (PDEs) (Raissi et al., 2019). By leveraging neural networks as universal function approximators (Leshno et al., 1993), PINNs replace traditional mesh-based discretizations with sampling-based collocation methods, enabling a straightforward extension to high-dimensional domains. This mesh-free formulation not only circumvents the "curse of dimensionality" inherent in grid-based approaches, but also allows continuous evaluation of the solution throughout the domain without explicit mesh generation (Cuomo et al., 2022).

Despite these advantages, achieving low training error with PINNs remains a major challenge (Wang et al., 2023; Urbán et al., 2025; Kiyani et al., 2025; De Ryck et al., 2024). Open questions include how to select and distribute collocation points, how to balance the PDE residual against boundary-condition penalties, and which optimization strategies most effectively minimize the composite loss (Krishnapriyan et al., 2021; Wang et al., 2021; McClenny & Braga-Neto, 2022). Recent work has pursued machine-precision accuracy through hierarchical or multilevel PINN architectures (**???**), demonstrating that PINNs can successfully tackle ill-posed problems that challenge conventional finite element methods. These approaches improve accuracy primarily through architectural design. In contrast, our work focuses on accuracy gains driven purely by the optimizer, offering a complementary perspective.

A different line of research has recently reexamined PINNs from the perspective of functional geometry (Müller & Zeinhofer, 2023; 2024; Jnini et al., 2024), providing a mathematically principled view of the training dynamics. In this vein, the ANaGRAM algorithm (Schwencke & Furtlehner, 2025) applies a natural-gradient update (Amari, 1998; Ollivier, 2015), based on a reinterpretation and generalization of the neural tangent kernel (NTK; Jacot et al. (2018)) as the kernel of the projection onto the neural network's tangent space. This leads to a notion of the empirical natural gradient that projects the true functional gradient onto the empirical tangent space, yielding significantly faster convergence and lower errors compared to standard optimizers on PDE benchmarks.

Nevertheless, while ANaGRAM improves over standard optimizers, it still falls short of the accuracy attained by classical mesh-based methods, such as the finite element method (Grossmann et al., 2024). Moreover, its final performance is highly affected by the way the pseudo-inverse of the feature matrix is computed. In particular, ANaGRAM sets a fixed level of *cutoff*: a value below which the singular values of the feature matrix are ignored, *i.e.* it controls how much loss signal is incorporated into an

update. ANaGRAM's cutoff is currently chosen manually, as no automatic selection procedure has been proposed.

In this paper, we study the performance and training dynamics of ANaGRAM, with a particular focus on the role of the chosen cutoff. Typically, the training loss of ANaGRAM exhibits the slow convergence at the early iterations followed by a sudden drop at the end of the training – similar behavior is shown by the eNGD method (Müller & Zeinhofer, 2023). We discover that it is closely connected to what we further refer as the *flattening phenomenon*, which we define and characterize using the *reconstruction error*: a novel metric that measures how much of the loss signal is lost by different choices of cutoffs. Relying on the adaptive multi-cutoff strategy, our new algorithm AMStraMGRAM manages to capitalize on this phenomenon, resulting in a significant improvement (of several orders of magnitude) on various PDE benchmarks. To complement our empirical findings, we also present a functional-analytic view linking cutoff (and ridge regularization) to (generalized) Green operator theory, clarifying why cutoff regularization is essential and not just a mere fix to stabilize training.

## 2 PROBLEM STATEMENT

### 2.1 DIFFERENTIAL OPERATORS AND PHYSICS-INFORMED NEURAL NETWORKS (PINNs)

Let $\Omega \subset \mathbb{R}^d$ be a domain. We introduce two operators, $D$ and $B$, defined on a Hilbert space $\mathcal{H}$ of real-valued functions, acting respectively on $\Omega$ and on its boundary $\partial\Omega$:

$$D : \begin{cases} \mathcal{H} & \to & \mathrm{L}^2(\Omega \to \mathbb{R}, \mu) \\ u & \mapsto & D[u] \end{cases} , \qquad B : \begin{cases} \mathcal{H} & \to & \mathrm{L}^2(\partial\Omega \to \mathbb{R}, \sigma) \\ u & \mapsto & B[u] \end{cases} . \qquad (1)$$

Here, $D$ denotes a differential operator, while $B$ represents a boundary operator. A function $u \in \mathcal{H}$ is said to be a *classical solution* to the *Partial Differential Equation* (PDE) associated with $D$ and $B$ if it satisfies

$$\begin{cases} D(u) = f \in \mathrm{L}^2(\Omega \to \mathbb{R}, \mu), & \text{in } \Omega, \\ B(u) = g \in \mathrm{L}^2(\partial\Omega \to \mathbb{R}, \sigma), & \text{on } \partial\Omega, \end{cases} \qquad (2)$$

A *physics-informed neural network* (PINN) approximates the solution $u$ by a parametric model $u_{\boldsymbol{\theta}}$, where $u_{\boldsymbol{\theta}}$ is a neural network with parameters $\boldsymbol{\theta} \in \mathbb{R}^P$. The learning objective is to minimize the empirical loss

$$\ell_{D,B}(\boldsymbol{\theta}) := \frac{1}{2S_D} \sum_{i=1}^{S_D} \left( D[u_{\boldsymbol{\theta}}](x_i^D) - f(x_i^D) \right)^2 + \frac{1}{2S_B} \sum_{i=1}^{S_B} \left( B[u_{\boldsymbol{\theta}}](x_i^B) - g(x_i^B) \right)^2 . \qquad (3)$$

### 2.2 PINNs OPTIMIZERS

Training PINNs is notoriously challenging. Issues such as spectral bias, where networks struggle to learn high-frequency components, and the difficulty of balancing residual and boundary loss terms—often with vastly different magnitudes— result in unsatisfactory performance of standard deep learning optimizers (Wang et al., 2021; De Ryck et al., 2024; Krishnapriyan et al., 2021; Liu et al., 2024).

To mitigate these challenges, researchers have proposed various strategies. These include adaptive sampling approaches that focus on regions with high error (Krishnapriyan et al., 2021), dynamic loss weighting schemes (McClenny & Braga-Neto, 2022), and architectural modifications (Wang et al., 2024). Another promising line of research has focused on modifying the optimizers. In particular, two main branches of optimization approaches for PINNs have emerged:

  (i) **Second-Order Methods.** These methods, based on Quasi-Newton techniques, particularly the BFGS algorithm (Nocedal & Wright, 1999, Chapter 6) and its memory-efficient approximation L-BFGS (Liu & Nocedal, 1989), address some of the training difficulties by considering the curvature of the loss landscape. This curvature arises from the non-linearities of both the neural network and the differential operators (Rathore et al., 2024). Recently, Urbán et al. (2025) extended this approach by modifying the self-scaled BFGS (SSBFGS;

Al-Baali, 1998) and self-scaled Broyden (SSBroyden; Al-Baali & Khalfan, 2005), along with other computational enhancements such as point resampling (Wu et al., 2023) and boundary condition enforcement (Wang et al., 2023), achieving state-of-the-art results (Kiyani et al., 2025).

(ii) **Natural Gradient Methods.** In contrast to second-order methods, natural gradient methods are **first-order** techniques[1] that provide a principled way to incorporate the geometry and metric structure of the problem space. Initially introduced in the context of information geometry by Amari (1998) and later extended by Ollivier (2015), these methods were introduced for PINNs by Müller & Zeinhofer (2023). In subsequent work, Schwencke & Furtlehner (2025) connected these methods to kernel methods, yielding an efficient implementation they linked to Green's function theory (Duffy, 2015).

## 2.3 Natural Gradient Methods for PINNs

As a preliminary observation highlighted in Schwencke & Furtlehner (2025, Section 4.1), PINNs can be interpreted as a quadratic regression problem. This viewpoint arises naturally once the parametric model $u_{\boldsymbol{\theta}}$ is replaced with the following compound model:

$$(D, B) \circ u : \left\{ \begin{array}{ccccc} \mathbb{R}^P & \to & \mathcal{H} & \to & \mathrm{L}^2(\Omega, \mu) \times \mathrm{L}^2(\partial\Omega, \sigma) \\ \boldsymbol{\theta} & \mapsto & u_{\boldsymbol{\theta}} & \mapsto & (D[u_{\boldsymbol{\theta}}], B[u_{\boldsymbol{\theta}}]) \end{array} \right. . \tag{4}$$

For ease of exposition, and without loss of generality, we restrict attention to regression in $\mathrm{L}^2(\Omega, \mu)$. Given $f \in \mathrm{L}^2(\Omega, \mu)$, we define the associated empirical loss

$$\ell(\boldsymbol{\theta}) := \frac{1}{2S} \sum_{i=1}^{S} \left( u_{\boldsymbol{\theta}}(x_i) - f(x_i) \right)^2, \tag{5}$$

which can be seen as a discretization of the functional loss

$$\mathcal{L}(u) := \tfrac{1}{2} \| u - f \|_{\mathrm{L}^2(\Omega,\mu)}^2, \qquad u \in \mathrm{L}^2(\Omega, \mu). \tag{6}$$

The natural gradient approach seeks to compute the optimal update direction in function space and then pull it back to parameter space. A single Fréchet derivative of the functional loss Equation (6) yields $\nabla\mathcal{L}_{|u} = u - f$. The key insight is that admissible updates are constrained to the tangent space of the parametric model,

$$T_{\boldsymbol{\theta}}\mathcal{M} := \mathrm{Im}(\mathrm{d}u_{\boldsymbol{\theta}}) = \mathrm{Span}\left( \partial_p u_{\boldsymbol{\theta}} : 1 \leqslant p \leqslant P \right) \subset \mathcal{H}, \tag{7}$$

where $\mathcal{M} := \mathrm{Im}(u) = \{ u_{\boldsymbol{\theta}} : \boldsymbol{\theta} \in \mathbb{R}^P \} \subset \mathcal{H}$ is the manifold of functions parametrized by $\boldsymbol{\theta}$. Thus, the optimal update in function space is the projection of $\nabla\mathcal{L}_{|u}$ onto the tangent space (*cf.* Figure 5),

$$u_{\boldsymbol{\theta}_{t+1}} \leftarrow u_{\boldsymbol{\theta}_t} - \eta\,\Pi_{T_{\boldsymbol{\theta}_t}\mathcal{M}}\big( \nabla\mathcal{L}_{u_{\boldsymbol{\theta}_t}} \big)\,; \qquad \boldsymbol{\theta}_{t+1} \leftarrow \boldsymbol{\theta}_t - \eta\,\mathrm{d}u_{\boldsymbol{\theta}_t}^{\dagger}\big( \Pi_{T_{\boldsymbol{\theta}_t}\mathcal{M}}\big( \nabla\mathcal{L}_{u_{\boldsymbol{\theta}_t}} \big) \big), \tag{8}$$

where the second equation is simply the pullback of the functional update to parameter space. We prove in Section H.1 that this update is equivalent to the Gram–matrix formulation:

$$\boldsymbol{\theta}_{t+1} \leftarrow \boldsymbol{\theta}_t - \eta\,G_{\boldsymbol{\theta}_t}^{\dagger}\nabla\ell(\boldsymbol{\theta}_t)\,; \qquad G_{\boldsymbol{\theta}_t\,p,q} := \langle \partial_p u_{\boldsymbol{\theta}_t}\,,\,\partial_q u_{\boldsymbol{\theta}_t} \rangle_{\mathrm{L}^2(\Omega,\mu)}\,. \tag{9}$$

## 2.4 ANaGRAM: Empirical Natural Gradient

The $O(P^3)$ complexity of matrix inversion in Equation (9) renders a direct implementation prohibitively expensive. ANaGRAM (Schwencke & Furtlehner, 2025) circumvents this by exploiting a motivated approximation. The key observation is that the update can be expressed in terms of the empirical feature matrix $\widehat{\phi} \in \mathbb{R}^{S \times P}$ and the empirical functional residuals $\widehat{\mathcal{L}_{\boldsymbol{\theta}}} \in \mathbb{R}^S$:

$$\boldsymbol{\theta}_{t+1} \leftarrow \boldsymbol{\theta}_t - \eta\,\widehat{\phi}^{\dagger}\widehat{\nabla\mathcal{L}}_{\boldsymbol{\theta}_t}\,; \qquad \widehat{\phi}_{i,p} := \partial_p u_{\boldsymbol{\theta}}(x_i)\,; \qquad \left( \widehat{\nabla\mathcal{L}_{\boldsymbol{\theta}}} \right)_i := u_{\boldsymbol{\theta}}(x_i) - f(x_i). \tag{10}$$

---

[1]contrary to a widespread misconception, which arises from their analogy in the context of information theory

Here, the pseudo-inverse is computed via singular value decomposition (SVD): $\widehat{\phi}^\dagger = \widehat{U}\widehat{\Delta}^\dagger \widehat{V}^T$ with $\widehat{\phi} = \widehat{V}\widehat{\Delta}\widehat{U}^T$, where $\widehat{U} \in \mathbb{R}^{P \times r_{svd}}$, $\widehat{\Delta} \in \mathbb{R}^{r_{svd} \times r_{svd}}$, $\widehat{V} \in \mathbb{R}^{S \times r_{svd}}$, and $r_{svd} = \min(P, S)$. This reduces computational cost to $O(\min(PS^2, P^2S))$, which is tractable in practice. A comparable complexity was later obtained by Guzmán-Cordero et al. (2025) using a Cholesky factorization approach.

For further details on the derivation of the empirical natural gradient, we refer to Schwencke & Furtlehner (2025). In what follows, we adopt a slight abuse of notation by omitting the explicit dependence on $\boldsymbol{\theta}$ whenever it is clear from context. When iteration indices matter, we explicitly write $t$ to emphasize the connection to $\boldsymbol{\theta}_t$.

## 2.5 REGULARIZATION

As discussed in Section G.1, the type of problem we consider is ill-conditioned, which necessitates the use of regularization. We distinguish between two main regularization schemes: (i) *ridge regression*, which consists in adding a factor $\alpha^2 I_d$ (or, according to conventions, $\alpha^{-2}I_d$) to the Gram matrix $G_{\boldsymbol{\theta}}$ in Equation (9) (or its approximation $\widehat{\mathcal{G}}_{\boldsymbol{\theta}}$), thereby making it invertible or (ii) *cutoff regularization*, a scheme that applies a binary threshold (used in ANaGRAM):

$$\widehat{\Delta}^\dagger_{t,i} = \begin{cases} \widehat{\Delta}^{-1}_{t,i}, & \text{if } \widehat{\Delta}_{t,i} \geqslant \alpha, \\ 0, & \text{otherwise.} \end{cases} \tag{11}$$

Here $\alpha$ denotes the cutoff threshold. This regularization is the focus of our analysis in Section 3. For completeness, we provide a geometric interpretation of each scheme in Section G. We further show that cutoff regularization extends previously established connections between natural gradient methods and Green's function theory (Schwencke & Furtlehner, 2025). In particular, we obtain:

**Theorem 1.** *The generalized Green's function of the operator $D$ in the regularized space $\mathcal{H}^\alpha_{D,\mathcal{H}_0}$ is given, for all $x, y \in \Omega$, by*

$$g_D(x, y) := D[k_D(x, \cdot)](y), \tag{12}$$

where $\mathcal{H}^\alpha_{D,\mathcal{H}_0}$ is a regularized space with reproducing kernel $k_D$, defined in Section G.4. As a consequence, we show that AMStraMGRAM converges in the NTK-regime as explained in Section G.5.

# 3 INSIGHTS ON ANAGRAM'S TRAINING DYNAMICS

In this section, we will look at relevant quantities of interest to understand this empirical phenomenon.

## 3.1 RECONSTRUCTION ERROR OF FUNCTIONAL GRADIENT

Let $\boldsymbol{\theta} \in \mathbb{R}^P$, the empirical feature matrix $\widehat{\phi} \in \mathbb{R}^{S \times P}$, and the empirical functional gradient $\widehat{\nabla \mathcal{L}} \in \mathbb{R}^S$ as defined in Equation (10). Let us consider various empirical tangent spaces formed by taking different ranges of right singular vectors of $\widehat{\phi} = \widehat{U}\widehat{\Delta}\widehat{V}^T$, i.e. $\widehat{T^M_N \mathcal{M}} = \mathrm{Span}(\widehat{V}_{t,i} : M \leqslant i \leqslant N)$. For $1 \leqslant N \leqslant r_{svd}$, reconstruction error measures how much information from the functional gradient signal is lost when considering only first $N$ components in SVD (the error caused by the projection onto the empirical tangent space $\widehat{T^0_N \mathcal{M}}$) is defined as follows

$$\mathrm{RCE}^S_N := \frac{1}{\sqrt{S}} \left\| \widehat{V}\Pi^0_N \widehat{V}^T \widehat{\nabla \mathcal{L}} - \widehat{\nabla \mathcal{L}} \right\|_{\mathbb{R}^S} = \frac{1}{\sqrt{S}}\|\Pi^\perp_{\widehat{T^0_N \mathcal{M}}} \widehat{\nabla \mathcal{L}} - \widehat{\nabla \mathcal{L}}\|, \tag{13}$$

where we define $\Pi^M_N \in \mathbb{R}^{r_{svd} \times r_{svd}}$ as a projection operator onto $\widehat{T^M_N \mathcal{M}}$:

$$\Pi^M_N = \sum_{p=M+1}^N \boldsymbol{e}^{(p)}\boldsymbol{e}^{(p)^T}, \tag{14}$$

with $(\mathbf{e}^{(p)})_{1 \leqslant p \leqslant r_{svd}}$ being the canonical basis of $\mathbb{R}^{r_{svd}}$.

**Proposition 1.** *$RCE^S_N$ is a non-increasing function of $N$, i.e. for all $1 \leqslant M, N \leqslant r_{svd}$:*

$$M \leqslant N \implies RCE^S_M \geqslant RCE^S_N. \tag{15}$$

*Furthermore, assuming that $(x_i)_{i=1}^S$ are i.i.d sampled from $\mu$, we have $\mu$-almost surely*

$$\lim_{S\to\infty} RCE_N^S = \left\| \nabla\mathcal{L}_{u_{\boldsymbol{\theta}}} - \Pi_{T_N^0\mathcal{M}}^{\perp} \nabla\mathcal{L}_{u_{\boldsymbol{\theta}}} \right\|_{L^2(\Omega,\mu)} = \left\| \Pi_{\left[T_N^0\mathcal{M}\right]^{\perp}}^{\perp} \nabla\mathcal{L}_{u_{\boldsymbol{\theta}}} \right\|_{L^2(\Omega,\mu)}, \tag{16}$$

*where $T_N^M\mathcal{M} = \mathrm{Span}(V_{t,i} : M \leqslant i \leqslant N)$, while $(V_{t,i})_{1\leqslant i\leqslant r_{svd}}$ are the right singular-vectors of the differential $du_{\boldsymbol{\theta}}$ ordered in a decreasing order according to their associated singular values.*

*Remark* 1. Note that $\widehat{V}_{t,i} \in \mathbb{R}^S$ for $i \in 1,\dots,N$, the right singular vectors of $\widehat{\phi}$, can be seen as discretized versions of $V_{t,i}$ from Proposition 1. Indeed, a weak convergence holds, *i.e.* $\forall h \in \mathcal{H}$, $\frac{1}{S}\sum_{j=1}^S \widehat{V}_{t,i,j} h_j = \frac{1}{S}\sum_{j=1}^S V_{t,i}(x_j) h(x_j) \overset{S\to\infty}{\to} \langle V_{t,i}, h\rangle_{L^2}$.

Proof of Proposition 1 can be found in Section H.3. From Proposition 1 RCE is related to the concept of *expressivity bottleneck* illustrated in Verbockhaven et al. (2024), and measures what part of the learning signal is not captured by truncating at $N$ components for natural gradient computation. Therefore, this metric allows us to explicitly estimate and compare different cutoff choices. Note that this metric incurs no additional computational cost since ANaGRAM already computes the required SVD.

## 3.2 EMPIRICAL OBSERVATIONS: FLATTENING

Here we illustrate the evolution of training loss and reconstruction error, where Figure 1 schematically outlines key stages of ANaGRAM's training dynamics. The plot of a real experiment is provided in Section E.

Let $\alpha$ is a cutoff level (also referred to as precision) and $r_{\text{cutoff}}$ denote the number of components retained by the cutoff, i.e., $r_{\text{cutoff}}(t) = \max\{j : \widehat{\Delta}_{t,j} \geqslant \alpha\}$. In Figure 1, we observe different stages of the training. First, the reconstruction error is above the wanted precision (Figure 6a). As the training progresses, the training loss drops and the reconstruction error drops until reaching the cutoff precision (Figure 6b). Eventually, the reconstruction error drops below the cutoff threshold (Figure 6c). During this phase, the training loss (corresponding to the RCE for 0 component (green line in the figure)) is not decreasing a lot.

Then, a phenomenon that we call "flattening" occurs: once the reconstruction error is small compared to the cutoff precision value, reconstruction error *flattens* over the interval $[N_{\text{flat}}, r_{\text{cutoff}}]$, where $N_{\text{flat}}$ is the smallest number such as

$$\mathrm{RCE}_{N_{\text{flat}}}^S - \mathrm{RCE}_{r_{\text{cutoff}}}^S \approx 0. \tag{17}$$

Eventually, the phenomenon propagates toward low numbers of retained components (Figure 6e) and $N_{\text{flat}} = 0$. Reconstruction error is now constant for all retained components and the training ends with training loss at cutoff precision. We refer a reader to Section H.3 to have a more theoretical insight on what is happening during the flattening.

*Remark* 2. This phenomenon sheds light on the sharp drop in training loss observed near the end of optimization, as reported in Schwencke & Furtlehner (2025). By combining Equations (5), (10) and (13) and using that $\Pi_0^0 = 0$, we obtain

$$\mathrm{RCE}_0^{S\,2} \overset{13}{=} \frac{1}{S}\left\| \widehat{V}\Pi_0^0 \widehat{V}^t \widehat{\nabla\mathcal{L}} - \widehat{\nabla\mathcal{L}} \right\|_{\mathbb{R}^S}^2 = \frac{1}{S}\left\| \widehat{\nabla\mathcal{L}} \right\|_{\mathbb{R}^S}^2 \overset{10}{=} \frac{1}{S}\sum_{i=1}^S \left(u_{\boldsymbol{\theta}}(x_i) - f(x_i)\right)^2 \overset{5}{=} \ell(\boldsymbol{\theta}). \tag{18}$$

Thus, the last iteration of flattening is **directly responsible for the sudden drop of train loss** at the end of the training.

*Remark* 3. We see that for higher precision than the cutoff value ($N > r_{\text{cutoff}}$), the RCE is still decreasing as we increase the number of components kept. This indicates that there is still information to capture in the functional eigenspace composed of components associated to lower eigenvalues, see also Section H.3.

The final interesting observation is that

$$\mathrm{RCE}_0^S - \mathrm{RCE}_{r_{\text{cutoff}}}^S \simeq 0 \qquad \Leftrightarrow \qquad \Pi_{r_{\text{cutoff}}}^0 \widehat{V}^T \widehat{\nabla\mathcal{L}} \approx 0. \tag{19}$$

Thus, the flattening phenomenon means that the projection of the signal onto the first $r_{\text{cutoff}}$ components retained by the cutoff is negligible. In other words, the optimization has extracted all the *usable* signal from these components at this cutoff level.

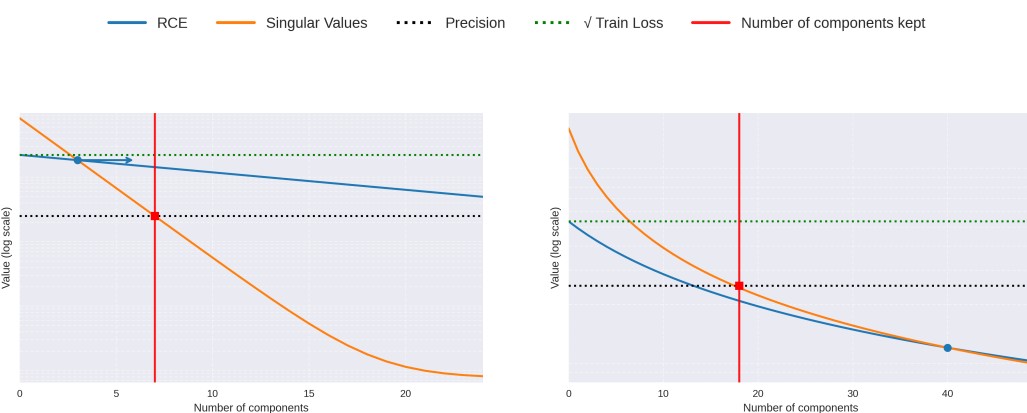

(a) Early iterations, RCE at intersection with singular values is above the desired precision threshold.

(b) The RCE and singular values intersection drops below precision.

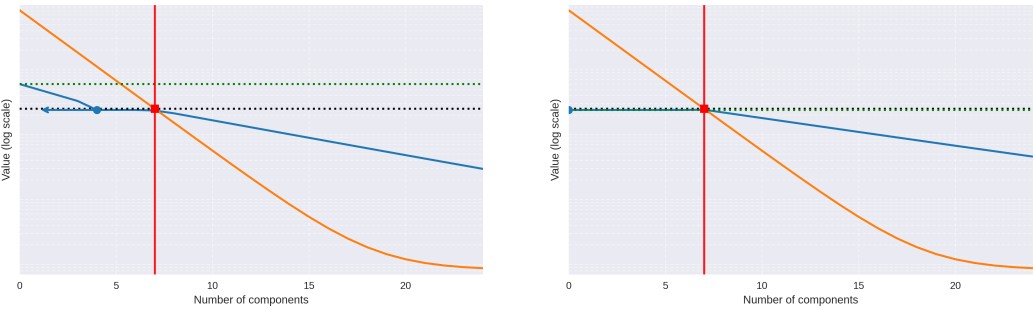

(c) Beginning of the flattening: a plateau of RCE starts from $r_{cutoff}$ and propagates toward zero.

(d) Final stage: full flattening and convergence.

Figure 1: **ANaGRAM training dynamics.** Legend (top) and four key phases: (a) initial evolution, (b) reconstruction–singular value intersection passes target precision, (c) emergence of the flattening regime, (d) complete flattening yielding final loss level. Despite changing scale, target precision is constant and fixed across all plots. The number of ANaGRAM's retained components $r_{cutoff}$ is at intersection of precision line with singular values curve.

## 3.3 INCOMPLETE FLATTENING AND ADAPTIVE STRATEGIES

In practice, for some experiments we observe that the flattening may remain incomplete with $\lim_{t\to\infty} N_{flat} = N_{flat}^{\infty} > 0$: the system remains in a state similar to that shown in Figure 1c and never (at least not within a reasonable number of iterations) reaches the configuration illustrated in Figure 1d. A natural question arises: *what happens if we adjust the cutoff to retain exactly $N_{flat}^{\infty}$ components?*

If we try this trick in practice (see Figure 7), then a single natural gradient step with an adjusted cutoff can be enough to get immediate and complete flattening ($N_{flat} = 0$) and eventually dramatically reduce training loss. This abrupt flattening when restricting cutoff to low number of feature is typically accompanied by a learning rate found by the line search to be very close to one. A possible explanation is that this may represent an iteration in the *lazy training* regime (NTK and the feature matrix are nearly constant), where we regress linearly (and thus fast) based on learned features. Link between flattening and lazy training is described in Appendix J.

This empirical insight motivates the use of an adaptive algorithm: by dynamically adjusting cutoffs, we can hope to accelerate convergence and achieve higher precision.

## 4 Algorithmic Design: Exploiting Flattening

Building upon the empirical analysis presented in Section 3, we develop a principled algorithm that controls and exploits the flattening phenomenon identified in ANaGRAM's training dynamics. Our approach is based on tracking the relationship between reconstruction error and singular values to automatically determine well-adapted cutoff in order to reach the target precision (error) $\epsilon$ at the end of the training. This well-adapted cutoff should vary from one iteration to another to adjust to the currently learned weights and training dynamics in such a way to avoid early flattening (if flattening happens too early, the training stagnates at higher values of losses) and when intersection between RCE and singular values goes below the target precision $\epsilon$, we enforce the flattening, so that the final training loss also drops to $\epsilon$.

### 4.1 Adaptive Cutoff Strategy

In what follows, we suggest an adaptive cutoff rank $\mathrm{r_{cutoff}}$ that indicates how much components of $\widehat{\Delta}$ are retained for the next update of ANaGRAM. Our algorithm operates by dynamically selecting cutoff ranks based on the relationship between reconstruction error and singular values:

$$\mathrm{r_{cutoff}}(t) = \begin{cases} \mathrm{r_{int}}(t) := \max\left\{j : \mathrm{RCE}_j^S(t) \leqslant \widehat{\Delta}_{t,j}\right\} & \text{if } \mathrm{RCE}_{\mathrm{r_{int}}(t)}^S(t) > \epsilon \text{ (intersection rank)}, \\ \mathrm{r_\epsilon}(t) := \max\left\{j : \mathrm{RCE}_j^S(t) \geqslant \epsilon\right\} & \text{if } \mathrm{RCE}_{\mathrm{r_{int}}(t)}^S(t) \leqslant \epsilon \text{ (precision rank)}. \end{cases} \quad (20)$$

The algorithm terminates when $\mathrm{r_\epsilon}(t) = 0$, indicating that the reconstruction error $\mathrm{RCE}_0^S$ that is equal to the training error is indeed below the predefined precision threshold.

**Target accuracy**   The parameter $\epsilon$ specifies the target residual precision that the algorithm aims to achieve. By design, if the algorithm converges (i.e., if the final rank reaches 0), the resulting training residual is guaranteed to lie at or below this prescribed level. As shown in Figure 2, once the smallest retained singular value reaches $\epsilon$, the algorithm enters the flattening regime, ensuring that the residual stabilizes beneath this threshold.

In practice, when an appropriate network architecture and a sufficiently informative set of collocation points are used (as illustrated in our Heat and Laplace experiments), choosing $\epsilon$ close to machine precision typically yields the best results. Conversely, if convergence is not observed, $\epsilon$ can be increased by a few orders of magnitude, accepting a lower target precision but ensuring convergence.

For ease of presentation, we provide only the core elements of AMStraMGRAM in Algorithm 1 consisting in adaptively choosing, which $\mathrm{r_{cutoff}}$ to apply for $\widehat{\Delta}$ at each update of ANaGRAM. The final algorithm is explained in Section C. Final Algortihm **??** addresses some irregularities observed in evolution of RCE and singular values that we explain in more details in Section C.4.

### 4.2 Geometrical Interpretation of the Adaptive Strategy

The algorithm exploits the geometric relationship between the empirical tangent space and the functional gradient. By tracking the intersection, we maximize the projection of the functional gradient onto the empirical tangent space while staying out of flattening. Once the intersection reach the precision level, we exploit the flattening phenomenon to achieve prescribed precision.

According to Proposition 1, the reconstruction error $\mathrm{RCE}_N^S$ measures how much of the functional gradient signal remains to be captured by the first $N$ components. The intersection point thus represents the good balance between signal capture and phase transition.

## 5 Experimental Results

We first compare in Table 1 our method implemented in JAX[2] with the ANaGRAM method (Schwencke & Furtlehner, 2025) on the benchmark problems presented in their paper, with modified datasets. As we see, for every equation, we perfom better.

---

[2]https://anonymous.4open.science/r/AMStraMGRAM-8D1B/

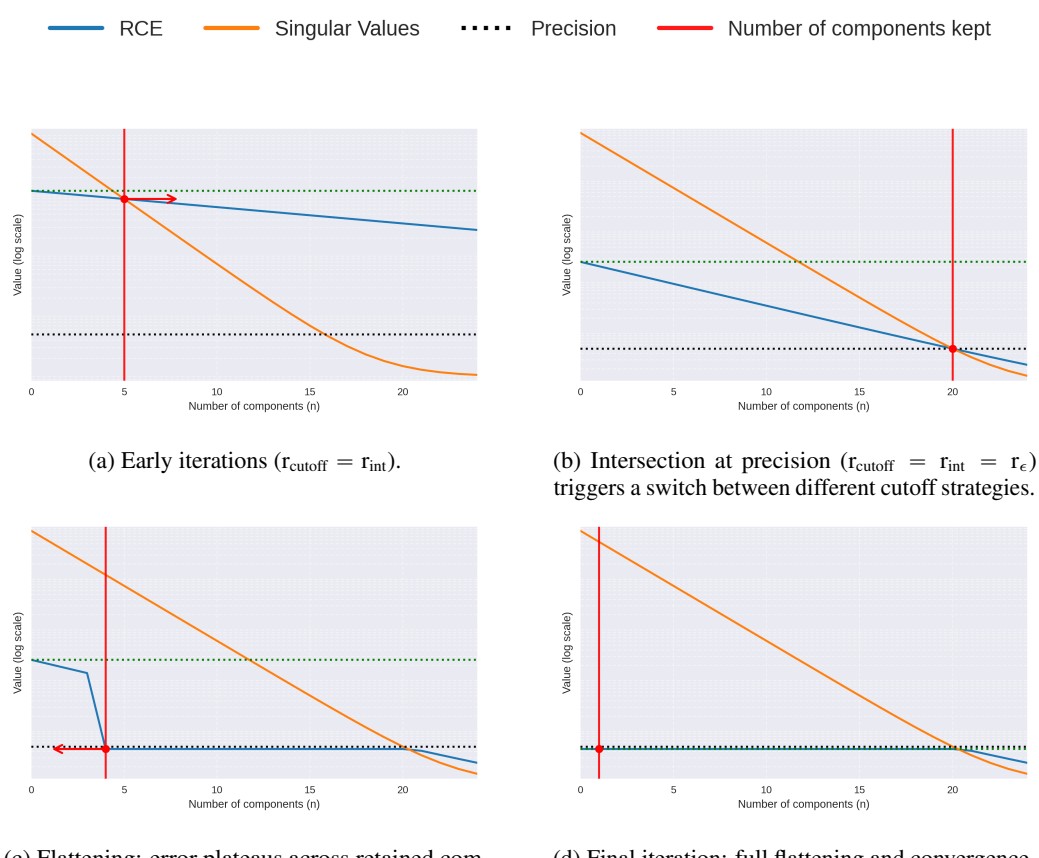

(a) Early iterations ($r_{cutoff} = r_{int}$).

(b) Intersection at precision ($r_{cutoff} = r_{int} = r_\epsilon$) triggers a switch between different cutoff strategies.

(c) Flattening: error plateaus across retained components $r_{cutoff} = r_\epsilon$.

(d) Final iteration: full flattening and convergence.

Figure 2: **Dynamics of the adaptive multi-cutoff strategy in AMStraMGRAM.** Progression from (a) initial exploration, (b) intersection reaches precision, (c) flattening onset, to (d) converged state. Red arrows (when present) indicate the retained rank dynamics (pointing right – increasing, pointing left – decreasing). Legends are shown below.

Table 1: Performance comparison between AMStraMGRAM (our method) and ANaGRAM Schwencke & Furtlehner (2025). The adaptive strategy demonstrates significant improvements across all benchmark problems, with L2 error improvements of up to 8 orders of magnitude.

| Experiment | Mean Squared Error (MSE) | | $L_2$ Error | |
|---|---|---|---|---|
| | Ours | ANaGRAM | Ours | ANaGRAM |
| Heat Equation | **6.29e-29 $\pm$ 6.78e-30** | 8.56e-11 $\pm$ 7.05e-11 | **2.32e-14 $\pm$ 1.14e-14** | 1.28e-06 $\pm$ 1.75e-06 |
| Laplace 2D | **1.46e-28 $\pm$ 1.87e-29** | 4.27e-13 $\pm$ 4.66e-13 | **2.24e-15 $\pm$ 2.52e-16** | 3.49e-09 $\pm$ 3.58e-09 |
| Laplace 5D | **2.04e-08 $\pm$ 1.16e-08** | 6.37e-08 $\pm$ 7.01e-08 | **2.12e-05 $\pm$ 8.15e-06** | 4.00e-05 $\pm$ 2.93e-05 |
| Allen–Cahn | **3.19e-11 $\pm$ 2.37e-11** | 2.19e-04 $\pm$ 4.16e-04 | **5.87e-05 $\pm$ 6.25e-06** | 4.32e-03 $\pm$ 5.93e-03 |

We then compare our method with the baseline methods from Urbán et al. (2025) on the benchmark problems presented in their paper. Note that in our case we do not need to enforce boundary constraints. The methodology of sampling is also sighltly different, as we sample the data from a fixed grid, following the methodology of Schwencke & Furtlehner (2025), while in Urbán et al. (2025) they perform batching of randomly sampled points.

Table 2: Performance comparison between AMStraMGRAM (our method) and baseline Urbán et al. (2025) methods. Our method demonstrates improvements across benchmark problems, without requiring enforcement of boundary constraints.

| Experiment | Mean Squared Error (MSE) | | $L_2$ Error | |
|---|---|---|---|---|
| | Ours | SSBroyden* | Ours | SSBroyden* |
| One-dimensional Burgers (1DB) | **2.99e-12 ± 9.26e-13** | 2.92e-10 ± 1.45e-10 | **1.5e-06 ± 9.43e-7** | 1.59e-06 ± 1.02e-6 |
| Non-Linear Poisson (k=1) | **8.51e-24 ± 2.24e-24** | 3.03e-16 ± 3.82e-16 | 6.81e-10 ± 1.41e-09 | **9.29e-12 ± 5.85e-12** |
| Allen–Cahn (AC) | 3.19e-11 ± 2.37e-11 | **6.42e-12 ± 5.52e-12** | 5.87e-05 ± 6.25e-06 | **3.94e-06 ± 1.72e-06** |

\* refer to method from Urbán et al. (2025) with adaptive sampling and hard constraint enforcement on boundary conditions.

## 6 LIMITATIONS

Despite its effectiveness, AMStraMGRAM can exhibit overfitting, particularly in problems with sharp features like the Allen–Cahn equation. The algorithm drives the training error to machine precision on the sampled points, but the learned function may develop high-frequency oscillations between them, especially in regions of high curvature where the approximation is the most challenging. These artifacts, visible as "overfitting lines" in Figure 3, are an imprint of the sampling lattice (see regions around $x = \pm 0.5$). They arise because the SVD cutoff effectively projects the update onto a low-rank subspace of the tangent space. This subspace is often aligned with the grid axes, leading to anisotropic smoothing that perfectly fits the data on the grid lines but interpolates poorly in the under-sampled regions between them. Once the flattening phase begins, the training enters a quasi-linear regime that can "lock in" these geometric artifacts.

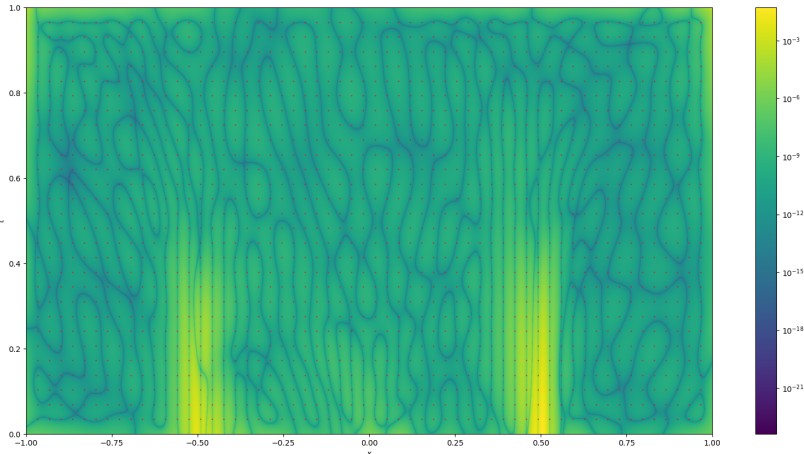

Figure 3: Allen–Cahn overfitting: residual lines align with sampling lines. Low-rank (post-cutoff) tangent projections fit exactly on sampled fibers while interpolation between them inherits weakly constrained oscillations in regions of steep interface curvature.

This phenomenon highlights that while our method significantly improves on ANaGRAM, the quality of the final solution remains fundamentally limited by the sampling strategy. Mitigating such overfitting requires co-designing the sampler and the optimizer. Potential remedies include adaptive sampling, where new collocation points are added in regions of high reconstruction error, or curriculum-based approaches that progressively refine the sampling grid.

**Disentangling solution accuracy from sampling error.** Our method attains machine precision on the residual and not on the error. This last metric depends crucially on the choice of points. Now that limitations of the optimizer are not the main bottleneck anymore, the investigation of a proper sampling/quadrature strategy should be the next step to further reduce $L^2$ error. We illustrate the relationship between collocation density (expressed as number of points per dimension) and error in Figure 4.

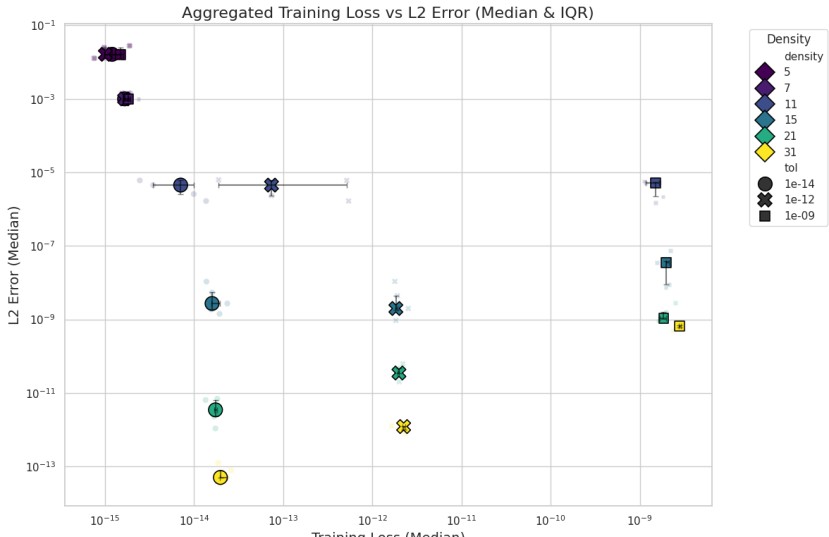

Figure 4: Relationship between collocation density and error for the Heat equation with fixed model size. The mesh density corresponds to the number of points per dimension in the interior of the domain. The 'tol' values correspond to the desired precision (cutoff). We observe that increasing the number of points improves the $L_2$ error without significant change in the residual, suggesting that the optimizer effectively minimizes the loss on the available points, and the remaining error is dominated by the discretization (sampling) error.

**Stability Under Resampling**   While AMStraMGRAM employs a fixed grid to ensure deterministic updates and stable cutoff tracking, standard PINN training often relies on minibatch sampling. In a batchwise setting, the truncated natural-gradient update becomes stochastic not only because of noise in gradient estimation, but also due to fluctuations in the empirical feature matrix. In Appendix I, we provide additional experiments illustrating that different stochastic mini-batches induce different empirical tangent spaces. Our results indicate that when the cutoff rank is either very low or very high, the choice of samples has only a minor effect on the resulting empirical tangent space. In contrast, for intermediate cutoff values, different batches can produce noticeably different update directions, and consequently lead to distinct training dynamics.

These observations suggest that developing principled stochastic batching schemes or adaptive sampling strategies for our algorithm constitutes an interesting direction for future work.

## 7   CONCLUSION

In this work, we have introduced AMStraMGRAM, an adaptive multi-cutoff strategy that enhances the ANaGRAM natural gradient method for training PINNs. Our work provides an analytical framework to explain ANaGRAM's convergence behavior, uncovering a *flattening* phenomenon that clarifies its training dynamics. The proposed algorithm automatically adjusts cutoff regularization. Notably, AMStraMGRAM exhibits "overfitting" as demonstrated in Allen-Cahn experiments. These results underscore the potential of natural gradient optimization for PINNs while highlighting the critical role of sampling strategies in realizing their full accuracy.

Future research will focus on integrating residual-based methods to further stabilize training, establishing rigorous dynamics analysis for the feature-development phase, and extending the approach to higher-dimensional PDEs and complex geometries. Exploring the interplay between network architecture and optimization—as well as further developing sampling techniques—will be essential to address the fundamental challenge of balancing optimization power with data representation. Ultimately, our findings suggest that with careful algorithmic design, PINNs can achieve the precision required for practical scientific computing, paving the way for mesh-free methods in computational science.

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

# A  ILLUSTRATION OF NATURAL GRADIENT

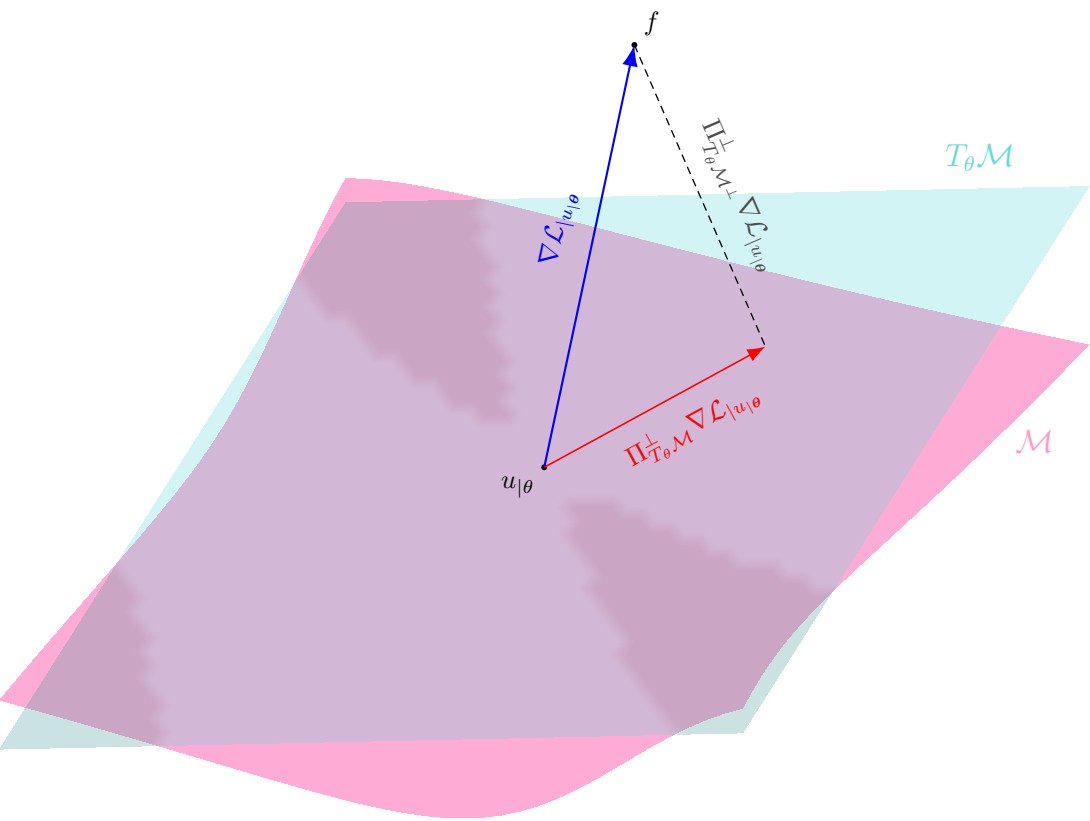

Figure 5: Illustration of the orthogonal projection of the functional gradient onto the tangent space. While the ideal update direction would be the functional gradient $\nabla\mathcal{L}_{|u_\theta}$ (shown in blue), our model constrains us to follow directions within the tangent space $T_\theta\mathcal{M}$ (shown as a green plane). The optimal feasible direction is thus the orthogonal projection $\Pi^\perp_{T_\theta\mathcal{M}}\left(\nabla\mathcal{L}_{|u_\theta}\right)$ (shown in red).

# B  OUR VOCABULARY

- **Domain** ($\Omega$).
- **Boundary** ($\partial\Omega$).
- **Differential operators** ($D, B$).
- **Cutoff** ($\alpha_t$). A threshold below which the components of the matrix $\widehat{\Delta}$ are truncated, *i.e.*
$$\widehat{\Delta} \leftarrow \begin{cases} \widehat{\Delta} & \text{if } \widehat{\Delta} \geqslant \alpha_t, \\ 0 & \text{else.} \end{cases}$$
- **Full rank** ($r_{\text{svd}}$). A full rank of feature matrix $\widehat{\phi}$ that we assume, without loss of generality, to be equal to $\min(P, S)$.
- **Rank** ($r_{\text{cutoff}}$). A number of $\widehat{\Delta}$ components that are retained when computing a pseudo-inverse of $\widehat{\Delta}$ in ANaGRAM. Depending on a current regime of the training and a desired effect, it can be set at $r_{\text{int}}$ or $r_\epsilon$.
- **Flattening.** The phenomenon described in Section 3.2, when reconstruction error starts to stabilize for a range of possible ranks.
- **Flat cutoff** ($N_{\text{flat}}$). A number of components that corresponds to the beginning of flattening in reconstruction error curve.

- **Feature matrix** ($\widehat{\phi} \in \mathbb{R}^{P \times S}$). It is defined by a jacobian $\partial_p u_{\boldsymbol{\theta}}(x_i)$, which is used in an ANaGRAM's update to "project" a functional gradient onto parameter space of $\boldsymbol{\theta}$.

- **Precision** ($\epsilon$). A hyperparameter of AMStraMGRAM that prescribes a target error level that the algorithm should achieve.

- **Intersection rank ($\mathbf{r_{int}}$).** Defined in Equation (20), roughly speaking it corresponds to a number of components at which reconstruction error and singular values curves are intersecting.

- **Precision rank ($\mathbf{r_{\epsilon}}$).** Defined in Equation (20), it corresponds to a number of components at which reconstruction error curve and precision level are intersecting.

- **Functional gradient** ($\nabla \mathcal{L}$). A Frechet derivative of squared $L^2$ loss $\mathcal{L}$, its negative gives the "ideal" update direction in non-parametric case.

- **Empirical functional gradient** ($\widehat{\nabla \mathcal{L}} \in \mathbb{R}^S$). A vector obtain by evaluating $\nabla \mathcal{L}$ on some finite number of samples $x_i \in \Omega$, for $i \in 1, \ldots, S$.

- **Parametric model** ($u_{\boldsymbol{\theta}}$). A function parametrized with $\boldsymbol{\theta}$ that serves to approximate a solution to a problem (regression or PDE). Typically, it is a neural network, where $\boldsymbol{\theta}$ are its full set of weights.

- **Differential of the model** ($d\, u_{\boldsymbol{\theta}}$). Defined as $d\, u_{\boldsymbol{\theta}}(h) = \sum_{p=1}^{P} h_p \frac{\partial u}{\partial \boldsymbol{\theta}_p} = \lim_{\varepsilon \to 0} \frac{u_{|\boldsymbol{\theta} + \varepsilon h} - u_{\boldsymbol{\theta}}}{\varepsilon}$. It measures how much $u_{\boldsymbol{\theta}}$ changes in a given direction $h$.

- **Tangent space** ($T_{\boldsymbol{\theta}} \mathcal{M}$). Image of a differential of the model, giving a space of possible updates for a model $u_{\boldsymbol{\theta}}$.

- **SVD components of** $\widehat{\phi}$ ($\widehat{U}, \widehat{\Delta}, \widehat{V}$). In particular, $\widehat{\phi} = \widehat{U} \widehat{\Delta} \widehat{V}^T$, where $\widehat{U} \in \mathbb{R}^{P \times S}$ is a left singular vector matrix, $\widehat{\Delta} \in \mathbb{R}^{r_{svd} \times r_{svd}}$ is a diagonal matrix with singular values on a diagonal ordered in a decreasing order and $\widehat{V}$ is a right singular vector matrix.

- **Functional singular vectors** ($V_{t,i}$). Right singular vectors of the differential $du_{\boldsymbol{\theta}}$.

- **Empirical tangent space** ($T_N^M \mathcal{M}$). A subspace of tangent space $T_{\boldsymbol{\theta}} \mathcal{M}$, restricted to a span of the right functional singular vectors $V_{t,i}$ corresponding to a range of components from $M$ to $N$, *i.e.* $\mathrm{Span}(V_{t,i} : 1 \leqslant M \leqslant N \leqslant \bar{N})$.

- **Discretized empirical tangent space** ($\widehat{T_N^M \mathcal{M}}$). A version of $T_N^M \mathcal{M}$ discretized on a set of samples $\{x_i\}_{i=1}^{S}$ coming from $\Omega$.

- **Reconstruction error** ($\mathrm{RCE}_N^S$). A measure identifying the portion of the functional gradient signal that is lost when restricting $\widehat{\nabla \mathcal{L}}$ to $\widehat{T_N^0 \mathcal{M}}$.

- **Feature development phase.** The early phase in the training, during which high volatility is observed in both quantities of interest with high sensitivity to the choice of $r_{cutoff}$.

- **Flattening phase.** The later phase in the training, during which reconstruction error starts to flatten for some values of $N$, at the same time singular values dominate over reconstruction error for all retained components, resulting in a drop of training loss.

## C  PRACTICAL IMPLEMENTATION CONSIDERATIONS

While the principled algorithm discussed in the main paper and summarized in Algorithm 1 provides a sound framework, empirical observations reveal that additional mechanisms are necessary for robust performance across diverse PDE problems. This section describes additional modifications to make the algorithm more practical.

### C.1  THE DUAL CUTOFF STRATEGY: ADDRESSING EMPIRICAL CHALLENGES

Our experiments reveal that the single cutoff approach, while theoretically elegant, suffers from numerical instabilities and incomplete convergence in practice. We observed three critical issues:

1. **Ignition failure:** The intersection between reconstruction error and singular values sometimes fails to evolve, preventing the algorithm from reaching lower error values.

2. **Retreating dynamics:** The intersection rank may decrease during training, disrupting convergence.

3. **Incomplete flattening:** Without additional stabilization, the flattening phenomenon may not complete, leading to suboptimal final accuracy.

To address these challenges, we introduce a dual cutoff strategy inspired by the staged design of rocket launches:

### C.2 THREE-PHASE TRAINING DYNAMICS

#### C.2.1 IGNITION PHASE

We initialize two cutoffs:

- **Minimum cutoff ($r_{min}$):** Set at the intersection point $r_{int}(t)$

- **Maximum cutoff ($r_{max}$):** Set at the "elbow" of the singular value curve (see algorithm 4)

The algorithm performs two natural gradient steps per iteration, one with each cutoff. If the intersection position remains static after both updates, we increment $r_{max}$ by one to promote exploration of additional gradient components.

This phase ends when $r_{min}$ reaches $r_{max}$—an event we term **liftoff**.

#### C.2.2 ASCENT PHASE

During ascent, both cutoffs track the moving intersection, but with a stability mechanism:

$$r_{max}(t) = \max(r_{max}(t-1), r_{int}(t)). \tag{21}$$

This monotonicity constraint prevents the intersection rank from falling to zero, which would disrupt training dynamics.

#### C.2.3 STAGE SEPARATION AND PRECISION LOCKING

When $\text{RCE}^S_{r_{int}(t)}(t) \leqslant \epsilon$, we trigger **stage separation**:

- $r_{min}$ is fixed at the precision level: $r_{min} = r_\epsilon(t)$

- $r_{max}$ continues tracking the intersection to maintain stability

The algorithm continues until $r_{min} = 0$ (**booster return**), indicating complete convergence. The final algorithm that combines all three stages is mentioned in Algorithm **??**.

---

**Algorithm 1:** Sketch of the Adaptative MultiCutoff Strategy for ANaGRAM (AMStraMGRAM)

---

**Input:** $u_{\boldsymbol{\theta}} : \mathbb{R}^P \to \mathrm{L}^2(\Omega, \mu), \boldsymbol{\theta}_0 \in \mathbb{R}^P, f \in \mathrm{L}^2(\Omega, \mu), (x_i) \in \Omega^S, \epsilon > 0, T_{\max} \in \mathbb{N}$

// Initialization

1   $t \leftarrow 0$

2   $\widehat{\phi}_0 \leftarrow (\partial_p u_{\boldsymbol{\theta}_0}(x_i))_{i,p}$ for $i \in 1, \ldots, S$ and $p \in 1, \ldots, P$

3   $\widehat{U}_0, \widehat{\Delta}_0, \widehat{V}_0^T \leftarrow \mathrm{SVD}\left(\widehat{\phi}_0\right)$

4   $\widehat{\nabla \mathcal{L}}_0 \leftarrow (u_{\boldsymbol{\theta}_0}(x_i) - f(x_i))_i$ for $i \in 1, \ldots, S$

5   Compute $(\mathrm{RCE}_j^S)$ for all $j \in 1, \ldots r_{\text{svd}}$ following Equation (13)

6   **repeat**

     // Compute adaptive ranks

7      Compute $r_{\text{int}}$ and $r_\epsilon$ using expressions from Equation (20)

     // Determine a final cutoff rank

8      **if** $RCE_{r_{int}}^S > \epsilon$ **then**

9        $r_{\text{cutoff}} \leftarrow r_{\text{int}}$                                   // Track intersection

10     **else**

11        $r_{\text{cutoff}} \leftarrow r_\epsilon$                                     // Lock on precision

     // Natural gradient step

12     Set $\widehat{\Delta}_t \leftarrow \begin{cases} \widehat{\Delta}_{t,i} & \text{if } i \leqslant r_{\text{cutoff}}, \\ 0 & \text{else}; \end{cases}$

13     Get new $\boldsymbol{\theta}_{t+1}$ after one ANaGRAM step with Equation (10)

     // Update for next iteration

14     $\widehat{\phi}_{t+1} \leftarrow \left(\partial_p u_{\boldsymbol{\theta}_{t+1}}(x_i)\right)_{i,p}$

15     $\widehat{U}_{t+1}, \widehat{\Delta}_{t+1}, \widehat{V}_{t+1}^T \leftarrow \mathrm{SVD}\left(\widehat{\phi}_{t+1}\right)$

16     $\widehat{\nabla \mathcal{L}}_{t+1} \leftarrow \left(u_{\boldsymbol{\theta}_{t+1}}(x_i) - f(x_i)\right)_i$

17     Recompute $\mathrm{RCE}_j^S$ for all $j \in 1, \ldots r_{\text{svd}}$ following Equation (13)

18     $t \leftarrow t + 1$

19   **until** $r_\epsilon = 0$ *or* $t \geqslant T_{\max}$

    **Output:** $\boldsymbol{\theta}_t$

---

## C.3 COMPLETE PRACTICAL ALGORITHM

---

**Algorithm 2:** AMStraMGRAM : Adaptive Multicutoff Strategy Modification for ANaGRAM

---

**Input:**• $u : \mathbb{R}^P \to \mathrm{L}^2(\Omega, \mu)$ `// neural network architecture`

    • $\boldsymbol{\theta}_0 \in \mathbb{R}^P$ `// initialization of the neural network`

    • $f \in \mathrm{L}^2(\Omega, \mu)$ `// target function of the quadratic regression`

    • $(x_i) \in \Omega^S$ `// a batch in` $\Omega$

    • $\epsilon > 0$ `// precision level of the optimization`

**1 begin** Initialization

**2**    $\lambda \leftarrow \textit{False}$ `// Liftoff indicator`

**3**    $\widehat{\phi}_{\boldsymbol{\theta}_0} \leftarrow (\partial_p u_{\boldsymbol{\theta}_0}(x_i))_{1 \leqslant i \leqslant S,\, 1 \leqslant p \leqslant P}$    `// Computed` *via* `auto-differentiation`

**4**    $\widehat{U}_{\boldsymbol{\theta}_0}, \widehat{\Delta}_{\boldsymbol{\theta}_0}, \widehat{V}^t_{\boldsymbol{\theta}_0} \leftarrow \mathtt{SVD}\left(\widehat{\phi}_{\boldsymbol{\theta}_0}\right)$

**5**    $\widehat{\nabla \mathcal{L}}_{\boldsymbol{\theta}_0} \leftarrow (u_{\boldsymbol{\theta}_0}(x_i) - f(x_i))_{1 \leqslant i \leqslant S}$

**6**    $\mathrm{RCE}^S_0 \leftarrow \mathtt{ReconstructionErrors}\left(\widehat{V}^t_{\boldsymbol{\theta}_0}, \widehat{\nabla \mathcal{L}}_{\boldsymbol{\theta}_0}\right)$

**7**    $\mathrm{r}_{\max 0} \leftarrow \mathtt{FindElbow}\left((1, \ldots, \mathrm{r}_{\mathrm{svd}}), \widehat{\Delta}_{\boldsymbol{\theta}_0}\right)$

**8 repeat**

**9**    $\mathrm{r}_{1t} \leftarrow \#\left\{ \mathrm{RCE}^S_{0_j} \leqslant \widehat{\Delta}_{\boldsymbol{\theta}_{t_j}} : 1 \leqslant j \leqslant \mathrm{r}_{\mathrm{svd}} \right\}$

**10**    $\mathrm{r}_{2t} \leftarrow \#\left\{ \mathrm{RCE}^S_{0_j} \geqslant \epsilon : 1 \leqslant j \leqslant \mathrm{r}_{\mathrm{svd}} \right\}$

     `/* with # standing for the cardinal`                        `*/`

**11**    $\mathrm{r}_{\min t} \leftarrow \min(\mathrm{r}_{1t}, \mathrm{r}_{2t})$

**12**    $\mathrm{r}_{\max t} \leftarrow \max(\mathrm{r}_{1t}, \mathrm{r}_{\max t-1})$

**13**    **if** *not* $\lambda_t$ **then**

**14**      **if** $\mathrm{r}_{\min t} \geqslant \mathrm{r}_{\max t}$ **then**

**15**        $\lambda_t \leftarrow \mathrm{True}$

**16**      **else if** $\mathrm{r}_{\min t-1} = \mathrm{r}_{\min t}$ **then**

**17**        $\mathrm{r}_{\max t} \leftarrow \mathrm{r}_{\max t} + 1$

**18**    **foreach** $r_{cutoff} \in \{r_{\max t}, r_{\min t}\}$ **do**

**19**      $\widehat{\Delta}_{\boldsymbol{\theta}_t} \leftarrow \left(\widehat{\Delta}_{\boldsymbol{\theta}_t,p} \text{ if } p \geqslant \mathrm{r}_{\mathrm{cutoff}} \text{ else } 0\right)_{1 \leqslant p \leqslant P}$

**20**      $\widehat{\nabla \mathcal{L}}_{\boldsymbol{\theta}_t} \leftarrow (u_{\boldsymbol{\theta}_t}(x_i) - f(x_i))_{1 \leqslant i \leqslant S}$

**21**      $d_{\boldsymbol{\theta}_t} \leftarrow \widehat{V}_{\boldsymbol{\theta}_t} \widehat{\Delta}^\dagger_{\boldsymbol{\theta}_t} \widehat{U}^t_{\boldsymbol{\theta}_t} \widehat{\nabla \mathcal{L}}_{\boldsymbol{\theta}_t}$

**22**      $\eta_t \leftarrow \arg\min_{\eta \in \mathbb{R}^+} \sum_{1 \leqslant i \leqslant S} \left(f(x_i) - u_{\boldsymbol{\theta}_t - \eta d_{\boldsymbol{\theta}_t}}(x_i)\right)^2$    `// via line search`

**23**      $\boldsymbol{\theta}_{t+1} \leftarrow \boldsymbol{\theta}_t - \eta_t d_{\boldsymbol{\theta}_t}$

**24**      $\widehat{\phi}_{\boldsymbol{\theta}_{t+1}} \leftarrow \left(\partial_p u_{\boldsymbol{\theta}_{t+1}}(x_i)\right)_{1 \leqslant i \leqslant S,\, 1 \leqslant p \leqslant P}$             `// Computed`
     *via* `auto-differentiation`

**25**      $\widehat{U}_{\boldsymbol{\theta}_{t+1}}, \widehat{\Delta}_{\boldsymbol{\theta}_{t+1}}, \widehat{V}^t_{\boldsymbol{\theta}_{t+1}} \leftarrow \mathtt{SVD}\left(\widehat{\phi}_{\boldsymbol{\theta}_{t+1}}\right)$

**26 until** $r_{1t} = 0$ *or* $t \geqslant T_{\max}$

---

## C.4 EMPIRICAL JUSTIFICATION FOR DESIGN CHOICES

The dual cutoff strategy addresses specific empirical challenges we observed:

**Dual gradient steps:** Without the second cutoff, training dynamics sometimes stagnate. The dual approach provides both stability (via $\mathrm{r}_{\min}$) and exploration (via $\mathrm{r}_{\max}$).

**Elbow initialization:** The elbow point marks where singular values cease contributing meaningful signal, providing a natural upper bound for exploration.

**Monotonic $\mathrm{r}_{\max}$:** Prevents catastrophic retreat of the intersection point, which we observed in complex equations like Allen-Cahn.

**Stage separation timing:** Triggered precisely when the intersection error drops below target precision, ensuring optimal utilization of the flattening phenomenon.

We see in the next section how this practical algorithm successfully improve empirical robustness.

# D ALGORITHMIC DETAILS

---

**Algorithm 3:** Find elbow

---
**1 Function** FindElbow
  **Input:** - $(x_i) \in \mathbb{R}^m$ // an increasing sequence of $m \in \mathbb{N}$ points in $\mathbb{R}$

       - $\widehat{f} \in \mathbb{R}^m$ // a decreasing function evaluated at points $(x_i)$

       /* Clockwise normal vector to $\left(x_m - x_1, \widehat{f}_m - \widehat{f}_1\right)$            */

**2**    $\overrightarrow{n} \leftarrow \left(\widehat{f}_m - \widehat{f}_1, x_1 - x_m\right) \in \mathbb{R}^2$

**3**    $(s_j)_{1 \leqslant j \leqslant m} \leftarrow \left(\left\langle \overrightarrow{n}, \left(x_j - x_1, \widehat{f}_j - \widehat{f}_1\right)\right\rangle_{\mathbb{R}^2}\right)_{1 \leqslant j \leqslant m}$

  **Output:** $\underset{1 \leqslant j \leqslant m}{\arg\max} \, s_j$

**4 end**

---

**Algorithm 4:** Reconstruction Errors

---
**1 Function** ReconstructionErrors
  **Input:** - $\widehat{V}^t \in \mathbb{R}^{r_{svd}, S}$ // right singular vectors of the Jacobian $\widehat{\phi}$

       - $\widehat{\nabla\mathcal{L}} \in \mathbb{R}^S$ // Evaluated functional gradient

**2**    **begin** Initialization

**3**      $\widehat{\Sigma} \leftarrow 0 \in \mathbb{R}^S$ // cumulative approximation of $\widehat{\nabla\mathcal{L}}$

**4**      $\text{RCE}^S \leftarrow 0 \in \mathbb{R}^{r_{svd}}$ // cumulated reconstruction erros

**5**      $\widehat{c} \leftarrow \widehat{V}^t \widehat{\nabla\mathcal{L}} \in \mathbb{R}^{r_{svd}}$

**6**    **end**

**7**    **foreach** $j \in (1, \ldots, r_{svd})$ **do**

**8**      $\widehat{\Sigma} \leftarrow \widehat{\Sigma} + \widehat{c}_j$

**9**      $\text{RCE}_j^S \leftarrow \left\|\widehat{\Sigma} - \widehat{c}\right\|_2$

**10**    **end**

  **Output:** $\text{RCE}^S$

**11 end**

---

# E EMPIRICAL EXAMPLE OF ANAGRAM TRAINING DYNAMICS

In Figure 6, we analyze ANaGRAM's training on the heat equation with a fixed cutoff threshold $\alpha = 10^{-3}$ and line search for the learning rate. The training loss coincides with $\left\|\widehat{\nabla\mathcal{L}}\right\|^2$. We can see the flattening phenomenon to occur on Iteration 120 and completed at 150. As discussed in the main paper, sometimes the flattening can be incomplete, and for many iterations remain without any further progress ($N_{\text{flat}}$ never reaching zero). In this case, changing a cutoff threshold results in an immediate and complete flattening for all first components up to $r_{\text{cutoff}}$, which is demonstrated in Figure 7 for Iteration 120 of Figure 6.

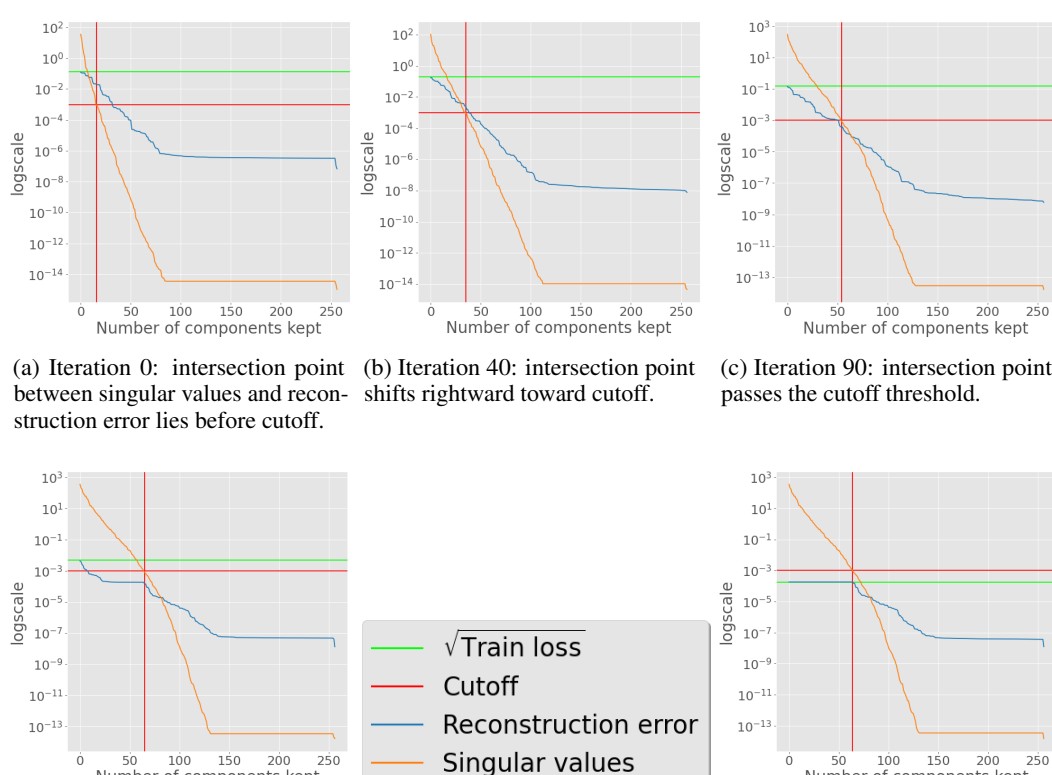

(a) Iteration 0: intersection point between singular values and reconstruction error lies before cutoff.

(b) Iteration 40: intersection point shifts rightward toward cutoff.

(c) Iteration 90: intersection point passes the cutoff threshold.

(d) Iteration 120. Beginning of *flattening*: reconstruction errors stabilizes at constant level before cutoff.

(e) Iteration 150: Complete flattening. Training loss reaches the flattened reconstruction error level.

Figure 6: **Evolution of quantities of interest during ANaGRAM training on heat equation.** The dynamics reveal two distinct phases culminating in reconstruction error flattening.

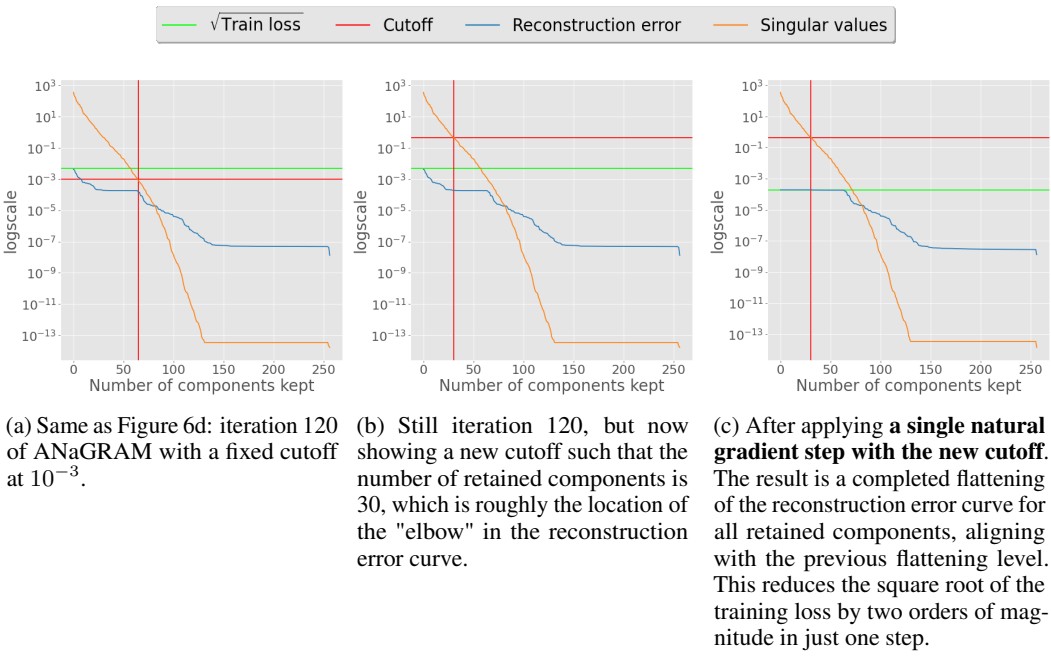

(a) Same as Figure 6d: iteration 120 of ANaGRAM with a fixed cutoff at $10^{-3}$.

(b) Still iteration 120, but now showing a new cutoff such that the number of retained components is 30, which is roughly the location of the "elbow" in the reconstruction error curve.

(c) After applying **a single natural gradient step with the new cutoff**. The result is a completed flattening of the reconstruction error curve for all retained components, aligning with the previous flattening level. This reduces the square root of the training loss by two orders of magnitude in just one step.

Figure 7: Illustration of "instant flattening" through adaptive cutoff adjustment. A single step with adjusted cutoff completes the flattening process.

# F   DEEP DIVE ON SELECTED EXPERIMENTS

In this section we look at curves of training and estimations obtained with AMStraMGRAM on benchmark of PDEs.

## F.1   ONE DIMENSIONAL BURGERS EQUATION

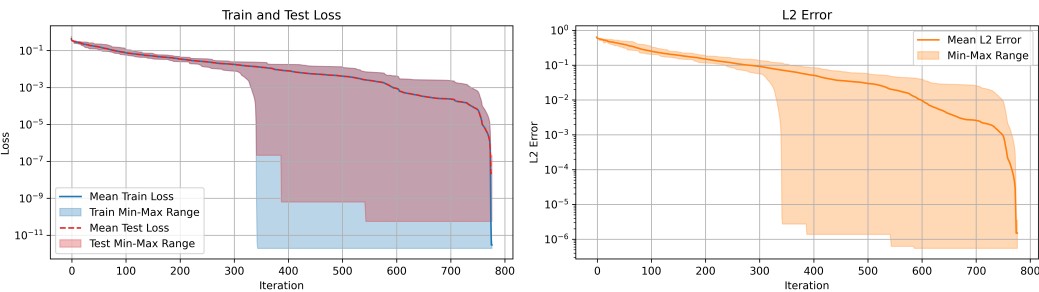

Figure 8: Training metrics for the One-Dimensional Burgers equation, showing convergence behavior with our adaptive multi-cutoff strategy.

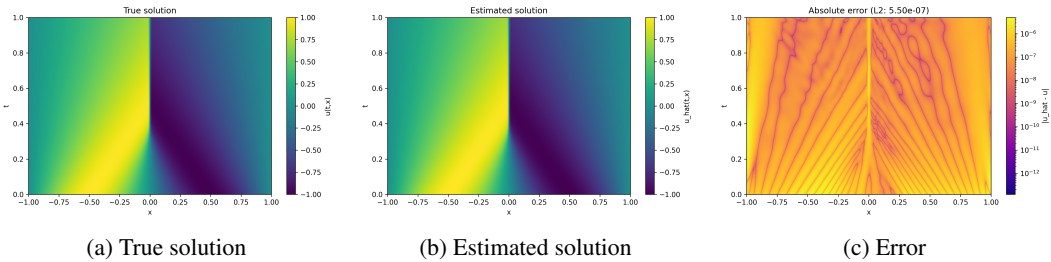

(a) True solution          (b) Estimated solution          (c) Error

Figure 9: Results for One Dimensional Burgers Equation with cutoff $10^{-6}$.

## F.2   HEAT EQUATION

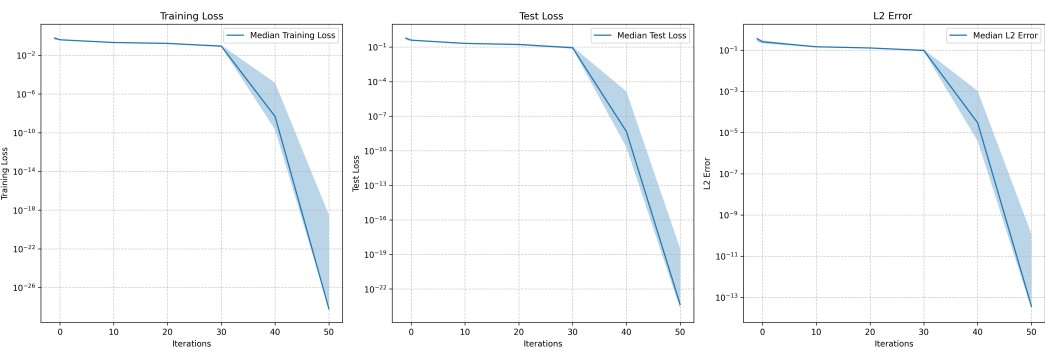

Figure 10: Convergence results for the Heat equation showing the $L_2$ error over iterations. Our method (AMStraMGRAM) converges faster and reaches a lower final error than ANaGRAM and baselines. Variability across runs is due to differing feature development speed from the random initialization.

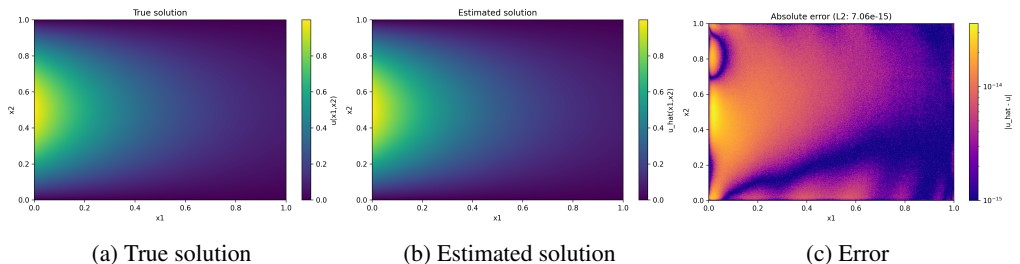

(a) True solution       (b) Estimated solution       (c) Error

Figure 11: Results for the Heat equation (solution cutoff $10^{-14}$). The error remains uniformly low over the domain, illustrating the effectiveness of the adaptive multi-cutoff strategy.

### F.3 LAPLACE EQUATIONS (L2D AND L5D)

For the Laplace equation in 2D, our method also demonstrates remarkable performance improvements over the baselines. The convergence is both faster and reaches a significantly lower error plateau.

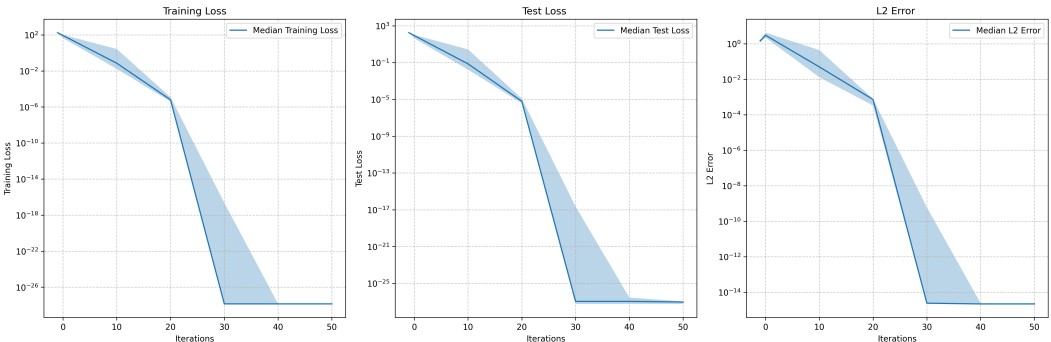

Figure 12: Convergence results for the Laplace 2D problem, showing the $L_2$ error over iterations. Our method (AMStraMGRAM) achieves both faster convergence and lower final error compared to ANaGRAM and other baseline methods. The observed variance between runs can be explained by different speed of convergence depending on the initialization.

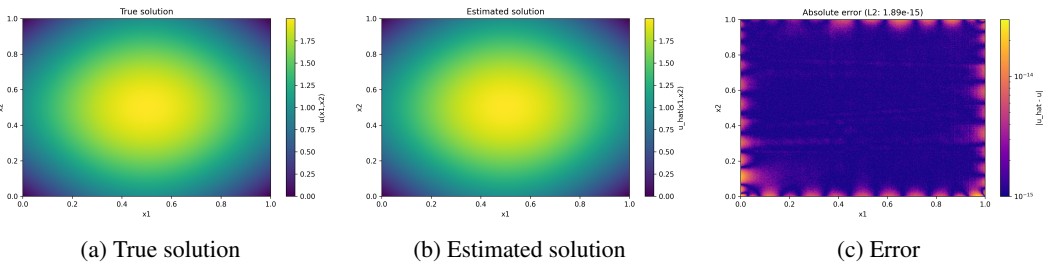

(a) True solution       (b) Estimated solution       (c) Error

Figure 13: Results for Laplace 2D Equation with cutoff $10^{-6}$.

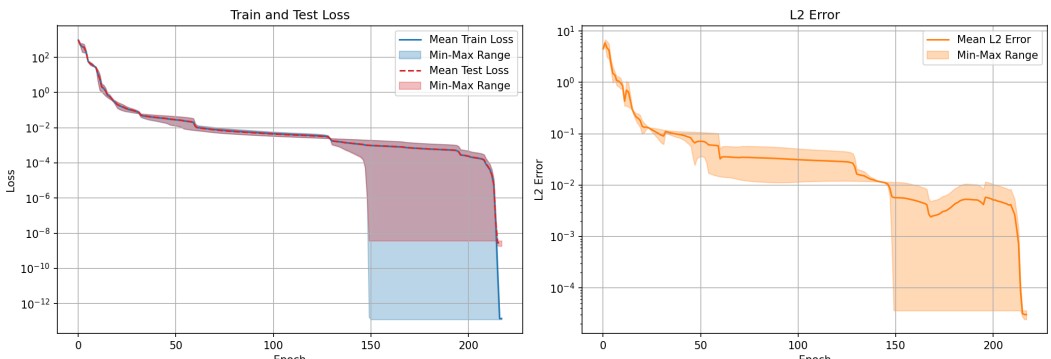

Figure 14: Convergence results for the Laplace 5D problem, showing the $L_2$ error over iterations. Our method (AMStraMGRAM) achieves faster convergence but not lower final error compared to ANaGRAM and other baseline methods. We see that seeds change the speed of convergence of the algorithm

### F.4 NON LINEAR POISSON EQUATION

To compare ourselves with Urbán et al. (2025), we select (K=1).

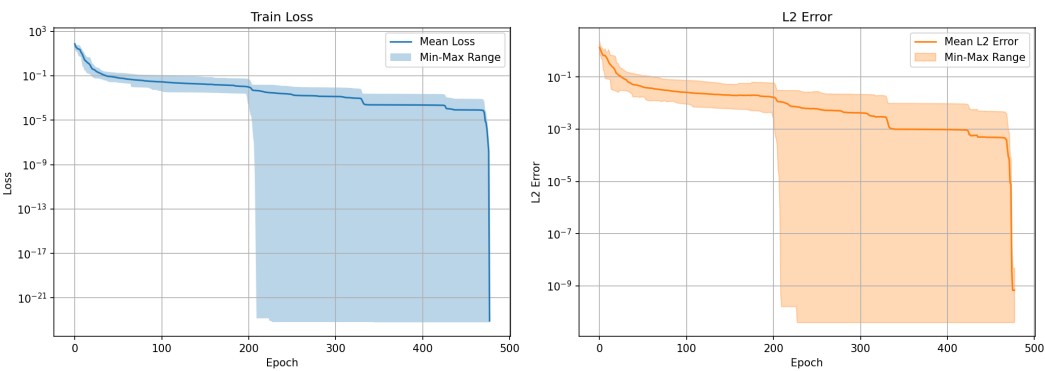

Figure 15: Convergence results for the Non Linear Poisson equation, showing the $L_2$ error over iterations. Our method (AMStraMGRAM) achieves both faster convergence and lower final error compared to ANaGRAM and other baseline methods.

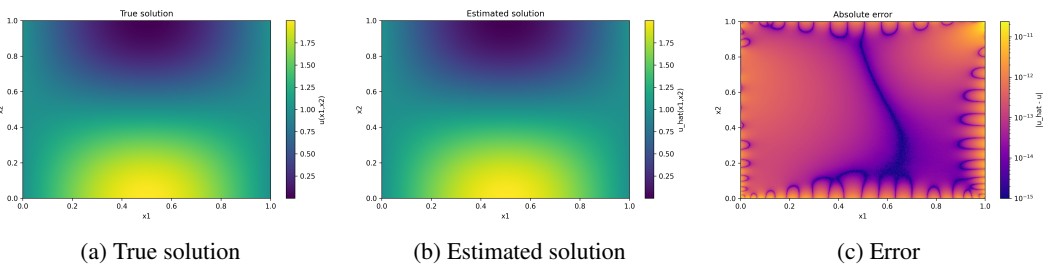

(a) True solution      (b) Estimated solution      (c) Error

Figure 16: Results for the Nonlinear Poisson equation (cutoff $10^{-4}$).

## F.5 ALLEN-CAHN EQUATION

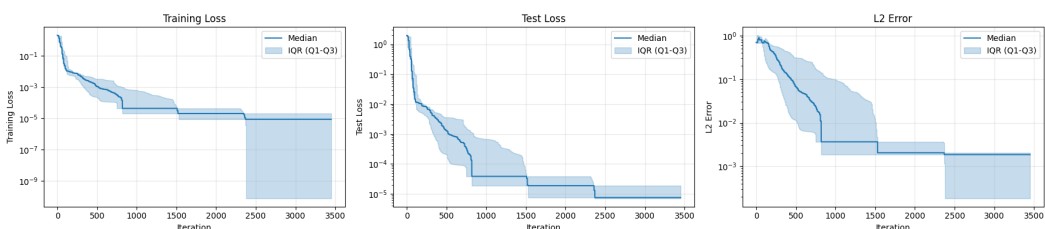

Figure 17: Training curves for the Allen-Cahn equation, showing the evolution of loss and error over iterations.

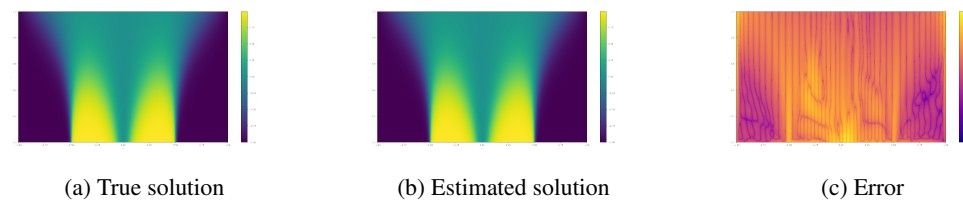

(a) True solution         (b) Estimated solution         (c) Error

Figure 18: Results on the Allen-Cahn equation, showing the error distribution (left), model prediction (middle), and true solution (right). The error is mostly present in regions with the "sharpest" transitions, which exemplifies the challenge of accurately capturing sharp interfaces still remains even for our advanced optimization approach.

## G GEOMETRICAL INTERPRETATION OF REGULARIZATIONS

### G.1 WHY REGULARIZATION IS NECESSARY

We recall that our goal is to solve the operator equation $D[u] = f$ by minimizing the squared residual

$$\|D[u] - f\|_{\mathrm{L}^2(\Omega,\mu)}^2. \tag{22}$$

For simplicity, assume $D$ is linear. Then the mapping

$$u \in \mathcal{C}^\infty(\Omega) \longmapsto \|D[u]\|_{\mathrm{L}^2(\Omega,\mu)} \tag{23}$$

defines a semi-norm on $\mathcal{C}^\infty(\Omega)$. We can "upgrade" this semi-norm into a true norm by introducing the following generalized Sobolev norm:

$$\|\cdot\|_{\widetilde{\mathcal{H}}_D} : \begin{cases} \mathcal{C}^\infty(\Omega \to \mathbb{R}) & \to & \mathbb{R}^+ \\ u & \mapsto & \sqrt{\|u\|_{\mathrm{L}^2(\Omega\to\mathbb{R},\mu)}^2 + \|D[u]\|_{\mathrm{L}^2(\Omega\to\mathbb{R},\mu)}^2} \end{cases} \tag{24}$$

Clearly, for any $u$,

$$\|u\|_{\mathrm{L}^2(\Omega,\mu)} \leqslant \|u\|_{\widetilde{\mathcal{H}}_D}, \tag{25}$$

which guarantees that $\|\cdot\|_{\widetilde{\mathcal{H}}_D}$ is *definite*, i.e. $\|u\|_{\widetilde{\mathcal{H}}_D} = 0 \iff u = 0$.

Completing $\mathcal{C}^\infty(\Omega)$ with respect to $\|\cdot\|_{\widetilde{\mathcal{H}}_D}$ yields a generalized Sobolev space $\left(\mathcal{H}_D, \|\cdot\|_{\mathcal{H}_D}\right)$. This Hilbert space is the largest subspace of $\mathrm{L}^2(\Omega, \mu)$ on which $D$ is continuous. Indeed, for every $u \in \mathcal{H}_D$,

$$\|D[u]\|_{\mathrm{L}^2(\Omega,\mu)} \leqslant \|u\|_{\mathcal{H}_D}. \tag{26}$$

Since our goal is to solve $D[u] = f$, we need $D$ to be continuously invertible. That is, we need the reverse inequality of Equation (26) to hold (up to a constant $\alpha > 0$). Formally, if $D$ were algebraically

invertible (bijective as a mapping), this condition would read:

$$\left( \exists \alpha > 0, \forall u \in \mathcal{H}_D, \|u\|_{\mathcal{H}_D} \leqslant \alpha \|D[u]\|_{\mathrm{L}^2(\Omega \to \mathbb{R}, \mu)} \right)$$

$$\iff \left( \exists \alpha > 0, \forall u \in \mathcal{H}_D, \|D^{-1}[D[u]]\|_{\mathcal{H}_D} \leqslant \alpha \|D[u]\|_{\mathrm{L}^2(\Omega \to \mathbb{R}, \mu)} \right) \quad . \tag{27}$$

$$\iff \left( \exists \alpha > 0, \forall f \in \mathrm{L}^2(\Omega \to \mathbb{R}, \mu), \|D^{-1}[f]\|_{\mathcal{H}_D} \leqslant \alpha \|f\|_{\mathrm{L}^2(\Omega \to \mathbb{R}, \mu)} \right)$$

**Operator ill-conditioning.** Even if $D$ is bijective, Equation (27) may fail to hold, i.e. $D$ can be ill-conditioned. Suppose there exists a subspace $\mathcal{H}_K \subset \mathcal{H}_D$ such that $D$ acts compactly on $\mathcal{H}_K$ with infinite rank. Then $D$ admits a singular value decomposition (Kress, 2014, Theorem 15.16): for $u \in \mathcal{H}_K$,

$$D[u] = \sum_{n \in \mathbb{N}} e_n \lambda_n \langle v_n , u \rangle_{\mathcal{H}_D} , \tag{28}$$

with $(v_n)$ orthonormal in $\mathcal{H}_D$, $(e_n)$ orthonormal in $\mathrm{L}^2(\Omega, \mu)$, and $\lambda_n \to 0$ as $n \to \infty$.

For Equation (27) to hold, we would need $\inf_n \lambda_n > 0$, contradicting $\lambda_n \to 0$. This is exactly the classical inverse problem setting: $D$ is bijective but ill-conditioned, and regularization is unavoidable. Among the many schemes developed, Tikhonov regularization is the canonical example (Kirsch, 2021).

**Non-bijectivity.** If $D$ is not bijective, two additional issues may occur.

NON-SURJECTIVITY. If $\mathrm{Im}\, D$ is a closed subspace, we can still obtain a solution by replacing the target $f$ with its projection $\Pi_{\mathrm{Im}\, D} f$. Note that minimizing $\|D[u] - f\|^2_{\mathrm{L}^2(\Omega, \mu)}$ yields precisely this least-squares solution.

NON-INJECTIVITY. The lack of injectivity is a much more subtle issue. Since $D$ is linear and continuous, its null space $\mathrm{Ker}\, D$ is a closed subspace of $\mathcal{H}_D$. In principle, one could restrict the domain of $D$ to $\mathrm{Ker}\, D^{\perp}$ to make it injective. The problem, however, is that identifying $\mathrm{Ker}\, D$ is typically just as hard as solving the original problem itself, since it amounts to characterizing all $u \in \mathcal{H}_D$ such that $D[u] = 0$. Therefore, unless one can rely on theoretical results that explicitly describe $\mathrm{Ker}\, D$, or construct a subspace $\mathcal{H}_0 \subset \mathcal{H}_D$ for which $\mathrm{Ker}\, D \cap \mathcal{H}_0$ is explicitly known (so that $D$ can be restricted to $\mathcal{H}_0$), it is generally impossible to "get rid of" $\mathrm{Ker}\, D$ in practice.

On the other hand, if we do not filter out $\mathrm{Ker}\, D$, this has the unwanted consequence of introducing "spurious" low-energy signals. To be concrete, suppose we approximate our solution in a space $\mathcal{H}_K$ with orthonormal basis $(u_n)_{n \in \mathbb{N}}$. Assume there exists a subsequence $(u_n^S) \notin \mathrm{Ker}\, D$ converging towards $\mathrm{Ker}\, D$. Since $\mathrm{Ker}\, D$ is closed (by continuity of $D$), this means

$$\lim_{n \to \infty} \|\Pi_{\mathrm{Ker}\, D} u_n^S - u_n^S\|^2_{\mathcal{H}_D} = 0. \tag{29}$$

Equivalently, after extraction, this can be rewritten for all $n \in \mathbb{N}$ as

$$\frac{\|\Pi_{\mathrm{Ker}\, D} u_n^S\|^2_{\mathcal{H}_D}}{\|u_n^S\|^2_{\mathcal{H}_D}} \geqslant 1 - 2^{-n}. \tag{30}$$

Now consider normalized vectors $u_n^S / \|u_n^S\|_{\mathcal{H}_D}$. We have

$$
\begin{aligned}
0 < \left\| D\left[ \tfrac{u_n^S}{\|u_n^S\|_{\mathcal{H}_D}} \right] \right\|_{\mathcal{H}_D}^2 &= \left\| D\left[ \tfrac{\Pi_{\operatorname{Ker} D^\perp} u_n^S + \Pi_{\operatorname{Ker} D} u_n^S}{\|u_n^S\|_{\mathcal{H}_D}} \right] \right\|_{\mathcal{H}_D}^2 \\
&= \left\| D\left[ \tfrac{\Pi_{\operatorname{Ker} D^\perp} u_n^S}{\|u_n^S\|_{\mathcal{H}_D}} \right] + \underbrace{D\left[ \tfrac{\Pi_{\operatorname{Ker} D} u_n^S}{\|u_n^S\|_{\mathcal{H}_D}} \right]}_{=0} \right\|_{\mathcal{H}_D}^2 \\
&= \left\| D\left[ \tfrac{\Pi_{\operatorname{Ker} D^\perp} u_n^S}{\|u_n^S\|_{\mathcal{H}_D}} \right] \right\|_{\mathcal{H}_D}^2 \\
&\overset{(26)}{\leqslant} \left\| \tfrac{\Pi_{\operatorname{Ker} D^\perp} u_n^S}{\|u_n^S\|_{\mathcal{H}_D}} \right\|_{\mathcal{H}_D}^2 \\
&= 1 - \frac{\|\Pi_{\operatorname{Ker} D} u_n^S\|_{\mathcal{H}_D}^2}{\|u_n^S\|_{\mathcal{H}_D}^2} \\
&\overset{(30)}{\leqslant} 2^{-n}.
\end{aligned}
\tag{31}
$$

In particular, if (for simplicity) the normalized $(u_n / \|u_n\|_{\mathcal{H}_D})$ are right singular vectors of $D$, then the vectors $\left( u_n^S / \|u_n^S\|_{\mathcal{H}_D} \right)$ will correspond to singular values vanishing at least as fast as $(2^{-n})$. Crucially, however, these vanishing singular values do not reflect an intrinsic ill-conditioning of $D$, but rather an *artificial* ill-conditioning induced by the choice of approximation space $\mathcal{H}_K$. In other words, the spurious instability arises from how we approximate the operator, not from the operator itself. For more details on this approximation-induced phenomenon, see Adcock & Huybrechs (2019; 2020).

These remarks highlight the *inevitable need for regularization* in practice. In the next section, we will provide a geometric interpretation of the two regularization schemes introduced in Section 2.5, emphasizing how fundamentally different they are in nature.

*Remark* 4. The above discussion becomes even more critical when we restrict ourselves to a finite-dimensional approximation space $\mathcal{H}_{\mathrm{app}} \subset \mathcal{H}_D$. In this case, the restriction $D_{\mathrm{app}}$ is automatically compact, since it is of finite rank. As a consequence, both types of ill-conditioning described above may occur simultaneously. This highlights once again why regularization is not merely convenient but *unavoidable* in numerical practice.

## G.2 RIDGE-REGRESSION

Returning to the definition given in Section 2.5, recall that Ridge regression amounts to adding $\alpha^2 I_d$ (for some $\alpha > 0$) to the Gram matrix $G_{\boldsymbol{\theta}}$ introduced in Equation (9):

$$
\boldsymbol{\theta}_{t+1} \leftarrow \boldsymbol{\theta}_t - \eta\, G_{\boldsymbol{\theta}_t}^\dagger \nabla \ell(\boldsymbol{\theta}_t)\,; \qquad\qquad G_{\boldsymbol{\theta}_t\, p,q} := \langle \partial_p u_{\boldsymbol{\theta}_t}\,,\, \partial_q u_{\boldsymbol{\theta}_t} \rangle_{\mathrm{L}^2(\Omega,\mu)}\,.
\tag{9}
$$

We can reformulate this observation in the following way: given our model

$$
u : \mathbb{R}^P \to \mathrm{L}^2(\Omega \to \mathbb{R}, \mu),
\tag{32}
$$

consider the *regularized model*

$$
u^\alpha : \left\{ \begin{array}{ccl} \mathbb{R}^P & \to & \mathrm{L}^2(\Omega,\mu) \times \mathbb{R}^P \\ \boldsymbol{\theta} & \mapsto & (u_{\boldsymbol{\theta}}, \alpha\, \boldsymbol{\theta}). \end{array} \right.
\tag{33}
$$

The Gram matrix of this regularized model is exactly $G_{\boldsymbol{\theta}} + \alpha^2 I_d$. Suppose further that regression is performed with respect to some function $f \in \mathrm{L}^2(\Omega,\mu)$. Then we must adapt the objective to the regularized model, replacing $f$ with the pair

$$
(f, \alpha\, \boldsymbol{\theta}) \in \mathrm{L}^2(\Omega,\mu) \times \mathbb{R}^P.
\tag{34}
$$

A straightforward computation shows that, for all $1 \leqslant p \leqslant \min(P, S)$,

$$\langle \partial_p u_{\boldsymbol{\theta}}^{\alpha}, (f, \alpha\,\boldsymbol{\theta}) - u_{\boldsymbol{\theta}}^{\alpha}\rangle_{\mathrm{L}^2(\Omega,\mu)\times\mathbb{R}^P} = \langle \partial_p u_{\boldsymbol{\theta}}, f - u_{\boldsymbol{\theta}}\rangle_{\mathrm{L}^2(\Omega,\mu)} + \alpha\underbrace{\left\langle \boldsymbol{e}^{(p)}, \boldsymbol{\theta} - \boldsymbol{\theta}\right\rangle_{\mathbb{R}^P}}_{=0} \tag{35}$$

$$= \langle \partial_p u_{\boldsymbol{\theta}}, f - u_{\boldsymbol{\theta}}\rangle_{\mathrm{L}^2(\Omega,\mu)}.$$

Thus, regression of $(f, \alpha\,\boldsymbol{\theta})$ with the regularized model is exactly equivalent to Ridge regression. Equivalently, Ridge regression corresponds to replacing the original model $u$ by the regularized model $u^{\alpha}$, and replacing the objective $f$ by $(f, \alpha\,\boldsymbol{\theta})$. From this point of view, the choice of $\alpha\,\boldsymbol{\theta}$ as the secondary target may be interpreted as a *default assumption* in the absence of prior information on the parameters: one simply uses the current parameters as a reference target.

We can now extract several fundamental facts:

1. As $\alpha \to 0$, the regularized model $u^{\alpha}$ tends in operator norm to the unregularized model $(u, 0)$ (i.e. $u$ by abuse of notation). Indeed,

$$\sup_{\|\boldsymbol{\theta}\|_{\mathbb{R}^P}=1} \|\mathrm{d}u_{\boldsymbol{\theta}}^{\alpha} - (\mathrm{d}u_{\boldsymbol{\theta}}, 0)\|_{\mathrm{L}^2(\Omega,\mu)\times\mathbb{R}^P} = \alpha \sup_{\|\boldsymbol{\theta}\|_{\mathbb{R}^P}=1} \|\boldsymbol{\theta}\|_{\mathbb{R}^P} = \alpha. \tag{36}$$

2. The model $u^{\alpha}$ is injective and continuous. Since $\mathrm{d}u_{\boldsymbol{\theta}}$ is continuous (as $\mathbb{R}^P$ is finite-dimensional), the only possible source of non-injectivity is $\mathrm{Ker}\,\mathrm{d}u_{\boldsymbol{\theta}}^{\alpha}$. But

$$\mathrm{Ker}\,\mathrm{d}u_{\boldsymbol{\theta}}^{\alpha} = \mathrm{Ker}\,\mathrm{d}u_{\boldsymbol{\theta}} \cap \mathrm{Ker}(\alpha I_{\mathbb{R}^P}) \subset \mathrm{Ker}(\alpha I_{\mathbb{R}^P}) = \{0\}, \tag{37}$$

hence injectivity. Restricting $u^{\alpha}$ to its image makes it algebraically bijective, and the inverse is continuous since

$$\alpha \|\boldsymbol{\theta}\|_{\mathbb{R}^P} \leqslant \|\mathrm{d}u_{\boldsymbol{\theta}}^{\alpha}\|_{\mathrm{L}^2(\Omega,\mu)\times\mathbb{R}^P}. \tag{38}$$

By the equivalence stated in Equation (27), this implies that $(\mathrm{d}u_{\boldsymbol{\theta}}^{\alpha})^{-1}$ is continuous. Consequently, $\mathrm{Im}\,\mathrm{d}u_{\boldsymbol{\theta}}^{\alpha}$ is closed in $\mathrm{L}^2(\Omega,\mu) \times \mathbb{R}^P$, since it is the inverse image of a closed set under $(\mathrm{d}u_{\boldsymbol{\theta}}^{\alpha})^{-1}$. Therefore least-squares solution is well-defined.

3. The least-squares solution of $u^{\alpha} = (f, 0)$ is influenced by $\alpha$ as follows: $(f, 0)$ is projected onto

$$\mathrm{Im}\,\mathrm{d}u_{\boldsymbol{\theta}}^{\alpha} = \mathrm{Span}\left((\partial_p u_{\boldsymbol{\theta}}, \alpha e^{(p)}) : 1 \leqslant p \leqslant P\right). \tag{39}$$

In particular, even if $f \in \mathrm{Im}\,\mathrm{d}u_{\boldsymbol{\theta}}$ and $f \neq 0$, we still have $(f, 0) \notin \mathrm{Im}\,\mathrm{d}u_{\boldsymbol{\theta}}^{\alpha}$ (since $\mathrm{d}u_{\boldsymbol{\theta}}(0) = 0$). Consequently,

$$\left(\Pi_{\mathrm{Im}\,\mathrm{d}u_{\boldsymbol{\theta}}^{\alpha}}^{\perp}(f, 0)\right)_1 \neq f, \tag{40}$$

where the subscript 1 denotes projection onto the first component in $\mathrm{L}^2(\Omega,\mu) \times \mathbb{R}^P$.

We illustrate these phenomena in Figure 19a.

Building on the above analysis, we now show that Ridge regression can be extended to the functional setting. To this end, let us reconsider the operator $D : \mathcal{H}_D \to \mathrm{L}^2(\Omega,\mu)$ introduced in Section G.1. Analogously to what we did for the parametric model $u$, we define the *regularized operator* at level $\alpha > 0$ as

$$D^{\alpha} : \begin{cases} \mathcal{H}_D & \to & \mathrm{L}^2(\Omega,\mu) \times \mathcal{H}_D \\ u & \mapsto & (D[u], \alpha u) \end{cases}. \tag{41}$$

The corresponding target becomes the *regularized objective* $(f, \alpha u)$.

At this level of generality, the equivalence with Gram-matrix regularization no longer holds, since we are dealing with infinite-dimensional operators for which no direct Gram-matrix representation exists. Nevertheless, the fundamental properties remain valid, namely:

1. When $\alpha \to 0$, the regularized operator $D^{\alpha}$ converges to $(D, 0)$ in the operator-norm sense, i.e. to $D$ by a mild abuse of notation. Indeed, we have

$$\sup_{\|u\|_{\mathcal{H}_D}=1} \|D^{\alpha}[u] - (D, 0)\|_{\mathrm{L}^2(\Omega,\mu)\times\mathcal{H}_D} = \alpha \sup_{\|u\|_{\mathcal{H}_D}=1} \|u\|_{\mathcal{H}_D} = \alpha. \tag{42}$$

2. The operator $D^\alpha$ is injective and continuous. Indeed, $D$ is continuous by the very construction of $\mathcal{H}_D$ (see Section G.1), and injectivity follows since

$$\operatorname{Ker} D^\alpha = \operatorname{Ker} D \cap \operatorname{Ker}(\alpha I_{\mathcal{H}_D}) \subseteq \operatorname{Ker}(\alpha I_{\mathcal{H}_D}) = \{0\}. \tag{43}$$

Restricting $D^\alpha$ to its image makes it algebraically bijective, and the inverse is continuous: we have $\alpha \|u\|_{\mathcal{H}_D} \leqslant \|D^\alpha[u]\|_{\mathrm{L}^2(\Omega,\mu) \times \mathcal{H}_D}$, which by the equivalence in Equation (27) implies that $\left(D^\alpha\right)^{-1}$ is continuous. Consequently, $\operatorname{Im} D^\alpha$ is closed in $\mathrm{L}^2(\Omega, \mu) \times \mathcal{H}_D$, since it is the inverse image of a closed set under $\left(D^\alpha\right)^{-1}$. Therefore least-squares solution is well-defined.

3. Least-squares solutions of the regularized problem $D^\alpha[u] = (f, 0)$ are impacted by $\alpha$ in the following way: we are projecting $(f, 0)$ onto

$$\operatorname{Im} D^\alpha = \operatorname{Span}\Big( (D[h], \alpha h) \, : \, h \in \mathcal{H}_D \Big). \tag{44}$$

In particular, even if $f \in \operatorname{Im} D$ with $f \neq 0$, we have $(f, 0) \notin \operatorname{Im} D^\alpha$ (since $D[0] = 0$), and hence

$$\left(\Pi_{\operatorname{Im} D^\alpha}^{\perp}(f, 0)\right)_1 \neq f, \tag{45}$$

where the subscript 1 denotes the first coordinate in $\mathrm{L}^2(\Omega, \mu) \times \mathcal{H}_D$.

We illustrate these phenomena in Figure 19b.

In summary, Ridge regression can be interpreted as a modification of the operator $D$, rendering it injective and continuously invertible on its image. However, this comes at a price: the regularized solutions are *never* exact solutions of the original equation $D[u] = f$, even when $\alpha$ is arbitrarily small, since we are in fact solving a different operator equation. This marks a fundamental distinction from cutoff regularization, which instead acts directly on the approximation space, as we shall see in the next section.

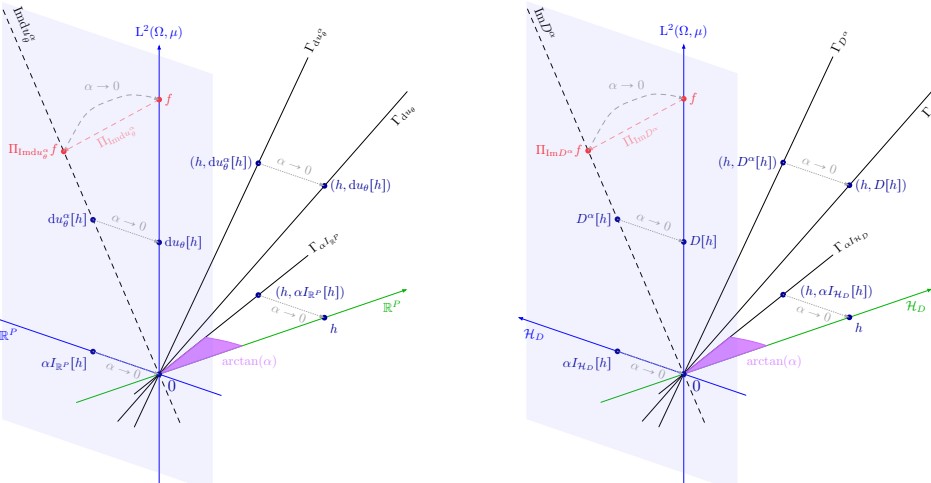

(a) **Illustration of parametric Ridge regression.** The green region represents the solution space, while the blue regions denote the target spaces. As $\alpha \to 0$, the regularized graph $\Gamma_{\mathrm{d}u_{\boldsymbol{\theta}}^{\alpha}}$ of $\mathrm{d}u_{\boldsymbol{\theta}}^{\alpha}$ approaches the graph $\Gamma_{\mathrm{d}u_{\boldsymbol{\theta}}}$ of $\mathrm{d}u_{\boldsymbol{\theta}}$, with the angle between them vanishing at rate $\arctan(\alpha)$. The key consequence is that the projection of the objective $f$ onto $\operatorname{Im} \mathrm{d}u_{\boldsymbol{\theta}}^{\alpha}$ follows a non-linear path as $\alpha \to 0$, coinciding with $\Pi_{\operatorname{Im} \mathrm{d}u_{\boldsymbol{\theta}}} f$ only asymptotically.

(b) **Illustration of functional Ridge regression.** The green region represents the solution space, while the blue regions denote the target spaces. As $\alpha \to 0$, the regularized graph $\Gamma_{D^\alpha}$ of $D^\alpha$ approaches the graph $\Gamma_{\mathcal{H}_D}$ of $D$, with the angle between them vanishing at rate $\arctan(\alpha)$. The key consequence is that the projection of the objective $f$ onto $\operatorname{Im} D^\alpha$ follows a non-linear path as $\alpha \to 0$, coinciding with $\Pi_{\operatorname{Im} D} f$ only asymptotically.

Figure 19: Illustrations of Ridge regression.

## G.3 Cutoff regression

As in Section G.2, let us return to the setting of Section 2.5. In Equation (11), we introduced cutoff regularization from the SVD perspective: given the differential $\mathrm{d}u_{\boldsymbol{\theta}}$ of the model $u$, at the point $\boldsymbol{\theta}$, and its singular value decomposition $\mathrm{d}u_{\boldsymbol{\theta}} = V_{\boldsymbol{\theta}} \Delta_{\boldsymbol{\theta}} U_{\boldsymbol{\theta}}^T$, the cutoff-regularized pseudo-inverse $\mathrm{d}u_{\boldsymbol{\theta}}^{\dagger_\alpha}$ at level $\alpha > 0$ is defined as

$$\mathrm{d}u_{\boldsymbol{\theta}}^{\dagger_\alpha} := U_{\boldsymbol{\theta}} \Delta_{\boldsymbol{\theta}}^{\dagger_\alpha} V_{\boldsymbol{\theta}}^T \; ; \qquad \Delta_{\boldsymbol{\theta},p}^{\dagger_\alpha} := \begin{cases} \Delta_{\boldsymbol{\theta},p}^{-1} & \text{if } \Delta_{\boldsymbol{\theta},p} \geqslant \alpha \\ 0 & \text{otherwise} \end{cases}, \; 1 \leqslant p \leqslant P. \tag{46}$$

Let us reinterpret this construction. Denote by $N_\alpha \in \mathbb{N}$ the number of singular values larger than $\alpha$. Equivalently, assuming $(\Delta_{\boldsymbol{\theta},p})_{1 \leqslant p \leqslant P}$ is non-increasing,

$$N_\alpha := \arg\max_{p \in \mathbb{N}} \{ \Delta_{\boldsymbol{\theta},p} \geqslant \alpha \}. \tag{47}$$

Define

$$\Theta_\alpha := \mathrm{Span}\{U_{\boldsymbol{\theta},p} : 1 \leqslant p \leqslant N_\alpha\}, \qquad T_{N_\alpha}^0 \mathcal{M} := \mathrm{Span}\{V_{t,p} : 1 \leqslant p \leqslant N_\alpha\}, \tag{48}$$

so that $T_{N_\alpha}^0 \mathcal{M} = \mathrm{d}u_{\boldsymbol{\theta}}(\Theta_\alpha)$. We then have

$$\left( \mathrm{d}u_{\boldsymbol{\theta}|\Theta_\alpha}^{|T_{N_\alpha}^0 \mathcal{M}} \right)^{-1} = \mathrm{d}u_{\boldsymbol{\theta}}^{\dagger_\alpha}, \tag{49}$$

meaning that the restriction $\mathrm{d}u_{\boldsymbol{\theta}}^\alpha := \mathrm{d}u_{\boldsymbol{\theta}|\Theta_\alpha}$ of $\mathrm{d}u_{\boldsymbol{\theta}}$ to the domain $\Theta_\alpha$ becomes invertible once its codomain is restricted to its image $T_{N_\alpha}^0 \mathcal{M}$, with inverse given precisely by the cutoff pseudo-inverse $\mathrm{d}u_{\boldsymbol{\theta}}^{\dagger_\alpha}$. Moreover, for any $h \in \Theta_\alpha$,

$$\|\mathrm{d}u_{\boldsymbol{\theta}}(h)\|_{\mathrm{L}^2(\Omega,\mu)} = \left\| V_{\boldsymbol{\theta}} \Delta_{\boldsymbol{\theta}} U_{\boldsymbol{\theta}}^T h \right\|_{\mathrm{L}^2(\Omega,\mu)} \overset{V_{\boldsymbol{\theta}} \text{ unitary}}{=} \left\| \Delta_{\boldsymbol{\theta}} U_{\boldsymbol{\theta}}^T h \right\|_{\mathbb{R}^P} \overset{h \in \Theta_\alpha}{\geqslant} \alpha \left\| U_{\boldsymbol{\theta}}^T h \right\|_{\mathbb{R}^P} \overset{U_{\boldsymbol{\theta}} \text{ unitary}}{=} \alpha \|h\|_{\mathbb{R}^P}. \tag{50}$$

In other words, Equation (27) is satisfied by $\mathrm{d}u_{\boldsymbol{\theta}}^\alpha$.

Thus, while ridge regularization modifies the model itself, cutoff regularization instead restricts the domain of the model so that, on this restricted domain, Equation (27) holds and the model becomes invertible. We summarize the fundamental properties:

1. We have

$$\bigcap_{\alpha > 0} \left( \mathbb{R}^P \backslash \Theta_\alpha \right) = \mathrm{Ker}\, \mathrm{d}u_{\boldsymbol{\theta}}, \tag{51}$$

   that is, $\lim_{\alpha \to 0} \mathbb{R}^P \backslash \Theta_\alpha = \mathrm{Ker}\, \mathrm{d}u_{\boldsymbol{\theta}}$, since for all $\alpha > \beta$ we have $\Theta_\alpha \subset \Theta_\beta$ and then $\mathbb{R}^P \backslash \Theta_\beta \subset \mathbb{R}^P \backslash \Theta_\alpha$. Similarly, $\lim_{\alpha \to 0} T_{N_\alpha}^0 \mathcal{M} = \mathrm{Im}\, \mathrm{d}u_{\boldsymbol{\theta}}$. Moreover, for each $\alpha > 0$, the restriction $\mathrm{d}u_{\boldsymbol{\theta}}^\alpha$ coincides with $\mathrm{d}u_{\boldsymbol{\theta}}$ on $\Theta_\alpha$.

2. By Equation (50), $\mathrm{d}u_{\boldsymbol{\theta}}^\alpha$ is injective and continuous. Restricting it to its image $T_{N_\alpha}^0 \mathcal{M}$ makes it bijective and bicontinuous, with inverse exactly the cutoff pseudo-inverse $\mathrm{d}u_{\boldsymbol{\theta}}^{\dagger_\alpha}$. In particular $\mathrm{d}u(\Theta_\alpha)$ is closed in $\mathrm{L}^2(\Omega,\mu)$, since it is the inverse image of a closed set under $\mathrm{d}u_{\boldsymbol{\theta}}^{\dagger_\alpha}$. Therefore least-squares solution is well-defined.

3. Solving the least-squares problem $\mathrm{d}u_{\boldsymbol{\theta}}^\alpha = f$ is now altered in the following way: the target $f$ is first projected onto $T_{N_\alpha}^0 \mathcal{M} = \mathrm{Im}\, \mathrm{d}u_{\boldsymbol{\theta}}^\alpha$. In particular, if for some $\alpha > 0$ we already have $f \in \mathrm{Im}\, \mathrm{d}u_{\boldsymbol{\theta}}^\alpha$, then the regularized least-squares formulation recovers an *exact solution* to the problem. This stands in sharp contrast with Ridge regression, where such exact recovery can only occur *asymptotically* in the limit $\alpha \to 0$.

As in Section G.2, we now need to reinterpret the cutoff regularization in order to extend it to the functional setting. Let us return once more to the operator $D : \mathcal{H}_D \to \mathrm{L}^2(\Omega,\mu)$ introduced in Section G.1. In general, one cannot define an SVD for such an operator (except when it is compact). We must therefore appeal to the spectral theorem for bounded self-adjoint operators, which relies on the notion of a *projection-valued measure* (also called a resolution of the identity). For our purposes, it will be sufficient to simply state the definition.

**Definition 1** (Projection-valued measure). Let $(X, \mathcal{A})$ be a measurable space, where $\mathcal{A}$ denotes its $\sigma$-algebra, and let $\mathcal{H}$ be a Hilbert space. A *projection-valued measure* (PVM) is a map

$$\pi : \mathcal{A} \to \mathcal{L}_b(\mathcal{H} \to \mathcal{H}),$$

where $\mathcal{L}_b(\mathcal{H} \to \mathcal{H})$ denotes the set of bounded operators on $\mathcal{H}$, such that for every $A \in \mathcal{A}$, $\pi(A)$ is an orthogonal projection on $\mathcal{H}$, and the following properties hold:

1. $\pi(\varnothing) = 0$ and $\pi(X) = I_{\mathcal{H}}$, where $I_{\mathcal{H}}$ is the identity operator on $\mathcal{H}$;

2. $\pi(A \cap B) = \pi(A)\pi(B)$ for all $A, B \in \mathcal{A}$;

3. For every countable family $(A_i)_{i=1}^{\infty}$ of disjoint sets in $\mathcal{A}$,

$$\pi\left(\bigcup_{i=1}^{\infty} A_i\right) = \sum_{i=1}^{\infty} \pi(A_i),$$

where the series converges in the strong operator topology.

Since projection-valued measures are measures, one can define integrals with respect to them. We refer to (Berezansky et al., 1996, Chapter 13) for details. We may now state the spectral theorem.

**Theorem 2.** *Let $\mathcal{H}$ be a Hilbert space and let $A : \mathcal{H} \to \mathcal{H}$ be a self-adjoint operator. Then there exists a projection-valued measure $\pi$ on the Borel $\sigma$-algebra of $\mathbb{R}$ such that*

$$A = \int_{\mathbb{R}} \lambda \, \pi(d\lambda) = \int_{\sigma(A)} \lambda \, \pi(d\lambda), \tag{52}$$

*where $\sigma(A)$ denotes the spectrum of $A$.*

A proof can be found in (Berezansky et al., 1996, Theorem 4.1, Section 4.1, Chapter 13). In particular, since $\pi$ is a projection-valued measure, we have by Definition 1:

$$I_{\mathcal{H}} = \int_{\mathbb{R}} \pi(\mathrm{d}\lambda). \tag{53}$$

Since $D$ is continuous, we can define its adjoint $D^* : \mathrm{L}^2(\Omega, \mu) \to \mathcal{H}_D$, and hence the self-adjoint operator $D^*D : \mathcal{H}_D \to \mathcal{H}_D$. Applying Theorem 2, we obtain a projection-valued measure $\pi_D$ on $\mathbb{R}$ endowed with its Borel $\sigma$-algebra, such that

$$D^*D = \int_{\mathbb{R}_+} \lambda \, \pi_D(\mathrm{d}\lambda), \qquad\qquad I_{\mathcal{H}_D} = \int_{\mathbb{R}_+} \pi_D(\mathrm{d}\lambda), \tag{54}$$

where the integration is restricted to $\mathbb{R}_+$ since $D^*D$ is a positive operator. We can then define

$$\Pi_D^{\alpha} := \int_{\alpha^2}^{+\infty} \pi_D(\mathrm{d}\lambda), \tag{55}$$

which is an orthogonal projection in $\mathcal{H}_D$ since $\pi_D$ is a projection-valued measure. We then define the regularized space $\mathcal{H}_D^{\alpha}$ at level $\alpha > 0$ by

$$\mathcal{H}_D^{\alpha} := \mathrm{Im}\,\Pi_D^{\alpha} \subset \mathcal{H}_D. \tag{56}$$

For any $u \in \mathcal{H}_D^\alpha$, we compute

$$\|D[u]\|_{\mathrm{L}^2(\Omega,\mu)}^2 = \langle D[u], D[u] \rangle_{\mathrm{L}^2(\Omega,\mu)} = \langle u, D^*D[u] \rangle_{\mathcal{H}_D}$$

$$= \left\langle u, \int_{\mathbb{R}_+} \lambda \pi_D(\mathrm{d}\lambda) u \right\rangle_{\mathcal{H}_D}$$

$$\stackrel{u \in \mathcal{H}_D^\alpha}{=} \left\langle u, \int_{\mathbb{R}_+} \lambda_1 \pi_D(\mathrm{d}\lambda_1) \int_{\alpha^2}^{+\infty} \pi_D(\mathrm{d}\lambda_2) u \right\rangle_{\mathcal{H}_D}$$

$$\stackrel{\pi_D \text{ PVM}}{=} \left\langle u, \int_{\alpha^2}^{+\infty} \lambda \pi_D(\mathrm{d}\lambda) u \right\rangle_{\mathcal{H}_D} \tag{57}$$

$$= \int_{\alpha^2}^{+\infty} \lambda \langle u, \pi_D(\mathrm{d}\lambda) u \rangle_{\mathcal{H}_D}$$

$$\geqslant \alpha^2 \int_{\alpha^2}^{+\infty} \langle u, \pi_D(\mathrm{d}\lambda) u \rangle_{\mathcal{H}_D} \stackrel{u \in \mathcal{H}_D^\alpha}{=} \alpha^2 \langle u, u \rangle_{\mathcal{H}_D} = \alpha^2 \|u\|_{\mathcal{H}_D}^2 .$$

That is,

$$\|D[u]\|_{\mathrm{L}^2(\Omega,\mu)} \geqslant \alpha \|u\|_{\mathcal{H}_D} , \tag{58}$$

so that Equation (27) is verified on $\mathcal{H}_D^\alpha$. We denote

$$D^\alpha := D_{|\mathcal{H}_D^\alpha}, \tag{59}$$

the restriction of $D$ to the domain $\mathcal{H}_D^\alpha$. We can now list the fundamental properties:

1. We have

$$\bigcap_{\alpha > 0} (\mathcal{H}_D \backslash \mathcal{H}_D^\alpha) = \operatorname{Ker} D \tag{60}$$

    that is, $\lim_{\alpha \to 0} \mathcal{H}_D \backslash \mathcal{H}_D^\alpha = \operatorname{Ker} D$, since for all $\alpha > \beta$, $\mathcal{H}_D^\alpha \subset \mathcal{H}_D^\beta$ by Property 3 of Definition 1. Moreover, by continuity of $D$, we also have $\lim_{\alpha \to 0} D[\mathcal{H}_D^\alpha] = \operatorname{Im} D$. Finally, for each $\alpha > 0$, $D^\alpha$ coincides with $D$ on $\mathcal{H}_D^\alpha$ by construction.

2. As established by Equation (27), $D^\alpha$ is injective and continuous. When restricted to its image, it is therefore bijective and bicontinuous, hence invertible. In particular $D[\mathcal{H}_D^\alpha]$ is closed in $\mathrm{L}^2(\Omega,\mu)$, since it is the inverse image of a closed set under $(D^\alpha)^{-1}$. Therefore least-squares solution is well-defined.

3. The least-squares solution of $D^\alpha = f$ is now modified as follows: one projects $f$ onto $\operatorname{Im} D^\alpha$. In particular, if for some $\alpha > 0$ we already have $f \in \operatorname{Im} D^\alpha$, then the regularized least-squares formulation recovers an *exact solution* to the problem $D[u] = f$. This stands in sharp contrast with Ridge regression, where such exact recovery can only occur *asymptotically* in the limit $\alpha \to 0$.

### G.4 Connection to Green's Function

To further highlight the difference between the two regularization schemes, we now reinterpret them through the lens of Green's functions of the operator $D$. Schwencke & Furtlehner (2025, Theorem 2) established in the finite-dimensional case a connection between the natural gradient for PINNs and Green's functions. Their proof relies on Schwencke & Furtlehner (2025, Proposition 3), which will be our starting point. We restate the relevant definitions and results for completeness.

**Definition 2** (Schwencke & Furtlehner, 2025, Definition 9: generalized Green's function). Let $\mathcal{H}$ be an Hilbert space, $D : \mathcal{H} \to \mathrm{L}^2(\Omega,\mu)$ be a linear differential operator, $\mathcal{H}_0 \subset \mathcal{H}$ a subspace isometrically embedded in $\mathcal{H}$ and $f \in \mathrm{L}^2(\Omega,\mu)$. A generalized Green's function of $D$ on $\mathcal{H}_0$ is then any kernel function $g : \Omega \times \Omega \to \mathbb{R}$ such that the operator:

$$R_{\mathcal{H}_0} : \begin{cases} \mathrm{L}^2(\Omega \to \mathbb{R}, \mu) & \to & \mathcal{H} \\ f & \mapsto & \left( x \in \Omega \mapsto \int_\Omega g(x,s) f(s) \mu(\mathrm{d}s) \right) \end{cases},$$

verifies the equation:

$$D \circ R_{\mathcal{H}_0} = \Pi_{D[\mathcal{H}_0]}^\perp \tag{61}$$

**Proposition 2** (Schwencke & Furtlehner, 2025, Proposition 3). *Let $D : \mathcal{H} \to L^2(\Omega, \mu)$ be a linear differential operator, and $\mathcal{H}_0 := \mathrm{Span}(u_p : 1 \leqslant p \leqslant P) \subset \mathcal{H}$ a subspace isometrically embedded in $\mathcal{H}$. Then the generalized Green's function of $D$ on $\mathcal{H}_0$ is given by: for all $x, y \in \Omega$*

$$g_{\mathcal{H}_0}(x, y) := \sum_{1 \leqslant p, q \leqslant P} u_p(x) \, G^\dagger_{p,q} D[u_q](y), \tag{62}$$

*with: for all $1 \leqslant p, q \leqslant P$,*

$$G_{p,q} := \langle D[u_p] \,, \, D[u_q] \rangle_{L^2(\Omega \to \mathbb{R}, \mu)}. \tag{63}$$

**Our goal.** We aim to

(i) generalize Schwencke & Furtlehner (2025, Proposition 3) to arbitrary Reproducing Kernel Hilbert Spaces;

(ii) establish a direct connection to the regularization framework introduced earlier. This will provide a novel reinterpretation of the Green's function in the regularized operator setting.

**Operator framework.** Consider the operator $D : \mathcal{H}_D \to L^2(\Omega, \mu)$ from Section G.1, and assume that there exists an RKHS $\mathcal{H}_0$ isometrically embedded in $\mathcal{H}_D$ (for instance, any finite-dimensional RKHS, see Schwencke & Furtlehner, 2025, Corollary 1). For Schwencke & Furtlehner (2025, Definition 9) to be well-posed, the range $D[\mathcal{H}_0]$ must be a closed subspace of $L^2(\Omega, \mu)$. As argued earlier, this is guaranteed if $D$ is continuously invertible: indeed, in this case

$$D[\mathcal{H}_0] = (D^{-1})^{-1}[\mathcal{H}_0], \tag{64}$$

and the inverse image of a closed subspace under a continuous operator is closed.

**Key observation.** Thus, to generalize Schwencke & Furtlehner (2025, Proposition 3), we require $D$ to be continuously invertible. Conveniently, this is precisely the property enforced by the regularization schemes we introduced earlier.

In what follows, we first focus on the cutoff regularization, which offers the clearest interpretation in terms of Green's functions. We then briefly revisit the case of Ridge regression. Before delving further into our main goal, let us first establish two general facts.

**Lemma 1.** *Let $\left(\mathcal{H}_0, \|\cdot\|_{\mathcal{H}_0}\right)$ be an RKHS on a set $X$ with reproducing kernel $k$. Suppose that $\|\cdot\|_{bis}$ is a norm equivalent to $\|\cdot\|_{\mathcal{H}_0}$. Then $\left(\mathcal{H}_0, \|\cdot\|_{bis}\right)$ is also an RKHS.*

*Proof.* The key point is to show that there exists a reproducing kernel for the inner product $\langle \cdot \,, \, \cdot \rangle_{bis}$ associated with $\|\cdot\|_{bis}$. Our argument follows the simple reasoning in Paulsen & Raghupathi (2016, Definitions 1–2).

Since, for every $x \in X$, the point evaluation functional

$$\delta_x : u \in \mathcal{H}_0 \mapsto u(x) \tag{65}$$

is continuous with respect to $\|\cdot\|_{\mathcal{H}_0}$ by the definition of an RKHS, it is also continuous with respect to the equivalent norm $\|\cdot\|_{bis}$ Therefore, by the Riesz representation theorem, for each $x \in X$, there exists a unique element $k^{bis}_x \in \mathcal{H}_0$ such that for all $u \in \mathcal{H}_0$

$$\langle k^{bis}_x \,, \, u \rangle_{bis} = u(x). \tag{66}$$

In particular, this defines a reproducing kernel for the norm $\|\cdot\|_{bis}$, given by

$$k_{bis}(x, y) = \langle k^{bis}_x \,, \, k^{bis}_y \rangle_{bis} = k^{bis}_x(y), \qquad \forall x, y \in X. \tag{67}$$

Hence $\left(\mathcal{H}_0, \|\cdot\|_{bis}\right)$ is indeed an RKHS. $\square$

**Lemma 2.** *Let $\mathcal{H}_A, \mathcal{H}_B$ be two Hilbert spaces. If $U : \mathcal{H}_A \to \mathcal{H}_B$ is an isometry, then*

$$U^*U = I_{\mathcal{H}_A}, \qquad\qquad UU^* = \Pi_{\mathrm{Im}\, U}. \tag{68}$$

*In particular $\mathrm{Im}\, U$ is closed in $\mathcal{H}_B$.*

*Proof.* The first identity follows from the fact that for all $x, y \in \mathcal{H}_\mathcal{A}$,

$$\langle x, U^*U[y]\rangle_{\mathcal{H}_\mathcal{A}} = \langle U[x], U[y]\rangle_{\mathcal{H}_\mathcal{B}} = \langle x, y\rangle_{\mathcal{H}_\mathcal{A}}. \tag{69}$$

Thus $(U^*U(y) - y) \in \mathcal{H}_\mathcal{A}^\perp$, *i.e.* $U^*U = \mathrm{I}_{\mathcal{H}_\mathcal{A}}$. For the second, the key point is to show that $\operatorname{Im} U$ is closed, i.e. $\operatorname{Im} U = \overline{\operatorname{Im} U}$.

Let $y \in \overline{\operatorname{Im} U}$, and $(y_n) \in \operatorname{Im} U^\mathbb{N}$ with $y_n \to y$. Since $(y_n)$ is Cauchy, and $y_n = U(x_n)$ for some $(x_n) \in \mathcal{H}_\mathcal{A}^\mathbb{N}$, we have

$$\|U(x_n) - U(x_m)\|_{\mathcal{H}_\mathcal{B}} = \|x_n - x_m\|_{\mathcal{H}_\mathcal{A}}, \tag{70}$$

so $(x_n)$ is also Cauchy and converges to $x \in \mathcal{H}_\mathcal{A}$, since $\mathcal{H}_\mathcal{A}$ is complete. Since $U$ is an isometry, we have for all $x \in \mathcal{H}_\mathcal{A}$

$$\|U(x)\|_{\mathcal{H}_\mathcal{B}} = \|x\|_{\mathcal{H}_\mathcal{A}}. \tag{71}$$

In particular, $U$ is bounded with operator norm $\|U\| = 1$, and hence continuous. Thus $U(x) = y$, hence $y \in \operatorname{Im} U$. We conclude that $\operatorname{Im} U$ is closed in $\mathcal{H}_\mathcal{B}$. Finally:

- For $y \in \operatorname{Im} U$, say $y = U(x)$, we have
$$UU^*(y) = U(U^*U)(x) = U(x) = y. \tag{72}$$

- For $y \in (\operatorname{Im} U)^\perp$, we check that $UU^*(y) = 0$. Indeed, for any $z \in \mathcal{H}_\mathcal{B}$,
$$\langle z, UU^*(y)\rangle_{\mathcal{H}_\mathcal{B}} = \langle UU^*(z), y\rangle_{\mathcal{H}_\mathcal{B}} = 0, \tag{73}$$
since $UU^*(z) \in \operatorname{Im} U$. Thus $UU^*(y) \in \mathcal{H}_\mathcal{B}^\perp$, *i.e.* $UU^*(y) = 0$. $\qquad\square$

We are interested in the restriction of $D$ to the domain $\mathcal{H}_0$. Since the restriction $D^*D : \mathcal{H}_D \to \mathcal{H}_D$ does not, *a priori*, map $\mathcal{H}_0$ into itself, we first need to adapt the setting in order to apply the spectral theorem of Theorem 2.

Because $\mathcal{H}_0 \subset \mathcal{H}_D$ isometrically, we have for all $u, v \in \mathcal{H}_0$:

$$\begin{aligned}
\langle D[u], D[v]\rangle_{\mathrm{L}^2(\Omega,\mu)} &= \langle D[\Pi_{\mathcal{H}_0} u], D[\Pi_{\mathcal{H}_0} v]\rangle_{\mathrm{L}^2(\Omega,\mu)} \\
&= \langle \Pi_{\mathcal{H}_0} u, D^*D[\Pi_{\mathcal{H}_0} v]\rangle_{\mathcal{H}_D} \\
&= \langle u, (\Pi_{\mathcal{H}_0} D^*D\Pi_{\mathcal{H}_0})[v]\rangle_{\mathcal{H}_D},
\end{aligned} \tag{74}$$

where we used in the last step that $\Pi_{\mathcal{H}_0}$ is self-adjoint.

We can therefore apply the spectral theorem Theorem 2 to the bounded self-adjoint operator $\Pi_{\mathcal{H}_0} D^*D\Pi_{\mathcal{H}_0} : \mathcal{H}_0 \to \mathcal{H}_0$, obtaining the analogue of the decomposition in Equation (54):

$$\Pi_{\mathcal{H}_0} D^*D\Pi_{\mathcal{H}_0} = \int_{\mathbb{R}_+} \lambda\, \pi_D^{\mathcal{H}_0}(\mathrm{d}\lambda), \qquad\qquad I_{\mathcal{H}_0} = \int_{\mathbb{R}_+} \pi_D^{\mathcal{H}_0}(\mathrm{d}\lambda). \tag{75}$$

**Regularized spaces.** Fixing $\alpha > 0$, and analogously to Equations (55) and (56), we define the regularized projection and subspace:

$$\Pi_{D,\mathcal{H}_0}^\alpha := \int_{\alpha^2}^{+\infty} \pi_D^{\mathcal{H}_0}(\mathrm{d}\lambda), \qquad\qquad \mathcal{H}_{D,\mathcal{H}_0}^\alpha := \operatorname{Im} \Pi_{D,\mathcal{H}_0}^\alpha \subset \mathcal{H}_0 \subset \mathcal{H}_D. \tag{76}$$

Let $k : \Omega \times \Omega \to \mathbb{R}$ be the reproducing kernel of $\mathcal{H}_0$. Then, by Paulsen & Raghupathi (2016, Theorem 2.5), $\mathcal{H}_{D,\mathcal{H}_0}^\alpha$ is an RKHS with reproducing kernel

$$k_\alpha(x, y) := \Pi_{D,\mathcal{H}_0}^\alpha[k(x, \cdot)](y), \qquad \forall x, y \in \Omega. \tag{77}$$

**Norm equivalence.** Since $\mathcal{H}_{D,\mathcal{H}_0}^\alpha \subset \mathcal{H}_0 \subset \mathcal{H}_D$, inequality in Equation (26) remains valid, i.e. for all $u \in \mathcal{H}_{D,\mathcal{H}_0}^\alpha$:

$$\|D[u]\|_{\mathrm{L}^2(\Omega,\mu)} \leqslant \|u\|_{\mathcal{H}_D}. \tag{78}$$

Furthermore, by an argument entirely analogous to Equation (57), we also have

$$\|D[u]\|_{\mathrm{L}^2(\Omega,\mu)} \geqslant \alpha \|u\|_{\mathcal{H}_D}, \qquad \forall u \in \mathcal{H}_{D,\mathcal{H}_0}^\alpha. \tag{79}$$

In particular, the functional

$$\|\cdot\|_D : \begin{cases} \mathcal{H}_{D,\mathcal{H}_0}^\alpha & \to & \mathbb{R} \\ u & \mapsto & \|D[u]\|_{\mathrm{L}^2(\Omega,\mu)} \end{cases} \tag{80}$$

defines a norm equivalent to $\|\cdot\|_{\mathcal{H}_D}$ on $\mathcal{H}_{D,\mathcal{H}_0}^\alpha$. By Lemma 1, the pair $\left(\mathcal{H}_{D,\mathcal{H}_0}^\alpha, \|\cdot\|_D\right)$ is itself an RKHS with a reproducing kernel $k_D$.

**Isometry property.** The crucial observation is that $D$ is an isometry with respect to this norm. Indeed, for all $u, v \in \mathcal{H}_{D,\mathcal{H}_0}^\alpha$,

$$\langle u \,,\, v \rangle_D = \langle D[u] \,,\, D[v] \rangle_{\mathrm{L}^2(\Omega,\mu)}. \tag{81}$$

This allows us to characterize the associated Green's function.

**Theorem 1.** *The generalized Green's function of the operator $D$ in the regularized space $\mathcal{H}_{D,\mathcal{H}_0}^\alpha$ is given, for all $x, y \in \Omega$, by*

$$g_D(x,y) := D[k_D(x,\cdot)](y), \tag{12}$$

*Proof.* For all $f \in \mathrm{L}^2(\Omega, \mu)$ and $x \in \Omega$,

$$
\begin{aligned}
\int_\Omega g_D(x,s) f(s) \mu(\mathrm{d}s) &= \langle g_D(x,\cdot) \,,\, f \rangle_{\mathrm{L}^2(\Omega,\mu)} \\
&= \langle D[k_D(x,\cdot)] \,,\, f \rangle_{\mathrm{L}^2(\Omega,\mu)} \\
&= \langle k_D(x,\cdot) \,,\, D^* f \rangle_D \\
&= \big(D^* f\big)(x).
\end{aligned} \tag{82}
$$

Since $D$ is an isometry, Lemma 2 gives $DD^* = \Pi_{D[\mathcal{H}_{D,\mathcal{H}_0}^\alpha]}$. Therefore,

$$D\Big[x \mapsto \int_\Omega g_D(x,s) f(s) \mu(\mathrm{d}s)\Big] = D\big[D^* f\big] = \Pi_{D[\mathcal{H}_{D,\mathcal{H}_0}^\alpha]} f, \tag{83}$$

which precisely shows that $g_D$ is a generalized Green's function. $\qquad\square$

The key insight of Theorem 1 is that, in the PINNs setting—and most notably in our algorithm—we implicitly construct the reproducing kernel $k_D$ associated with the norm $\|\cdot\|_D$ on the regularized tangent space $T_\theta^\alpha \mathcal{M}$ of the neural network manifold $\mathcal{M}$, at cutoff level $\alpha$. This kernel is precisely the PINNs NNTK introduced by Schwencke & Furtlehner (2025).

A crucial consequence is that the regularization of the Gram matrix is not merely a "numerical trick" to guarantee stability: it is the very mechanism that ensures the Green's function is well defined.

**Conceptual interpretation.** This perspective also offers a profound interpretation of the procedure: rather than attempting to invert the operator $D$ directly, we build a kernel $k_D$ whose associated metric makes $D$ an isometry, and thus ensures that $D^*$ acts as the generalized left-inverse of $D$. The magic of the kernel lies in the following facts:

(i) We never need to compute $D^*$ explicitly, since it is implicitly encoded in the relation

$$\langle D[k_D(x,\cdot)] \,,\, f \rangle_{\mathrm{L}^2(\Omega,\mu)} = \langle k_D(x,\cdot) \,,\, D^* f \rangle_D. \tag{84}$$

(ii) The same formula allows us to directly evaluate the generalized solution $D^* f$: indeed, for all $x \in \Omega$, the reproducing property gives

$$D^* f(x) = \langle k_D(x,\cdot) \,,\, D^* f \rangle_D. \tag{85}$$

**Comparison with Ridge regression.** An analogous analysis holds for Ridge regression. However, instead of inverting $D$ "via isometry," we invert the augmented operator $\big(D, \alpha \mathrm{I}_{\mathcal{H}_D}\big)$.

## G.5 Convergence of AMStraMGRAM in the NTK-regime

Let us consider we are in the so-called NTK-regime (Jacot et al., 2018) under infinite-width assumption. In this case the NTK converges to a fixed kernel $k$. Under mild assumptions(**?**), we may assume that $k$ is $2m$-times differentiable. Let us denote $\mathcal{H}(k)$ the Reproducing Kernel Hilbert Space associated to $k$ with its associated norm $\|\cdot\|_k$.

We have the following result.

**Proposition 3.** *Let $(X, \|\cdot\|_X)$ be a finite–dimensional normed space and let $k : X \times X \to \mathbb{R}$ be a positive semidefinite kernel of class $\mathcal{C}^{2m}$ with respect to the product topology on $X \times X$.*

*Then every $f \in \mathcal{H}(k)$ is of class $\mathcal{C}^m$ on $X$. Furthermore, for every multi-index $|\alpha| \leqslant m$ and $x \in X$, there exist $k_x^\alpha \in \mathcal{H}(k)$ such that for all $f \in \mathcal{H}(k)$*

$$\partial^\alpha f(x) = \langle k_x^\alpha \,, f \rangle_{\mathcal{H}(k)} \,. \tag{86}$$

*Proof.* We begin by fixing some notation. For multi-indices $\alpha, \beta$, we denote by $\partial^{(\alpha,\beta)} k$ the mixed partial derivatives of $k$ with respect to its two kernel arguments. Concretely, for a simple index $i$, we set

$$\partial^{(i,0)} k(x,y) := \lim_{\varepsilon \to 0} \frac{k(x + \varepsilon\, e_i, y) - k(x,y)}{\varepsilon}, \tag{87}$$

and for a simple index $j$,

$$\partial^{(i,j)} k(x,y) := \lim_{\varepsilon \to 0} \frac{\partial^{(i,0)} k(x, y + \varepsilon\, e_j) - \partial^{(i,0)} k(x,y)}{\varepsilon}. \tag{88}$$

Note that $\partial^{(\alpha,\beta)} k$ is a mixed partial derivative of total order $|\alpha| + |\beta|$. This hints at the requirement that $k$ be of class $\mathcal{C}^{2m}$ in order to control derivatives of functions in $\mathcal{H}(k)$ up to order $m$, as we shall see.

By the Schwarz–Clairaut–Young theorem on the symmetry of mixed derivatives, these partial derivatives may be computed in any order, even when alternating between the two kernel variables.

Suppose now that such an element $k_x^\alpha$, as described in Equation (86), exists for $x \in X$ and a multi–index $\alpha$, and moreover lies in $\mathcal{H}(k)$. Then for each $y \in X$, the reproducing property gives

$$\langle k_x^\alpha \,, k_y \rangle = k_x^\alpha(y) \stackrel{equation\ 86}{=} \partial^\alpha k_y(x) = \partial^{(\alpha,0)} k(x,y). \tag{89}$$

This entirely characterizes $k_x^\alpha$ as a function in $\mathcal{F}(X \to \mathbb{R})$. Hence, three things remain to be checked:

    (i) that for all $x \in X$, $\partial^{(\alpha,0)} k(x,\cdot) \in \mathcal{H}(k)$,

    (ii) that Equation (86) indeed holds for all $f \in \mathcal{H}(k)$,

    (iii) that these partial derivatives are continuous.

We will establish these properties by induction on the order $m = |\alpha|$ of the partial derivative.

**m = 1, k of class $\mathcal{C}^2$.** Fix $1 \leqslant i \leqslant \dim(X)$ and $y \in X$.

*(i) $\partial^{(i,0)} k(x, \cdot) \in \mathcal{H}(k)$.* Since $X \times X$ is equipped with the product topology, for any ball $B$ centered at $y$ and contained in $X$, the set $B \times X$ is a neighborhood of $(y, x)$ for every $x \in X$. Then for any $n > N$, with $N$ large enough so that $y + 2^{-n} e_i \in B$, we have $k_{y+2^{-n}e_i} \in \mathcal{H}$, and we may define

$$c_n^i := \frac{k_{y+2^{-n}e_i} - k_y}{2^{-n}} \ \in \ \mathcal{H}. \tag{90}$$

Furthermore, for any $x \in X$,

$$\frac{k_{y+2^{-n}e_i}(x) - k_y(x)}{2^{-n}} = \frac{k\big((y,x) + (2^{-n}e_i, 0)\big) - k\big((y,x)\big)}{2^{-n}}, \tag{91}$$

and this converges to $\partial^{(i,0)} k(x,y)$ as $n \to \infty$. In other words, the sequence $(c_n^i)_{n \geqslant N}$ converges pointwise to the desired function.

It remains to show that the convergence also holds in $\|\cdot\|_{\mathcal{H}(k)}$. Since convergence in $\|\cdot\|_{\mathcal{H}(k)}$ implies pointwise convergence, uniqueness of the limit will then ensure that $\partial^{(i,0)} k(x,\cdot) \in \mathcal{H}(k)$.

Since $\mathcal{H}(k)$ is complete, it is enough to show that $(c_n^i)$ is Cauchy. To lighten the notation, let us write $h_n := 2^{-n}$ and, with a slight abuse, use $k(x + h_n, \cdot)$ instead of $k(x + h_n e_i, \cdot)$. For $p, q \geqslant N$, we compute:

$$\left\langle c_p^i,\, c_q^i \right\rangle_{\mathcal{H}(k)} = \left\langle \tfrac{k_{x+h_p} - k_x}{h_p},\, \tfrac{k_{x+h_q} - k_x}{h_q} \right\rangle_{\mathcal{H}(k)} \tag{92}$$

$$= \frac{1}{h_p h_q} \Big[ k(x + h_p, x + h_q) + k(x, x) - k(x + h_p, x) - k(x + h_q, x) \Big]$$

$$= \frac{1}{h_p} \left[ \frac{k(x + h_p, x + h_q) - k(x + h_p, x)}{h_q} - \frac{k(x + h_q, x) - k(x, x)}{h_q} \right].$$

Taking the limit as $q \to \infty$, this gives

$$\lim_{q \to \infty} \left\langle c_p^i,\, c_q^i \right\rangle_{\mathcal{H}(k)} = \frac{1}{h_p} \Big[ \partial^{(i,0)} k(x + h_p, x) - \partial^{(i,0)} k(x, x) \Big], \tag{93}$$

and therefore

$$\lim_{p \to \infty} \lim_{q \to \infty} \left\langle c_p^i,\, c_q^i \right\rangle_{\mathcal{H}(k)} = \partial^{((i,i),0)} k(x, x). \tag{94}$$

Note that if $p = q$, then since $k$ is of class $\mathcal{C}^2$ on $X \times X$ (and in particular at $(x, x)$), we also have

$$\lim_{p \to \infty} \left\langle c_p^i,\, c_p^i \right\rangle_{\mathcal{H}(k)} = \partial^{((i,i),0)} k(x, x). \tag{95}$$

Finally, observe that

$$\left\langle c_n^i - c_m^i,\, c_n^i - c_m^i \right\rangle_{\mathcal{H}(k)} = \left\langle c_n^i,\, c_n^i \right\rangle_{\mathcal{H}(k)} + \left\langle c_m^i,\, c_m^i \right\rangle_{\mathcal{H}(k)} - 2 \left\langle c_n^i,\, c_m^i \right\rangle_{\mathcal{H}(k)}. \tag{96}$$

Hence,

$$\lim_{m \to \infty} \lim_{n \to \infty} \left\langle c_n^i - c_m^i,\, c_n^i - c_m^i \right\rangle_{\mathcal{H}(k)} = 0, \tag{97}$$

which shows that $(c_n^i)$ is indeed Cauchy in $\mathcal{H}(k)$.

*Remark* 5. Using the symmetry of $k$ in Equation (93), we also obtain

$$\lim_{p \to \infty} \lim_{q \to \infty} \left\langle c_p^i,\, c_q^i \right\rangle_{\mathcal{H}(k)} = \partial^{(i,i)} k(x, x), \tag{98}$$

which shows that for all $x \in X$,

$$\partial^{(i,i)} k(x, x) = \partial^{((i,i),0)} k(x, x). \tag{99}$$

In particular, this identity requires only that $k$ be of class $\mathcal{C}^1$ in each of its variables separately. However, this is not sufficient to ensure convergence of the diagonal terms

$$\lim_{p \to \infty} \left\langle c_p^i,\, c_p^i \right\rangle_{\mathcal{H}(k)} = \partial^{((i,i),0)} k(x, x). \tag{100}$$

This clarifies why the stronger assumption $k \in \mathcal{C}^{2m}$ is imposed in order to recover $\mathcal{C}^m$ regularity for functions in $\mathcal{H}(k)$.

$(ii)$ **Equation (86) holds**. Let $f \in \mathcal{H}(k)$ and $x \in X$. Then for any $n \geqslant N$,

$$\frac{f(x) - f(x + 2^{-n} e_i)}{2^{-n}} = \left\langle c_n^i,\, f \right\rangle_{\mathcal{H}(k)}. \tag{101}$$

Since $c_n^i$ converges to $\partial^{(i,0)} k_x$ in $\mathcal{H}(k)$, the right-hand side converges to $\left\langle \partial^{(i,0)} k_x,\, f \right\rangle_{\mathcal{H}(k)}$ in $\mathbb{R}$. This proves both that $\partial^i f(x)$ exists and that it is reproduced by $\partial^{(i,0)} k_x$.

$(iii)$ $\boldsymbol{\partial^i f(\cdot)}$ **is continuous**. This is essentially an adaptation of the proof of the **??**, now applied to $\partial^{(i,0)} k_{(\cdot)}$. Namely, given $x \in X$ and a sequence $(x_n) \in X^{\mathbb{N}}$ converging to $x$,

$$\left| \partial^i f(x_n) - \partial^i f(x) \right| \leqslant \left\| \partial^{(i,0)} k_{x_n} - \partial^{(i,0)} k_x \right\|_{\mathcal{H}(k)} \| f \|_{\mathcal{H}(k)}. \tag{102}$$

Moreover, applying the reproducing property of Equation (86) to $\partial^{(\alpha,0)}k_x$ itself,

$$\left\| \partial^{(i,0)}k_{x_n} - \partial^{(i,0)}k_x \right\|^2_{\mathcal{H}(k)} = \partial^{(i,i)}k(x,x) + \partial^{(i,i)}k(x_n,x_n) - 2\,\partial^{(i,i)}k(x,x_n), \qquad (103)$$

which converges to 0 as $n \to \infty$, since $k$ is of class $\mathcal{C}^2$.

**Induction step: k of class $\mathcal{C}^{\mathbf{2(m+1)}}$.** Suppose that for every multi-index $\alpha$ with $|\alpha| \leq m$, we have $\partial^{(\alpha,0)}k_x \in \mathcal{H}(k)$ and that it satisfies Equation (86), thereby yielding continuous partial derivatives.

Fix now a multi-index $\alpha$ such that $|\alpha| = m$, let $1 \leq i \leq \dim(X)$, and $y \in X$. We shall denote by $[\alpha,i]$ the concatenation of the multi-index $\alpha$ and the index $i$.

As in the case $m = 1$, introduce the sequence

$$c_n^{[\alpha,i]} := \frac{\partial^{(\alpha,0)}k_{y+2^{-n}e_i} - \partial^{(\alpha,0)}k_y}{2^{-n}}. \qquad (104)$$

Every term of this sequence lies in $\mathcal{H}(k)$, since by the induction hypothesis both $\partial^{(\alpha,0)}k_{y+2^{-n}e_i}$ and $\partial^{(\alpha,0)}k_y$ belong to $\mathcal{H}(k)$.

By the same arguments as in the base case, we see that this sequence converges pointwise to $\partial^{([\alpha,i],0)}k_y$ and is Cauchy in $\mathcal{H}(k)$. Hence it converges in $\|\cdot\|_{\mathcal{H}(k)}$ to $\partial^{([\alpha,i],0)}k_y$, showing that

$$\partial^{([\alpha,i],0)}k_y \in \mathcal{H}(k). \qquad (105)$$

This proves point $(i)$.

Points $(ii)$ and $(iii)$ follow by entirely analogous arguments, and we shall not repeat the details here. $\square$

**Corollary 1.** *Assume $\Omega$ is of finite measure for $\mu$. Let $D$ be an operator of order at most $m$. Then $D : \mathcal{H}(k) \to L^2(\Omega)$ is continuous.*

*Proof.* This is an immediate consequence of previous proposition. $\square$

Note in particular, that there is exist thus $C_D > 0$ such that for all $u \in \mathcal{H}(k)$

$$\|D[u]\|_{\mathrm{L}^2(\Omega)} \leq C_D \|u\|_k \qquad (106)$$

We can thus define the adjoint of $D : \mathcal{H}(k) \to \mathrm{L}^2(\Omega)$ and thus the gram operator $D^*D$. Equivalently this is given the Gram kernel:

$$\mathbb{G}(x,y) := \langle D[k(x,\cdot)],\, D[k(y,\cdot)] \rangle_{\mathrm{L}^2(\Omega)}, \qquad (107)$$

By the same mechanism with spectral theorem depicted in Section G.3, we may regularize this operator by cutoff at level $\alpha$, yielding an approximation space $\mathcal{H}(k,\alpha) \subset \mathcal{H}(k)$ such that for all $u \in \mathcal{H}(k,\alpha)$

$$\|D[u]\|^2_{\mathrm{L}^2(\Omega)} \geq \alpha^2 \|u\|^2_k. \qquad (108)$$

Overall this shows that $u \mapsto \|D[u]\|_{\mathrm{L}^2(\Omega)}$ is a norm equivalent to $\|\cdot\|_k$ in $\mathcal{H}(k,\alpha)$, we can then apply then results of Section G.4 to show that $D$ has a Green's function on $\mathcal{H}(k,\alpha)$.

In particular, we can then apply standard kernel regression results with Green's functions (**?**) to show convergence of AMStraMGRAM in the NTK-regime at this level of regularization $\alpha$.

The only missing step is then to show that AMStraMGRAM yields a stable regularization level. Let $(r_{\max,t})_{t\in\mathbb{N}}$ be the the maximum cutoff rank sequence and $\epsilon$ the precision level. We the have the following lemma that concludes the convergence of AMStraMGRAM.

**Lemma 3.** *The sequence $\left( \max\left(\Delta_{r_{\max,t}}, \epsilon\right) \right)_{t\in\mathbb{N}}$ is convergent.*

*Proof.* By Line 11 in Algorithm 2, we know that $(\Delta_{r_{\max,t}})$ is non-increasing. Furthermore we obviously have for all $t \in \mathbb{N}$, $\max\left(\Delta_{r_{\max,t}}, \epsilon\right) \geq \epsilon$. Therefore there exist $\alpha_\infty > \epsilon$ such that $\lim_{t\to\infty} \Delta_{r_{\max,t}} = \alpha_\infty$. $\square$

# H PROOFS

## H.1 STATEMENT AND PROOF OF PROPOSITION 4

We start by recalling the following statements from Schwencke & Furtlehner (2025).

**Definition** (Schwencke & Furtlehner, 2025, Definition 4). A linear operator $A : \mathcal{H} \to \mathcal{H}$ is an integral operator given that there is $k : \Omega \times \Omega \to \mathbb{K}$, $\mathbb{K} \in \{\mathbb{R}, \mathbb{C}\}$, such that: for all $f \in \mathcal{H}$, for all $x \in \Omega$

$$A(f)(x) = \langle k(x, \cdot), f \rangle_{\mathcal{H}}. \tag{109}$$

**Lemma** (Schwencke & Furtlehner, 2025, Lemma 1). *Let us be* $\mathcal{H}_0 := \mathrm{Span}(u_p : 1 \leqslant p \leqslant P) \subset \mathcal{H}$ *and consider the Gram matrix* $G_{pq} := \langle u_p, u_q \rangle_{\mathcal{H}}$ *of* $(u_p)$ *and its eigen-decomposition* $G = U\Delta^2 U^t$. *Then:*

$$L_p := \sum_{1 \leqslant q \leqslant P} u_q U_{q,p} \Delta_p^{\dagger}, \tag{110}$$

*is an orthonormal basis of* $\mathcal{H}_0$. *In particular,* $\Pi_{\mathcal{H}_0}$ *is an integral operator whose kernel is:*

$$k(x, y) = \sum_{1 \leqslant p,q \leqslant P} u_p(x) G_{p,q}^{\dagger} u_q(y). \tag{111}$$

*Furthermore* $L_p$ *are the left-singular vector of the so-called* **synthesis** *operator*[3]:

$$\mathcal{T} : \begin{cases} \mathbb{R}^P & \to & \mathcal{H}_0 \\ \alpha & \mapsto & \sum_{1 \leqslant p \leqslant P} \alpha_p u_p \end{cases}. \tag{112}$$

**Proposition 4.** *Given the scalar loss*

$$\ell(\boldsymbol{\theta}) := \mathcal{L}(u_{\boldsymbol{\theta}}) \overset{(6)}{=} \tfrac{1}{2} \|u_{\boldsymbol{\theta}} - f\|_{L^2(\Omega,\mu)}^2, \tag{113}$$

*the Natural Gradient update of Equation* (8)

$$u_{\boldsymbol{\theta}_{t+1}} \leftarrow u_{\boldsymbol{\theta}_t} - \eta \, \Pi_{T_{\boldsymbol{\theta}_t}\mathcal{M}}\big(\nabla \mathcal{L}_{u_{\boldsymbol{\theta}_t}}\big) ; \qquad \boldsymbol{\theta}_{t+1} \leftarrow \boldsymbol{\theta}_t - \eta \, \mathrm{d}u_{\boldsymbol{\theta}_t}^{\dagger}\big(\Pi_{T_{\boldsymbol{\theta}_t}\mathcal{M}}\big(\nabla \mathcal{L}_{u_{\boldsymbol{\theta}_t}}\big)\big) \tag{8}$$

*can be equivalently written as*

$$\boldsymbol{\theta}_{t+1} \leftarrow \boldsymbol{\theta}_t - \eta \, G_{\boldsymbol{\theta}_t}^{\dagger} \nabla \ell(\boldsymbol{\theta}_t) ; \qquad G_{\boldsymbol{\theta}_t \, p q} := \langle \partial_p u_{\boldsymbol{\theta}_t}, \partial_q u_{\boldsymbol{\theta}_t} \rangle_{L^2(\Omega,\mu)}. \tag{9}$$

*Proof.* Since the tangent space $T_{\boldsymbol{\theta}}\mathcal{M}$ of Equation (7):

$$T_{\boldsymbol{\theta}}\mathcal{M} := \mathrm{Im}(\mathrm{d}u_{\boldsymbol{\theta}}) = \mathrm{Span}\left(\partial_p u_{\boldsymbol{\theta}} : 1 \leqslant p \leqslant P\right) \subset \mathcal{H}, \tag{7}$$

is finite-dimensional, we may invoke Schwencke & Furtlehner (2025, Lemma 1). This result shows that the **Natural Neural Tangent Kernel (NNTK)**, given by

$$NNTK_{\boldsymbol{\theta}}(x, y) := \sum_{1 \leqslant p,q \leqslant P} \big(\partial_p u_{\boldsymbol{\theta}}(x)\big) G_{\boldsymbol{\theta} \, pq}^{\dagger} \big(\partial_q u_{\boldsymbol{\theta}}(y)\big)^t, \quad G_{\boldsymbol{\theta} \, p,q} := \langle \partial_p u_{\boldsymbol{\theta}}, \partial_q u_{\boldsymbol{\theta}} \rangle_{\mathcal{H}}, \tag{114}$$

is the kernel of the orthogonal projection $\Pi_{T_{\boldsymbol{\theta}}\mathcal{M}}^{\perp}$ onto $T_{\boldsymbol{\theta}}\mathcal{M}$. Therefore, by Equation (109), for all $x \in \Omega$,

$$\begin{aligned} \Pi_{T_{\boldsymbol{\theta}}\mathcal{M}}^{\perp}\big(\nabla \mathcal{L}_{|u_{\boldsymbol{\theta}}}\big)(x) &= \big\langle NNTK_{\boldsymbol{\theta}}(x, \cdot), \nabla \mathcal{L}_{|u_{\boldsymbol{\theta}}} \big\rangle_{\mathcal{H}} \\ &\overset{(114)}{=} \sum_{1 \leqslant p,q \leqslant P} \partial_p u_{\boldsymbol{\theta}}(x) G_{\boldsymbol{\theta} \, pq}^{\dagger} \big\langle \partial_q u_{\boldsymbol{\theta}}, \nabla \mathcal{L}_{|u_{\boldsymbol{\theta}}} \big\rangle_{\mathcal{H}}. \end{aligned} \tag{115}$$

Next, note that

$$\big\langle \partial_q u_{\boldsymbol{\theta}}, \nabla \mathcal{L}_{|u_{\boldsymbol{\theta}}} \big\rangle_{\mathcal{H}} = \mathrm{d}\mathcal{L}_{|u_{\boldsymbol{\theta}}}\big(\partial_q u_{\boldsymbol{\theta}}\big) \overset{\text{chain rule}}{=} \partial_q \mathcal{L}(u_{\boldsymbol{\theta}}) \overset{(113)}{=} \partial_q \ell(\boldsymbol{\theta}). \tag{116}$$

---

[3]Name and notation are taken from Adcock & Huybrechs (2019).

Therefore, by linearity of $\mathrm{d}u_{\boldsymbol{\theta}}^{\dagger}$,

$$\mathrm{d}u_{\boldsymbol{\theta}}^{\dagger}\left(\Pi_{T_{\boldsymbol{\theta}}\mathcal{M}}^{\perp}\left(\nabla\mathcal{L}_{|u_{\boldsymbol{\theta}}}\right)\right) \overset{(115),(116)}{=} \sum_{1\leqslant p,q\leqslant P} \mathrm{d}u_{\boldsymbol{\theta}}^{\dagger}(\partial_p u_{\boldsymbol{\theta}})\, G_{\boldsymbol{\theta}\,pq}^{\dagger}\, \partial_q \ell(\boldsymbol{\theta}). \tag{117}$$

Finally, observe that $\partial_p u_{\boldsymbol{\theta}} = \mathrm{d}u_{\boldsymbol{\theta}}(\boldsymbol{e}^{(p)})$, where $\boldsymbol{e}^{(p)}$ is the $p$-th canonical basis vector of $\mathbb{R}^P$. If $\mathrm{d}u_{\boldsymbol{\theta}}$ were invertible, we would directly obtain

$$\mathrm{d}u_{\boldsymbol{\theta}}^{\dagger}(\partial_p u_{\boldsymbol{\theta}}) = \boldsymbol{e}^{(p)}, \tag{118}$$

which would complete the argument. However, this invertibility does not hold in general.

To address this, recall that $\mathrm{d}u_{\boldsymbol{\theta}}$ can be identified with the synthesis operator $\mathcal{T}$ introduced in Equation (112) of Schwencke & Furtlehner (2025, Lemma 1). From the final part of that lemma, we know that $\operatorname{Im}\mathrm{d}u_{\boldsymbol{\theta}}^{\dagger} = \operatorname{Im}G_{\boldsymbol{\theta}}^{\dagger}$. Consequently,

$$G_{\boldsymbol{\theta}}^{\dagger}\boldsymbol{e}^{(p)} = G_{\boldsymbol{\theta}}^{\dagger}\mathrm{d}u_{\boldsymbol{\theta}}^{\dagger}(\partial_p u_{\boldsymbol{\theta}}). \tag{119}$$

Putting all pieces together yields the desired update rule, thereby completing the proof. $\square$

## H.2 Ridge-regression implementation ANaGRAM

In the following, we show that a Ridge-regression can be implemented in ANaGRAM's update rule given by Equation (10).

**Proposition 5.** *A Ridge-regression can be implemented in the SVD-based update Equation* (10) *by replacing the pseudo-inverse $\widehat{\Delta}^{\dagger}$ with*

$$\left(\frac{\widehat{\Delta}_{t,i}}{\widehat{\Delta}_{t,i}^2 + S\alpha}\right)_{1\leqslant i\leqslant r_{\mathrm{svd}}}. \tag{120}$$

*Proof.* As shown in (Schwencke & Furtlehner, 2025, Section E), the ANaGRAM's update of Equation (10):

$$\boldsymbol{\theta}_{t+1} \leftarrow \boldsymbol{\theta}_t - \eta\,\widehat{\phi}^{\dagger}\widehat{\nabla\mathcal{L}}_{\boldsymbol{\theta}_t}; \qquad \widehat{\phi}_{i,p} := \partial_p u_{\boldsymbol{\theta}}(x_i); \qquad \left(\widehat{\nabla\mathcal{L}}_{\boldsymbol{\theta}}\right)_i := u_{\boldsymbol{\theta}}(x_i) - f(x_i), \tag{10}$$

is equivalent to the update with the empirical matrix $\widehat{\mathcal{G}}_{\boldsymbol{\theta}}$:

$$\boldsymbol{\theta}_{t+1} \leftarrow \boldsymbol{\theta}_t - \eta\,\widehat{\mathcal{G}}_{\boldsymbol{\theta}_t}^{\dagger}\nabla\ell(\boldsymbol{\theta}_t)\,; \qquad\qquad \widehat{\mathcal{G}}_{\boldsymbol{\theta}_t} := \frac{1}{S}\widehat{\phi}_{\boldsymbol{\theta}_t}^t\,\widehat{\phi}_{\boldsymbol{\theta}_t}, \tag{121}$$

where $\ell$ is defined in Equation (5):

$$\ell(\boldsymbol{\theta}) := \frac{1}{2S}\sum_{i=1}^{S}\left(u_{\boldsymbol{\theta}}(x_i) - f(x_i)\right)^2. \tag{5}$$

Thus, we get immediately

$$\nabla\ell(\boldsymbol{\theta}_t) = \frac{1}{S}\widehat{\phi}^t\widehat{\nabla\mathcal{L}}_{\boldsymbol{\theta}} = \frac{1}{S}\widehat{U}\widehat{\Delta}\widehat{V}_{\boldsymbol{\theta}}^t\widehat{\nabla\mathcal{L}}_{\boldsymbol{\theta}}, \tag{122}$$

where we used the SVD decomposition of $\widehat{\phi}$:

$$. \tag{123}$$

Using Equation (123) again, we have

$$\widehat{\mathcal{G}}_{\boldsymbol{\theta}} = \frac{1}{S}\widehat{U}\widehat{\Delta}_{\boldsymbol{\theta}}^2\widehat{U}_{\boldsymbol{\theta}}^t, \tag{124}$$

thus for all $\alpha > 0$

$$\widehat{\mathcal{G}}_{\boldsymbol{\theta}} + \alpha I_d = \frac{1}{S}\widehat{U}\widehat{\Delta}_{\boldsymbol{\theta}}^2\widehat{U}_{\boldsymbol{\theta}}^t + \alpha\widehat{U}\widehat{U}^t = \widehat{U}\left(\operatorname{diag}\left(\frac{\widehat{\Delta}_{\boldsymbol{\theta}_i}^2}{S} + \alpha\right)_{1\leqslant i\leqslant r_{\mathrm{svd}}}\right)\widehat{U}_{\boldsymbol{\theta}}^t, \tag{125}$$

which implies

$$\left(\widehat{\mathcal{G}}_{\boldsymbol{\theta}} + \alpha I_d\right)^{-1} = \widehat{U}\left(\mathrm{diag}\left(\frac{S}{\widehat{\Delta}_{\boldsymbol{\theta}_i}^2 + S\alpha}\right)_{1 \leqslant i \leqslant \mathrm{r_{svd}}}\right)\widehat{U}_{\boldsymbol{\theta}}^t. \tag{126}$$

This finally yields

$$\left(\widehat{\mathcal{G}}_{\boldsymbol{\theta}} + \alpha I_d\right)^{-1}\nabla\ell(\boldsymbol{\theta}_t) \overset{(122)}{=} \widehat{U}\left(\mathrm{diag}\left(\frac{S}{\widehat{\Delta}_{\boldsymbol{\theta}_i}^2 + S\alpha}\right)_{1 \leqslant i \leqslant \mathrm{r_{svd}}}\right)\widehat{U}_{\boldsymbol{\theta}}^t\frac{1}{S}\widehat{U}\widehat{\Delta}\widehat{V}_{\boldsymbol{\theta}}^t\widehat{\nabla\mathcal{L}}_{\boldsymbol{\theta}} \tag{127}$$

$$= \widehat{U}\left(\mathrm{diag}\left(\frac{\widehat{\Delta}_{t,i}}{\widehat{\Delta}_{\boldsymbol{\theta}_i}^2 + S\alpha}\right)_{1 \leqslant i \leqslant \mathrm{r_{svd}}}\right)\widehat{V}_{\boldsymbol{\theta}}^t\widehat{\nabla\mathcal{L}}_{\boldsymbol{\theta}}, \tag{128}$$

which conludes the proof. □

## H.3 PROOF OF PROPOSITION 1

To prove Proposition 1, we need the following lemma:

**Lemma 4.** *For* $1 \leqslant M \leqslant N \leqslant r_{svd}$:

$$\left(RCE_M^S\right)^2 - \left(RCE_N^S\right)^2 = \frac{1}{S}\left\|\Pi_N^M\widehat{V}^T\widehat{\nabla\mathcal{L}}\right\|_{\mathbb{R}^S}^2. \tag{129}$$

*Proof.* Let us first recall the definition of the $RCE_N^S$ in Equation (13), namely

$$\mathrm{RCE}_N^S := \frac{1}{\sqrt{S}}\left\|\widehat{V}\Pi_N^0\widehat{V}^T\widehat{\nabla\mathcal{L}} - \widehat{\nabla\mathcal{L}}\right\|_{\mathbb{R}^S}. \tag{13}$$

Fixing $1 \leqslant N \leqslant M \leqslant \mathrm{r_{svd}}$ and applying the same reasoning as in Equation (139) to $RCE_M^S$ and $RCE_N^S$ (see the proof of Proposition 1 in Section H.3), we get

$$S\left(\mathrm{RCE}_M^S\right)^2 = \widehat{\nabla\mathcal{L}}_{\boldsymbol{\theta}}^t\widehat{\nabla\mathcal{L}}_{\boldsymbol{\theta}} - \sum_{p=1}^{M}\left(\widehat{V}_{\boldsymbol{\theta}_p}^t\widehat{\nabla\mathcal{L}}_{\boldsymbol{\theta}}\right)^2; \quad S\left(\mathrm{RCE}_N^S\right)^2 = \widehat{\nabla\mathcal{L}}_{\boldsymbol{\theta}}^t\widehat{\nabla\mathcal{L}}_{\boldsymbol{\theta}} - \sum_{p=1}^{N}\left(\widehat{V}_{\boldsymbol{\theta}_p}^t\widehat{\nabla\mathcal{L}}_{\boldsymbol{\theta}}\right)^2, \tag{130}$$

and therefore

$$S\left(\left(\mathrm{RCE}_N^S\right)^2 - \left(\mathrm{RCE}_M^S\right)^2\right) = \sum_{p=1}^{N}\left(\widehat{V}_{\boldsymbol{\theta}_p}^t\widehat{\nabla\mathcal{L}}_{\boldsymbol{\theta}}\right)^2 - \sum_{p=1}^{M}\left(\widehat{V}_{\boldsymbol{\theta}_p}^t\widehat{\nabla\mathcal{L}}_{\boldsymbol{\theta}}\right)^2$$

$$\overset{M \leqslant N}{=} \sum_{p=M+1}^{N}\left(\widehat{V}_{\boldsymbol{\theta}_p}^t\widehat{\nabla\mathcal{L}}_{\boldsymbol{\theta}}\right)^2 = \sum_{p=M+1}^{N}\left(\boldsymbol{e}^{(p)^t}\widehat{V}^t\widehat{\nabla\mathcal{L}}_{\boldsymbol{\theta}}\right)^2$$

$$= \sum_{p=M+1}^{N}\left(\widehat{\nabla\mathcal{L}}_{\boldsymbol{\theta}}^t\widehat{V}\boldsymbol{e}^{(p)}\right)\left(\boldsymbol{e}^{(p)^t}\widehat{V}^t\widehat{\nabla\mathcal{L}}_{\boldsymbol{\theta}}\right)$$

$$= \widehat{\nabla\mathcal{L}}_{\boldsymbol{\theta}}^t\widehat{V}\underbrace{\left(\sum_{p=M+1}^{N}\boldsymbol{e}^{(p)}\boldsymbol{e}^{(p)^t}\right)}_{=\Pi_N^M\ by\ Equation\ (14)}\widehat{V}^t\widehat{\nabla\mathcal{L}}_{\boldsymbol{\theta}} \tag{131}$$

$$= \left\langle\widehat{V}^t\widehat{\nabla\mathcal{L}}_{\boldsymbol{\theta}}, \Pi_N^M\widehat{V}^t\widehat{\nabla\mathcal{L}}_{\boldsymbol{\theta}}\right\rangle_{\mathbb{R}^S}$$

$$\overset{\substack{\Pi_N^{M^2}=\Pi_N^M \\ \Pi_N^{M^t}=\Pi_N^M}}{=} \left\langle\Pi_N^M\widehat{V}^t\widehat{\nabla\mathcal{L}}_{\boldsymbol{\theta}}, \Pi_N^M\widehat{V}^t\widehat{\nabla\mathcal{L}}_{\boldsymbol{\theta}}\right\rangle_{\mathbb{R}^S}$$

$$= \left\|\Pi_N^M\widehat{V}^t\widehat{\nabla\mathcal{L}}_{\boldsymbol{\theta}}\right\|_{\mathbb{R}^S}^2,$$

where we use in the penultimate equality, the fact that $\Pi_N^M$ is an orthogonal projection. □

*Remark* 6. The above lemma provides an interesting property that gives a further understanding of what is happening during the flattening, *i.e.* $\mathrm{RCE}_M^S - \mathrm{RCE}_N^S \approx 0$. In particular, as $\left(\mathrm{RCE}_M^S\right)^2 - \left(\mathrm{RCE}_N^S\right)^2 = \left(\mathrm{RCE}_M^S - \mathrm{RCE}_N^S\right)\left(\mathrm{RCE}_M^S + \mathrm{RCE}_N^S\right)$, therefore flattening for the components in the range $[N_{\mathrm{flat}}, \mathrm{r_{cutoff}}]$ means that $\frac{1}{S}\left\|\Pi_N^M \widehat{V}^T \widehat{\nabla\mathcal{L}}\right\|_{\mathbb{R}^S}^2 \approx 0$. In other words, the problem is "learned" for those components, as the projection of the functional gradient (which is propotional to the error) on their corresponding span is null. The proof of this lemma is provided in Section H.3.

**Proposition 1.** *$RCE_N^S$ is a non-increasing function of N, i.e. for all $1 \leqslant M, N \leqslant r_{svd}$:*

$$M \leqslant N \implies RCE_M^S \geqslant RCE_N^S. \tag{15}$$

*Furthermore, assuming that $(x_i)_{i=1}^S$ are i.i.d sampled from $\mu$, we have $\mu$-almost surely*

$$\lim_{S\to\infty} RCE_N^S = \left\|\nabla\mathcal{L}_{u_{\boldsymbol{\theta}}} - \Pi_{T_N^0\mathcal{M}}^{\perp}\nabla\mathcal{L}_{u_{\boldsymbol{\theta}}}\right\|_{L^2(\Omega,\mu)} = \left\|\Pi_{\left[T_N^0\mathcal{M}\right]^{\perp}}^{\perp}\nabla\mathcal{L}_{u_{\boldsymbol{\theta}}}\right\|_{L^2(\Omega,\mu)}, \tag{16}$$

*where $T_N^M\mathcal{M} = \mathrm{Span}(V_{t,i} : M \leqslant i \leqslant N)$, while $(V_{t,i})_{1\leqslant i\leqslant r_{svd}}$ are the right singular-vectors of the differential $du_{\boldsymbol{\theta}}$ ordered in a decreasing order according to their associated singular values.*

*Proof.* The first statement is a direct consequence of Lemma 4 proven above.

Let us now show that the second statement takes place. Since $\nabla\mathcal{L}_{u_{\boldsymbol{\theta}}} \in L^2(\Omega,\mu)$ and $\mathrm{Im}\, du_{\boldsymbol{\theta}} \subset L^2(\Omega,\mu)$, the law of large numbers yields that for all $1 \leqslant p, q \leqslant P$

$$\lim_{S\to\infty}\frac{1}{S}\sum_{i=1}^S \left[\nabla\mathcal{L}_{u_{\boldsymbol{\theta}}}(x_i)\right]^2 = \lim_{S\to\infty}\frac{1}{S}\widehat{\nabla\mathcal{L}}_{\boldsymbol{\theta}}^t\widehat{\nabla\mathcal{L}}_{\boldsymbol{\theta}} = \int_{\Omega}\left[\nabla\mathcal{L}_{u_{\boldsymbol{\theta}}}(x)\right]^2\mu(\mathrm{d}x) \quad a.s, \tag{132}$$

$$\lim_{S\to\infty}\frac{1}{S}\sum_{i=1}^S \partial_p u_{\boldsymbol{\theta}}(x_i)\nabla\mathcal{L}_{u_{\boldsymbol{\theta}}}(x_i) = \lim_{S\to\infty}\frac{1}{S}\widehat{\phi}_{\boldsymbol{\theta}_p}^t\widehat{\nabla\mathcal{L}}_{\boldsymbol{\theta}} = \int_{\Omega}\partial_p u_{\boldsymbol{\theta}}(x)\nabla\mathcal{L}_{u_{\boldsymbol{\theta}}}(x)\mu(\mathrm{d}x) \quad a.s, \tag{133}$$

$$\lim_{S\to\infty}\frac{1}{S}\sum_{i=1}^S \partial_p u_{\boldsymbol{\theta}}(x_i)\partial_q u_{\boldsymbol{\theta}}(x_i) = \lim_{S\to\infty}\frac{1}{S}\widehat{\phi}_{\boldsymbol{\theta}_p}^t\widehat{\phi}_{\boldsymbol{\theta}_q} = \int_{\Omega}\partial_p u_{\boldsymbol{\theta}}(x)\partial_q u_{\boldsymbol{\theta}}(x)\mu(\mathrm{d}x) \quad a.s. \tag{134}$$

In particular, this implies

$$\lim_{S\to\infty}\frac{1}{S}\widehat{\phi}^t\widehat{\phi} = G_{\boldsymbol{\theta}} = U_{\boldsymbol{\theta}}\Delta_{\boldsymbol{\theta}}^2 U_{\boldsymbol{\theta}}^t \quad a.s. \tag{135}$$

Since the eigenvectors (and eigenvalues) are continuous functions of the matrix coefficients (by polynomial dependence through the characteristic polynomial) and taking into account that $\frac{1}{S}\widehat{\phi}^t\widehat{\phi} = \frac{1}{S}\widehat{U}\widehat{\Delta}_{\boldsymbol{\theta}}^2\widehat{U}^t$, this yields

$$\lim_{S\to\infty}\widehat{U} = U_{\boldsymbol{\theta}} \quad a.s; \qquad\qquad \lim_{S\to\infty}\frac{1}{S}\widehat{\Delta}^2 = \Delta_{\boldsymbol{\theta}}^2 \quad a.s. \tag{136}$$

By continuity of the square root and the inverse on $\mathbb{R}_+^*$, we get that for all $1 \leqslant p \leqslant P$ such that $\Delta_{\boldsymbol{\theta}_p} > 0$

$$\lim_{S\to\infty}\sqrt{S}\widehat{\Delta}_{\boldsymbol{\theta}_p}^{-1} = \Delta_{\boldsymbol{\theta}_p}^{-1} \quad a.s, \tag{137}$$

and thus for all $1 \leqslant p \leqslant P$ such that $\Delta_{\boldsymbol{\theta}_p} > 0$, we have *almost surely*

$$\lim_{S\to\infty} \frac{1}{\sqrt{S}} \widehat{V}_{\boldsymbol{\theta}_p}^T \widehat{\nabla\mathcal{L}_{\boldsymbol{\theta}}} = \lim_{S\to\infty} \sqrt{S} \widehat{\Delta}_{\boldsymbol{\theta}_p}^{-1} \widehat{U}_{\boldsymbol{\theta}_p}^t \left( \sum_{q=1}^P \boldsymbol{e}^{(q)} \boldsymbol{e}^{(q)^t} \right) \frac{1}{S} \widehat{\phi}^t \widehat{\nabla\mathcal{L}_{\boldsymbol{\theta}}}$$

$$= \sum_{q=1}^P \left( \lim_{S\to\infty} \sqrt{S} \widehat{\Delta}_{\boldsymbol{\theta}_p}^{-1} \right) \left( \lim_{S\to\infty} \widehat{U}_{\boldsymbol{\theta}_p}^t \boldsymbol{e}^{(q)} \right) \left( \lim_{S\to\infty} \frac{1}{S} \widehat{\phi}_{\boldsymbol{\theta}_q}^t \widehat{\nabla\mathcal{L}_{\boldsymbol{\theta}}} \right)$$

$$= \sum_{q=1}^P \Delta_{\boldsymbol{\theta}_p}^{-1} U_{\boldsymbol{\theta}_p}^t \boldsymbol{e}^{(q)} \int_\Omega \partial_q u_{\boldsymbol{\theta}}(x) \nabla\mathcal{L}_{u_{\boldsymbol{\theta}}}(x) \mu(\mathrm{d}x) \tag{138}$$

$$= \int_\Omega \mathrm{d}u_{\boldsymbol{\theta}} \left( U_{\boldsymbol{\theta}_p} \Delta_{\boldsymbol{\theta}_p}^{-1} \right)(x) \nabla\mathcal{L}_{u_{\boldsymbol{\theta}}}(x) \mu(\mathrm{d}x)$$

$$= \int_\Omega V_{\boldsymbol{\theta}_p}(x) \nabla\mathcal{L}_{u_{\boldsymbol{\theta}}}(x) \mu(\mathrm{d}x),$$

where we used in the last equality, the identification of the singular vectors of $\mathrm{d}u_{\boldsymbol{\theta}}$ in (Schwencke & Furtlehner, 2025, Lemma 1 p. 24, section C.2). Returning to the definition of the $\mathrm{RCE}_N^S$ in Equation (13), namely

$$\mathrm{RCE}_N^S := \frac{1}{\sqrt{S}} \left\| \widehat{V}\Pi_N^0 \widehat{V}^T \widehat{\nabla\mathcal{L}} - \widehat{\nabla\mathcal{L}} \right\|_{\mathbb{R}^S}, \tag{13}$$

we get

$$S\left(\mathrm{RCE}_N^S\right)^2 = \left\langle \widehat{V}\Pi_N^0 \widehat{V}^t \widehat{\nabla\mathcal{L}_{\boldsymbol{\theta}}} - \widehat{\nabla\mathcal{L}_{\boldsymbol{\theta}}} \,,\, \widehat{V}\Pi_N^0 \widehat{V}^t \widehat{\nabla\mathcal{L}_{\boldsymbol{\theta}}} - \widehat{\nabla\mathcal{L}_{\boldsymbol{\theta}}} \right\rangle_{\mathbb{R}^S}$$

$$= \widehat{\nabla\mathcal{L}_{\boldsymbol{\theta}}}^t \widehat{\nabla\mathcal{L}_{\boldsymbol{\theta}}} + \widehat{\nabla\mathcal{L}_{\boldsymbol{\theta}}}^t \widehat{V} \overbrace{\Pi_N^0 \underbrace{\widehat{V}^t \widehat{V}}_{=I_d} \Pi_N^0}^{=\Pi_N^0} \widehat{V}^t \widehat{\nabla\mathcal{L}_{\boldsymbol{\theta}}} - 2\widehat{\nabla\mathcal{L}_{\boldsymbol{\theta}}}^t \widehat{V}\Pi_N^0 \widehat{V}^t \widehat{\nabla\mathcal{L}_{\boldsymbol{\theta}}}$$

$$= \widehat{\nabla\mathcal{L}_{\boldsymbol{\theta}}}^t \widehat{\nabla\mathcal{L}_{\boldsymbol{\theta}}} - \widehat{\nabla\mathcal{L}_{\boldsymbol{\theta}}}^t \widehat{V}\Pi_N^0 \widehat{V}^t \widehat{\nabla\mathcal{L}_{\boldsymbol{\theta}}} \tag{139}$$

$$= \widehat{\nabla\mathcal{L}_{\boldsymbol{\theta}}}^t \widehat{\nabla\mathcal{L}_{\boldsymbol{\theta}}} - \widehat{\nabla\mathcal{L}_{\boldsymbol{\theta}}}^t \widehat{V} \left( \sum_{p=1}^N \boldsymbol{e}^{(p)} \boldsymbol{e}^{(p)^t} \right) \widehat{V}^t \widehat{\nabla\mathcal{L}_{\boldsymbol{\theta}}}$$

$$= \widehat{\nabla\mathcal{L}_{\boldsymbol{\theta}}}^t \widehat{\nabla\mathcal{L}_{\boldsymbol{\theta}}} - \sum_{p=1}^N \left( \widehat{V}_{\boldsymbol{\theta}_p}^t \widehat{\nabla\mathcal{L}_{\boldsymbol{\theta}}} \right)^2,$$

where in the second equality, we use the fact that $\widehat{V}$ is orthogonal and $\Pi_N^0$ is a projection. Combining Equations (132) and (138), this yields

$$\lim_{S\to\infty} \left(\mathrm{RCE}_N^S\right)^2 = \int_\Omega \nabla\mathcal{L}_{u_{\boldsymbol{\theta}}}(x)^2 \mu(\mathrm{d}x) - \sum_{p=1}^N \left( \int_\Omega V_{\boldsymbol{\theta}_p}(x) \nabla\mathcal{L}_{u_{\boldsymbol{\theta}}}(x) \mu(\mathrm{d}x) \right)^2 \quad a.s. \tag{140}$$

By Fubini's theorem, we have *almost surely*

$$\sum_{p=1}^N \left( \int_\Omega V_{\boldsymbol{\theta}_p}(x) \nabla\mathcal{L}_{u_{\boldsymbol{\theta}}}(x) \mu(\mathrm{d}x) \right)^2 = \int_{\Omega^2} \nabla\mathcal{L}_{u_{\boldsymbol{\theta}}}(x) \left( \sum_{p=1}^N V_{\boldsymbol{\theta}_p}(x) V_{\boldsymbol{\theta}_p}(y) \right) \nabla\mathcal{L}_{u_{\boldsymbol{\theta}}}(y) \mu(\mathrm{d}x)\mu(\mathrm{d}y)$$

$$= \int_\Omega \nabla\mathcal{L}_{u_{\boldsymbol{\theta}}}(x) \Pi_{\mathrm{Span}(V_{\boldsymbol{\theta}_i} : 1\leqslant i\leqslant N)}^\perp \nabla\mathcal{L}_{u_{\boldsymbol{\theta}}}(x) \mu(\mathrm{d}x) \tag{141}$$

$$= \left\| \Pi_{\mathrm{Span}(V_{\boldsymbol{\theta}_i} : 1\leqslant i\leqslant N)}^\perp \nabla\mathcal{L}_{u_{\boldsymbol{\theta}}} \right\|_{\mathrm{L}^2(\Omega,\mu)}^2,$$

where in the second equality, we used (Schwencke & Furtlehner, 2025, Theorem 4 p. 23, section C.2) and the fact that $\left( \Pi_{\mathrm{Span}(V_{\boldsymbol{\theta}_i} : 1\leqslant i\leqslant N)}^\perp \right)^2 = \Pi_{\mathrm{Span}(V_{\boldsymbol{\theta}_i} : 1\leqslant i\leqslant N)}^\perp$ in the third. Therefore, from Equation (140) and Equation (141)

$$\lim_{S\to\infty} \left(\mathrm{RCE}_N^S\right)^2 = \left\| \nabla\mathcal{L}_{u_{\boldsymbol{\theta}}} \right\|_{\mathrm{L}^2(\Omega,\mu)}^2 - \left\| \Pi_{\mathrm{Span}(V_{\boldsymbol{\theta}_i} : 1\leqslant i\leqslant N)}^\perp \nabla\mathcal{L}_{u_{\boldsymbol{\theta}}} \right\|_{\mathrm{L}^2(\Omega,\mu)}^2 \quad a.s, \tag{142}$$

$$= \left\| \nabla\mathcal{L}_{u_{\boldsymbol{\theta}}} - \Pi_{\mathrm{Span}(V_{\boldsymbol{\theta}_i} : 1\leqslant i\leqslant N)}^\perp \nabla\mathcal{L}_{u_{\boldsymbol{\theta}}} \right\|_{\mathrm{L}^2(\Omega,\mu)}^2 \quad a.s, \tag{143}$$

where in the second equality, we use in the reverse order a reasoning similar to Equation (139). Finally, we obtain

$$\left\| \nabla \mathcal{L}_{u_{\boldsymbol{\theta}}} - \Pi^{\perp}_{\mathrm{Span}(V_{\boldsymbol{\theta}_i} : 1 \leqslant i \leqslant N)} \nabla \mathcal{L}_{u_{\boldsymbol{\theta}}} \right\|^2_{\mathrm{L}^2(\Omega,\mu)} = \left\| \Pi^{\perp}_{\mathrm{Span}(V_{\boldsymbol{\theta}_i} : 1 \leqslant i \leqslant N)^{\perp}} \nabla \mathcal{L}_{u_{\boldsymbol{\theta}}} \right\|^2_{\mathrm{L}^2(\Omega,\mu)}, \quad (144)$$

which comes from the canonical decomposition in Hilbert spaces, *i.e.* using that $\mathrm{Span}(V_{\boldsymbol{\theta}_i} : 1 \leqslant i \leqslant N)$ is a closed subspace and

$$\nabla \mathcal{L}_{u_{\boldsymbol{\theta}}} = \Pi^{\perp}_{\mathrm{Span}(V_{\boldsymbol{\theta}_i} : 1 \leqslant i \leqslant N)} \nabla \mathcal{L}_{u_{\boldsymbol{\theta}}} + \Pi^{\perp}_{\mathrm{Span}(V_{\boldsymbol{\theta}_i} : 1 \leqslant i \leqslant N)^{\perp}} \nabla \mathcal{L}_{u_{\boldsymbol{\theta}}}. \quad (145)$$

This completes the proof. $\qquad\square$

**Corollary 2.** *For* $1 \leqslant M \leqslant N \leqslant r_{svd}$:

$$\lim_{S \to \infty} \left( RCE_M^S \right)^2 - \left( RCE_N^S \right)^2 = \left\| \Pi^{\perp}_{T_N^M \mathcal{M}} \nabla \mathcal{L}_{u_{\boldsymbol{\theta}}} \right\|^2_{L^2(\Omega)} \quad (146)$$

*Proof.* Apply Proposition 1 to Equation (129) of Lemma 4. $\qquad\square$

# I  APPENDIX: EMPIRICAL ANALYSIS OF BATCHWISE TRUNCATED NATURAL GRADIENT

## I.1  EMPIRICAL ANALYSIS OF KERNEL ALIGNMENT

To quantify the stability of the natural gradient update under resampling, we introduce the *inverse kernel alignment* metric. This metric measures the similarity between the tangent spaces induced by the same kernel on different batches of training data. Formally, let $B_1, B_2$ be two independent training batches and $B_{\text{test}}$ a test batch. For each batch $k \in \{1, 2\}$, we compute the SVD of the feature matrix $\widehat{\phi}_k = \widehat{V}_k \widehat{\Delta}_k \widehat{U}_k^{\top}$. The truncated inverse Gram matrix at rank $r$ is given by $G_k^{\dagger(r)} = \sum_{i=1}^{r} \sigma_{k,i}^{-2} \boldsymbol{u}_{k,i} \boldsymbol{u}_{k,i}^{\top}$, where $\boldsymbol{u}_{k,i}$ is the $i$-th column of $\widehat{U}_k$. We define the alignment $\mathcal{A}(r)$ as the cosine similarity between $G_1^{\dagger(r)}$ and $G_2^{\dagger(r)}$ relative to the geometry of the test set:

$$\mathcal{A}(r) := \frac{\langle G_1^{\dagger(r)}, G_2^{\dagger(r)} \rangle_{\Sigma_{\text{test}}}}{\|G_1^{\dagger(r)}\|_{\Sigma_{\text{test}}} \|G_2^{\dagger(r)}\|_{\Sigma_{\text{test}}}}, \quad (147)$$

where $\Sigma_{\text{test}} = \widehat{\phi}_{\text{test}}^{\top} \widehat{\phi}_{\text{test}}$ and $\langle A, B \rangle_{\Sigma_{\text{test}}} = \mathrm{Tr}(A \Sigma_{\text{test}} B \Sigma_{\text{test}})$. This metric evaluates whether the optimization directions prescribed by different batches are consistent when applied to the test distribution. Computationally, this reduces to comparing the projected test features $W_k = \widehat{\phi}_{\text{test}} \widehat{U}_k$: the numerator becomes $\sum_{i,j=1}^{r} \sigma_{1,i}^{-2} \sigma_{2,j}^{-2} (\boldsymbol{w}_{1,i}^{\top} \boldsymbol{w}_{2,j})^2$, where $\boldsymbol{w}_{k,i}$ are columns of $W_k$.

Figure 20 and Figure 21 show the alignment curves for the heat equation. We observe a characteristic behavior:

- At low quantities of retained components (small $r$), the kernels are very similar. This corresponds to the largest eigenvalues, which capture the dominant modes of the loss landscape and are less sensitive to changes in the training samples.

- At large quantities of retained components (large $r$), the kernels again show high alignment, as they both describe almost the same space (approaching the full empirical tangent space).

- In the intermediate regime, the alignment drops, indicating that the subspaces spanned by the intermediate eigenvectors are more sensitive to the specific batch of data points.

This analysis confirms that truncating the spectrum (as done in AMStraMGRAM) or using the full spectrum is relatively stable, whereas intermediate truncations might be more sensitive to sampling noise.

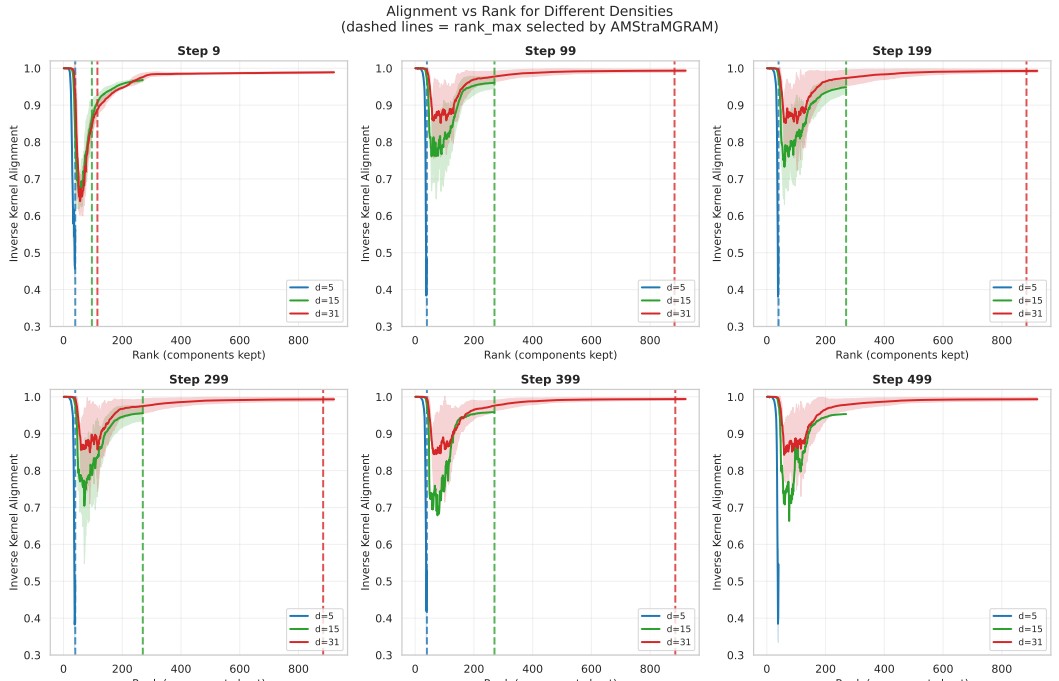

**Figure 20:** Inverse kernel alignment as a function of the rank $r$ (number of retained components) for the heat equation: Impact of Sample Density. The dashed lines indicate the rank selected by AMStraMGRAM.

## J   LAZY TRAINING FOR THE NATURAL GRADIENT (CLASSICAL VERSION)

We recall from (Chizat et al., 2019, Section 1.2) that lazy training is characterized by monitoring the following two quantities:

**Relative change of the loss**  defined by the ratio

$$\Delta(\ell)(\theta, h) := \frac{|\ell(\theta + h) - \ell(\theta)|}{\ell(\theta)}. \tag{148}$$

**Relative change of the model differential**  defined by the ratio

$$\Delta\big(\mathrm{d}u\big)(\theta, h) := \frac{\|\!|\mathrm{d}u_{\theta+h} - \mathrm{d}u_\theta\|\!|}{\|\!|\mathrm{d}u_\theta\|\!|}, \tag{149}$$

where $\|\!|\cdot\|\!|$ denotes the operator norm, that is,

$$\|\!|\mathrm{d}u_\theta\|\!| = \sup_{\|\delta\|_{\mathbb{R}^P}=1} \|\mathrm{d}u_\theta(\delta)\|_{\mathrm{L}^2(\Omega)}. \tag{150}$$

Lazy training occurs when $\Delta\big(\mathrm{d}u\big)(\theta, h) \ll \Delta(\ell)(\theta, h)$, meaning that the model differential remains essentially constant while the loss may vary significantly; *i.e.*, the training dynamics stay close to those of the linearized model.

Equivalently, this condition can be expressed by introducing the ratio

$$\kappa_{u_\theta}(\theta, h) := \frac{\Delta\big(\mathrm{d}u\big)(\theta, h)}{\Delta(\ell)(\theta, h)}, \tag{151}$$

and requiring that $\kappa_{u_\theta}(\theta, h) \ll 1$.

In Chizat et al. (2019), an explicit formula is derived in the setting of quadratic regression under standard gradient descent. However, the situation differs in our case because the optimization relies

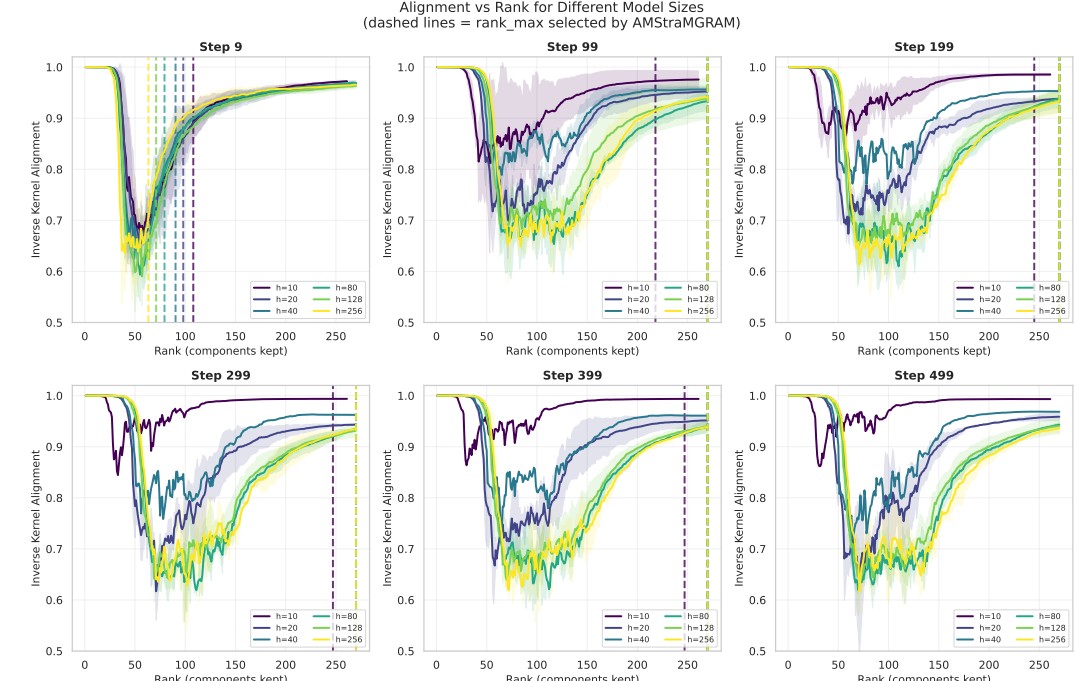

**Figure 21:** Inverse kernel alignment as a function of the rank $r$ (number of retained components) for the heat equation: Impact of Model Size. The dashed lines indicate the rank selected by AMStraMGRAM.

on the natural gradient. We therefore proceed step by step. As in Chizat et al. (2019), we start from the Taylor expansion

$$
\begin{aligned}
\ell(\theta + h) &= \ell(\theta) + \mathrm{d}\ell_\theta(h) + o(\|h\|_{\mathbb{R}^P}) \\
&= \mathcal{L}(u_\theta) + \langle \mathrm{d}u_\theta(h)\,,\,\nabla\mathcal{L}\rangle_{\mathrm{L}^2(\Omega)} + o(\|h\|_{\mathbb{R}^P}) \\
&= \tfrac{1}{2}\,\|u_\theta - f\|^2_{\mathrm{L}^2(\Omega)} + \langle \mathrm{d}u_\theta(h)\,,\,u_\theta - f\rangle_{\mathrm{L}^2(\Omega)} + o(\|h\|_{\mathbb{R}^P}),
\end{aligned}
\tag{152}
$$

from which we obtain, for $\|h\|_{\mathbb{R}^P}$ sufficiently small,

$$
|\ell(\theta + h) - \ell(\theta)| \simeq |\mathrm{d}\ell_\theta(h)| = \left|\langle \mathrm{d}u_\theta(h)\,,\,\nabla\mathcal{L}\rangle_{\mathrm{L}^2(\Omega)}\right| = \left|\langle \mathrm{d}u_\theta(h)\,,\,u_\theta - f\rangle_{\mathrm{L}^2(\Omega)}\right|.
\tag{153}
$$

Since $h$ follows the regularized natural gradient at level $N$, we have the identity

$$
\mathrm{d}u_\theta(h) = \eta\,\Pi_N(\nabla\mathcal{L}) = \eta\,\Pi_N(f - u_\theta).
\tag{154}
$$

Substituting equation 154 into equation 153 gives

$$
|\ell(\theta + h) - \ell(\theta)| \simeq \left|\langle \eta\,\Pi_N(f - u_\theta)\,,\,u_\theta - f\rangle_{\mathrm{L}^2(\Omega)}\right| = \eta\,\|\Pi_N(f - u_\theta)\|^2_{\mathrm{L}^2(\Omega)}.
\tag{155}
$$

Thus, the quantity $\Delta(\ell)$ in Equation (148) becomes, under the regularized natural gradient at level $N$, denoted $\Delta(\ell)(\theta, N)$,

$$
\Delta(\ell)(\theta, N) \simeq \frac{\eta\,\|\Pi_N(f - u_\theta)\|^2_{\mathrm{L}^2(\Omega)}}{\tfrac{1}{2}\,\|f - u_\theta\|^2_{\mathrm{L}^2(\Omega)}}.
\tag{156}
$$

Let us now focus on the relative change of the model differential $\Delta(\mathrm{d}u)$. Recall first that the SVD of $\mathrm{d}u_\theta$ is given by

$$
\mathrm{d}u_\theta = V_{\boldsymbol{\theta}}\,\Delta_{\boldsymbol{\theta}}\,U_{\boldsymbol{\theta}}^\top.
\tag{157}
$$

In particular,

$$\||\mathrm{d}u_\theta\|| = \sup_{\|\delta\|_{\mathbb{R}^P}=1} \|\mathrm{d}u_\theta(\delta)\|_{\mathrm{L}^2(\Omega)} = \Delta_{\theta_{\max}}. \tag{158}$$

Next, for any $\delta \in \mathbb{R}^P$, another Taylor expansion yields

$$\mathrm{d}u_{\theta+h}(\delta) = \mathrm{d}u_\theta(\delta) + \mathrm{d}^2 u_\theta(\delta, h) + o(\|h\|_{\mathbb{R}^P}), \tag{159}$$

from which we deduce, for $\|h\|_{\mathbb{R}^P}$ sufficiently small,

$$\||\mathrm{d}u_{\theta+h} - \mathrm{d}u_\theta\|| \simeq \||\mathrm{d}^2 u_\theta(\cdot, h)\|| \leqslant C_2 \, \|h\|_{\mathbb{R}^P}, \tag{160}$$

where

$$C_2 := \||\mathrm{d}^2 u_\theta\|| = \sup_{\substack{\|\delta_1\|_{\mathbb{R}^P}=1 \\ \|\delta_2\|_{\mathbb{R}^P}=1}} \|\mathrm{d}^2 u_\theta(\delta_1, \delta_2)\|_{\mathrm{L}^2(\Omega)}. \tag{161}$$

Combining equation 158 and equation 160 yields

$$\Delta(\mathrm{d}u) \lesssim \frac{C_2 \, \|h\|_{\mathbb{R}^P}}{\Delta_{\theta_{\max}}}. \tag{162}$$

Substituting this estimate into Equation (151) yields

$$\kappa_{u_\theta}(\theta, N) \lesssim \frac{C_2 \, \|h\|_{\mathbb{R}^P} \, \frac{1}{2} \|f - u_\theta\|^2_{\mathrm{L}^2(\Omega)}}{\Delta_{\theta_{\max}} \, \eta \, \|\Pi_N(f - u_\theta)\|^2_{\mathrm{L}^2(\Omega)}}. \tag{163}$$

We now estimate $\|h\|_{\mathbb{R}^P}$. Since $h$ follows the natural-gradient update of Equation (10),

$$\boldsymbol{\theta}_{t+1} \leftarrow \boldsymbol{\theta}_t - \eta \, \widehat{\phi}^\dagger \widehat{\nabla \mathcal{L}}_{\boldsymbol{\theta}_t}; \qquad \widehat{\phi}_{i,p} := \partial_p u_{\boldsymbol{\theta}}(x_i); \qquad \left(\widehat{\nabla \mathcal{L}}_{\boldsymbol{\theta}}\right)_i := u_{\boldsymbol{\theta}}(x_i) - f(x_i), \tag{10}$$

we obtain, omitting the learning rate $\eta$ for readability,

$$\frac{1}{\eta^2} \|h\|^2_{\mathbb{R}^P} = \left\|\widehat{\phi}^\dagger \widehat{\nabla \mathcal{L}}_{\boldsymbol{\theta}_t}\right\|^2_{\mathbb{R}^P}. \tag{164}$$

Using the SVD $\widehat{\phi}^\dagger = \widehat{U} \widehat{\Delta}^\dagger \widehat{V}^T$, this becomes

$$\begin{aligned}
\frac{1}{\eta^2} \|h\|^2_{\mathbb{R}^P} &= \left\|\widehat{U} \widehat{\Delta}^\dagger \widehat{V}^T \widehat{\nabla \mathcal{L}}_{\boldsymbol{\theta}_t}\right\|^2_{\mathbb{R}^P} \\
&= \widehat{\nabla \mathcal{L}}_{\boldsymbol{\theta}_t}^T \widehat{V} \widehat{\Delta}^\dagger (\widehat{U}^T \widehat{U}) \widehat{\Delta}^\dagger \widehat{V}^T \widehat{\nabla \mathcal{L}}_{\boldsymbol{\theta}_t} \\
&= \widehat{\nabla \mathcal{L}}_{\boldsymbol{\theta}_t}^T \widehat{V} (\widehat{\Delta}^\dagger)^2 \widehat{V}^T \widehat{\nabla \mathcal{L}}_{\boldsymbol{\theta}_t} \\
&= \sum_{i=1}^N \frac{\left\|\widehat{V}_i^T \widehat{\nabla \mathcal{L}}_{\boldsymbol{\theta}_t}\right\|^2_{\mathbb{R}^P}}{\widehat{\Delta}_i^2} \leqslant \frac{1}{\widehat{\Delta}_N^2} \left\|\Pi_N \widehat{\nabla \mathcal{L}}_{\boldsymbol{\theta}_t}\right\|^2_{\mathbb{R}^P},
\end{aligned} \tag{165}$$

where $N$ is the number of retained components.

By Equation (138) (see the proof of Proposition 1), we have almost surely

$$\lim_{S \to \infty} \frac{1}{\widehat{\Delta}_N^2} \left\|\Pi_N \widehat{\nabla \mathcal{L}}_{\boldsymbol{\theta}_t}\right\|^2_{\mathbb{R}^P} = \frac{1}{\Delta_N} \|\Pi_N(f - u_\theta)\|^2_{\mathrm{L}^2(\Omega)}. \tag{166}$$

Substituting this into Equation (163), we obtain almost surely

$$\kappa_{u_\theta}(\theta, N) \lesssim \frac{C_2 \, \|\Pi_N(f - u_\theta)\|_{\mathrm{L}^2(\Omega)} \, \frac{1}{2} \|f - u_\theta\|^2_{\mathrm{L}^2(\Omega)}}{\Delta_{\theta_{\max}} \, \Delta_N \, \|\Pi_N(f - u_\theta)\|^2_{\mathrm{L}^2(\Omega)}}. \tag{167}$$

We now use the decomposition

$$\|f - u_\theta\|^2_{\mathrm{L}^2(\Omega)} = \|\Pi_N(f - u_\theta)\|^2_{\mathrm{L}^2(\Omega)} + \left\|\Pi_N^\perp(f - u_\theta)\right\|^2_{\mathrm{L}^2(\Omega)}. \tag{168}$$

By construction of the flattening phase,

$$\left\|\Pi_N^{\perp}(f - u_\theta)\right\|_{L^2(\Omega)}^2 \leqslant \epsilon^2, \tag{169}$$

where $\epsilon$ is the target accuracy. Plugging into Equation (167) yields

$$\kappa_{u_\theta}(\theta, N) \lesssim \frac{C_2 \frac{1}{2}\left(\|\Pi_N(f - u_\theta)\|_{L^2(\Omega)} + \frac{\epsilon^2}{\|\Pi_N(f - u_\theta)\|_{L^2(\Omega)}}\right)}{\Delta_{\theta_{\max}}\Delta_N}. \tag{170}$$

Since $\|\Pi_N(f - u_\theta)\|_{L^2(\Omega)} \gtrsim \epsilon$, the second term is dominated, and redefining $C := \frac{(1+\epsilon)C_2}{2}$, we obtain

$$\kappa_{u_\theta}(\theta, N) \lesssim \frac{C\,\|\Pi_N(f - u_\theta)\|_{L^2(\Omega)}}{\Delta_{\theta_{\max}}\Delta_N} \leqslant \frac{C\,\|f - u_\theta\|_{L^2(\Omega)}}{\Delta_{\theta_{\max}}\Delta_N}. \tag{171}$$

Finally, note that $\Delta_{\theta_{\max}} = \Delta_0$ is typically several orders of magnitude larger than $\|f - u_\theta\|_{L^2(\Omega)}$, and that $\Delta_N$ decreases exponentially fast. Hence, as $N \to 0$, the ratio $\kappa_{u_\theta}(\theta, N)$ decays exponentially to zero, which is precisely the hallmark of the lazy-training regime.

