# OpenReview forum: "AMStraMGRAM : Adaptive Multi-cutoff Strategy Modification for ANaGRAM"
_ICLR.cc/2026/Conference — Submitted to ICLR 2026_

### Official Review · Reviewer_yRr7 · 2025-10-21

**Soundness:** 3
**Presentation:** 2
**Contribution:** 2
**Rating:** 2
**Confidence:** 4

**Summary:**

This paper introduces AMStraMGRAM, an adaptive optimization algorithm designed to improve the training of Physics-Informed Neural Networks (PINNs). The work builds directly upon ANaGRAM, a natural-gradient descent method that uses a fixed cutoff to regularize the singular value decomposition (SVD) of the feature matrix during training. AMStraMGRAM addresses the “flattening phenomenon” by dynamically adjusting the cutoff at each iteration (compared to the fixed cutoff in AnaGRAM). The algorithm tracks the intersection between the RCE curve and the singular values, switching between an "intersection rank" and a "precision rank" to ensure the training progresses efficiently towards a target precision. This adaptive, multi-cutoff strategy aims to avoid early stagnation and exploit the flattening phenomenon to force a final drop in the training loss. The method is validated on several benchmark PDEs (Heat, Laplace, Allen-Cahn), showing significant performance improvements over ANaGRAM.

**Strengths:**

1. The analysis of the "flattening phenomenon" via the Reconstruction Error (RCE) metric is novel.

2. The paper is theoretically depth as it provides a rigorous functional-analytic framework, explaining why regularization is necessary and drawing a clear distinction between ridge and cutoff regularization.

3. The proposed AMStraMGRAM algorithm is a direct simple but effective modification of the ANaGRAM. The cutoff of the rank is dynamically adjusted, based on the theoretical analysis.

**Weaknesses:**

1. While the theoretical framework is impressive, it primarily provides an explanation for the necessity of regularization and an interpretation of the method. It does not provide rigorous convergence analysis of the training loss for the adaptive cutoff scheme itself. It seems that convergence analysis has been established for natural gradient method to train PINN problems (arXiv:2408.00573) at the same time. So the main incrementation of AMStraMGRAM to ANaGRAM is the adaptive cutoff strategy of the rank, which is too limited for a top conference.

2. The experiments are not comprehensive in today's PINNs’ community. Although the AMStraMGRAM shows ‘L2 error improvements of up to 8 orders of magnitude’ compared to ANaGRAM, it is on very simple Heat and Laplace equation with simple smooth solutions, which is less meaningful. No experiments conducted on complex PINN problems (such as the NS equation with turbulence, KS equation with chaotic property, as in PINNacle NIPS24’) to show the practical improvements over other optimizers on practical problems.

**Questions:**

1. See weakness above.

2.How does the float precision affect the results of AMStraMGRAM? The L2 error is very small. The author uses 64p in Jax, while many works conduct experiments with default 32p in Pytorch.

3.In line 392, why the author compare ANaGRAM with the same equation but with ‘modified dataset’?

4.Is the feature matrix the same as the jacobian matrix J in arXiv:2408.00573? What is the difference between your (10) with existing natural gradient methods (arXiv: 1905.11675, arXiv:2408.00573) besides the SVD decomposition instead of the pesodu-inverse of the jacobian matrix?

5.Can it be combined with approximated natural gradient descent (NGD) methods like F-KAC?  In general deep learning, K-FAC is often used instead of 'exact' NGD.In particular, theoretical analysis of K-FAC has already been developed for shallow neural networks.

---

> ### Author Response · Authors · 2025-11-25
>
> We thank the reviewer for their positive assessment of the quality of our work.
>
> **Lack of convergence analysis** We agree that, in its current form, the paper primarily develops the theoretical justification for cutoff regularization and does not provide a full convergence analysis. However, we respectfully disagree with the relevance of the reference provided. From our understanding, arXiv:2408.00573 presents a statistical convergence analysis in the NTK/lazy training regime (https://proceedings.neurips.cc/paper/2019/hash/ae614c557843b1df326cb29c57225459-Abstract.html), which presupposes that the network is sufficiently large to remain close to its initialization throughout training.
>
> In contrast, the effectiveness of our algorithm stems precisely from the fact that it operates outside the NTK regime for most of training. The adaptive cutoff allows the model to remain in a feature-learning phase as long as needed, and only enters the NTK/lazy training regime during the final phase, which corresponds to the flattening behavior we describe. We will clarify this distinction in the revised version.
>
> **More challengin PDEs** As stated in previous responses, our primary goal is to reduce the optimization error, which we view as one of the pivotal challenges in PINNs and a key criticism raised in the literature. The problems you mention, however, involve additional difficulties: turbulence and chaotic dynamics require sophisticated techniques to control integration error (through advanced sampling or quadrature strategies) and approximation error (through specialized architectures).
>
> We will nevertheless include additional experiments on more complex settings, focusing specifically on how our algorithm reduces the residual error, independently of generalization performance, which must be addressed through appropriate sampling and architectural choices.
>
> **Use of float64 instead of float32.** We have not yet tested the method using float32. That said, recent work has reported that using float64 is an important factor in preventing PINN failure modes (see, e.g., (https://arxiv.org/abs/2205.09332) and (https://arxiv.org/abs/2505.10949).
>
> **About modified dataset** The use of a “modified dataset” is intended to ensure a fair evaluation of the optimizer, by mitigating overfitting caused by inadequate sampling. As our Allen–Cahn example already illustrates, when the sampling does not sufficiently resolve regions of rapid variation, the model may fit the training points yet generalize very poorly, reflecting integration error rather than optimization behavior. The same issue arises, for instance, in Burgers’ equation, where collocation points must be concentrated near the shock to obtain a meaningful assessment.
>
> More broadly, this choice highlights once again the important distinction we emphasize throughout the rebuttal: sampling and quadrature determine the integration error, whereas our method specifically targets optimization error. Adjusting the dataset was therefore necessary to ensure that the comparison between ANaGRAM, AMStraMGRAM, and other methods reflects genuine differences in optimization performance rather than artifacts caused by undersampling.
>
> **Feature matrix and its relationship with the empirical Jacobian** Yes, the feature matrix indeed corresponds to the empirical Jacobian matrix. The update described in (10) is precisely the vanilla ANaGRAM update introduced in the ANaGRAM paper (Schwencke and Furtlehner, ICLR 2025). As detailed in their contribution, the main difference with existing natural-gradient formulations lies in the underlying theoretical motivation: ANaGRAM introduces the notions of empirical tangent space and empirical natural gradient, and shows that the vanilla ANaGRAM update is a principled approximation of this geometric construction.
>
> **Combination with K-FAC** Combining our method with approximate NGD techniques such as K-FAC should be feasible, as it would simply require deriving an alternative approximation of the empirical natural gradient to replace the vanilla ANaGRAM update.
> We also note that our adaptive regularization strategy applies not only to the empirical natural gradient but equally to the standard natural gradient. We will make this clearer in the revised version.

---

> > ### Comment · Reviewer_yRr7 · 2025-11-28
> > **Thank you**
> >
> > Thanks for the reply. My main concern about the limited improvement (AMStraMGRAM to ANaGRAM (ICLR'25) is the adaptive cutoff strategy of the rank) is not solved, and I find no actual revisions to the manuscript. So I tend to the rejection at this stage, but hope the authors can combine the opinions by all reviewers to enrich the manuscript.

---

### Official Review · Reviewer_6JRF · 2025-10-21

**Soundness:** 3
**Presentation:** 2
**Contribution:** 2
**Rating:** 4
**Confidence:** 4

**Summary:**

This paper focuses on enhancing the optimization process of Physical Information Neural Networks (PINNs). Specifically, it improves ANaGRAM, an advanced natural gradient-based optimizer. Through an in-depth analysis of ANaGRAM's training dynamics, this paper identifies and defines a critical phenomenon termed “Flattening.” Building upon this theoretical insight, the authors propose AMStraMGRAM, an adaptive multi-cutoff strategy that dynamically and automatically adjusts the cutoff value, thereby eliminating the need for manual tuning of this hyperparameter.

The contributions of this paper can be divided into three closely interrelated aspects:
1) **Discovering and Explaining the “Flattening Phenomenon”**: The paper introduces a novel metric called “Reconstruction Error (RCE).” Through RCE, it reveals the training dynamics of ANaGRAM: Fixed truncation hinders or delays this process, whereas adaptive strategies can actively guide and leverage it.
2) **Proposing the AMStraMGRAM Adaptive Optimizer**: The algorithm can dynamically determine the optimal truncation value for each iteration by tracking the intersection point between the RCE and singular value curve in real time.
3) **Empirical Performance**: On multiple classical PDE benchmark problems, AMStraMGRAM reduces errors by several orders of magnitude compared to ANaGRAM, achieving machine accuracy on some problems and outperforming other types of advanced optimizers.

**Strengths:**

1.	**Identified and defined the “flattening phenomenon” in the training dynamics**: by introducing the novel metric of “reconstruction error,” it clearly reveals the underlying mechanism behind the sudden drop in the loss function during the late stages of ANaGRAM training.

2.	**Proposed AMStraMGRAM Algorithm**: By dynamically tracking the intersection point between the RCE and singular value curve, the algorithm intelligently selects the truncation rank, thereby avoiding issues of “premature flattening” (training stagnation) or “delayed convergence” (insufficient accuracy) caused by fixed truncation.

**Weaknesses:**

The main issues with the paper are as follows:

1.	The entire effort of this paper focuses on the adaptation of a single hyperparameter (the cutoff value) within the optimizer. While executed with great skill, this represents essentially a refinement in algorithmic engineering rather than a breakthrough in scientific concepts.

2.	While standard benchmark PDEs such as Laplace, Burgers, and Allen-Cahn equations are commonly used for testing, they lack more challenging cases. PINNs typically perform poorly for equations with solutions exhibiting large variations or strong multiscale characteristics. Without such tests, it remains unclear whether the method enhances the model's intrinsic fitting capability or merely finds smoother solutions more efficiently.

3.	All experiments were conducted within regular domains (e.g., rectangular domains). However, the vast majority of problems in science and engineering are defined over complex geometries (e.g., domains with internal cavities, non-convex regions, or manifolds). On complex domains, collocation sampling and boundary condition treatment themselves pose significant challenges. The paper urgently needs to incorporate experiments on irregular domains to test the robustness of its methods under more realistic settings.

4.	The paper demonstrates AMStraMGRAM's significant advantage in final accuracy but overlooks the computational time required to achieve this precision.

**Questions:**

1. AMStraMGRAM introduces a new hyperparameter—target accuracy $\varepsilon$. How sensitive is the algorithm to the choice of $\varepsilon$? How does the setting of $\varepsilon$ relate to accuracy?

2. Section 6 of the paper presents an excellent discussion on overfitting in the Allen-Cahn equation, highlighting the need for “co-designing the optimizer and sampler.” A natural question arises: Can AMStraMGRAM seamlessly collaborate with existing adaptive sampling methods, such as Residual-Adaptive Distribution (RAD)?

3. All experiments in this paper are based on standard multilayer perceptrons. However, many modern PINN studies employ specialized network architectures to mitigate issues such as spectral bias, including Fourier feature networks or modulated networks. Is AMStraMGRAM's adaptive strategy compatible with these architectures? Does its performance improvement depend on specific properties of the MLP? How will testing on networks with different spectral characteristics affect the emergence of the “flattening” phenomenon and the algorithm's final performance? This concerns the method's general applicability.

4. Beyond the overfitting observed in the Allen-Cahn equation, have the authors encountered instances of poor algorithmic performance or failure under other settings—such as varying network width/depth, activation functions, or equation parameters? Could the “decision boundary” of AMStraMGRAP be described more systematically—specifically, under what conditions (problem characteristics, hyperparameter selection) does the algorithm operate stably, and under which conditions might it fail? This is crucial for potential users applying the method in practice.

---

> ### Author Response · Authors · 2025-11-25
>
> We thank the reviewer for those insightful considerations.
>
> **Remark on a limited novelty** While we understand this viewpoint, we respectfully disagree. As you mention, our results show that the cutoff is a dominant performance bottleneck: fixed-rank regularization leads to either premature flattening or inability to reach high precision. Our contribution isolates this optimizer-level limitation and provides a principled mechanism to resolve it.
>
> Restated differently, cutoff adaptation addresses the optimization error of PINNs, which was a key bottleneck of PINNs as highlighted by reviewer 7H9Y. To the best of our knowledge, most work in PINNs has focused in lowering the integration error (for instance by investigating sampling strategies) or the approximation error (for instance by searching for refined architectures), which makes our work original.
> Moreover, we provide in the appendix a careful theoretical analysis of the role played by the cutoff and its theoretical importance beyond algorithmic engineering purposes.
>
> **More challenging PDEs** Thank you for highlighting this important point. Our goal is not to solve PDEs with multiscale or chaotic behavior, where sampling and approximation errors are at least as challenging as the optimization error, but rather to specifically address the optimization error in PINNs. The benchmark equations we selected allow us to isolate this effect, and we will make this scope more explicit in the revised version. Nevertheless, we believe that tackling the problems you mention requires a careful combination of complementary approaches, including adapted sampling strategies and principled architectures design.
>
> In this respect, we believe that our current choices already cover representative study cases (linear, nonlinear, higher dimensional, etc.). Nonetheless, we can certainly include additional experiments that more directly highlight the optimization capabilities of the algorithm, including scenarios where overfitting may arise, as in the Allen-Cahn example. At the same time, our strong performance on Burgers’ equation shows that **the method is not merely accelerating convergence toward smooth solutions**, since the solution develops a nearly discontinuous region in the middle.
>
> **Irregular domains** As a follow-up to the previous remark, we reiterate that our method addresses a different problem, namely, optimization issues. Moreover, developing methods for unconventional domains is a field of study in its own right.
>
> For completeness, we will include a brief illustration on non-convex domains (e.g., a star shape) and domains with internal cavities (e.g., a donut), without expanding into a full treatment of complicated domains, which lies beyond the scope of the present work.
>
> **Computational cost** We agree with this comment. The adaptive cutoff reuses quantities already computed in ANaGRAM, making the theoretical additional overhead constant. For completeness, we will include timing information comparing training durations as a function of the number of epochs in the revised version.
>
> **Target accuracy $\epsilon$.** The parameter epsilon specifies the target residual precision that the algorithm aims to achieve. By design, if the algorithm converges (i.e., if the final rank reaches 0), the resulting training residual is guaranteed to lie at or below this prescribed level. As shown in Figure 2, once the smallest retained singular value reaches epsilon, the algorithm enters the flattening regime, ensuring that the residual stabilizes beneath this threshold.
>
> In practice, when an appropriate network architecture and a sufficiently informative set of collocation points are used (as illustrated in our Heat and Laplace experiments), choosing epsilon close to machine precision typically yields the best results.
> Conversely, if convergence is not observed, epsilon can be increased by a few orders of magnitude, accepting a lower target precision but ensuring convergence. We will revise the manuscript to clarify this mechanism and provide a more intuitive explanation.

---

> > ### Author Response · Authors · 2025-11-25
> >
> > **Compatibility with adaptive sampling strategies** Thank you for this comment, which highlights an important subtlety. RAD is a dynamic sampling strategy, and abruptly changing the sampling points alters the empirical tangent space introduced in the ANaGRAM paper. As a result, the effect of modifying the sampling points is at least as significant as the effect of regularization, and a careful, progressively refined update scheme is required. Without such precautions, these abrupt changes destabilize the current adaptive cutoff strategy. Although the two approaches are compatible in principle, rigorously analyzing their interaction constitutes a separate research direction and lies outside the scope of the present submission.
> >
> > **Considering other NN architectures** As mentioned earlier, our paper focuses on addressing the optimization error in PINNs, whereas architectures choices primarily target the approximation error. Nevertheless, we agree that evaluating our method on a broader range of architectures would help demonstrate its general applicability. To this end, we will include additional experiments using Fourier features input mapping (https://arxiv.org/abs/2006.10739) and PIKANs (https://arxiv.org/pdf/2406.02917).
> >
> > **Failure cases for Amstramgram** We have not encountered catastrophic failure cases so far. Nevertheless, we agree that a more complete discussion of potential limitations would be valuable. In particular, the following situations may arise:
> >
> > * **The RCE does not fall below the prescribed precision.** This indicates a lack of approximation capacity rather than an optimization issue, meaning that the chosen model is simply not expressive enough. We will illustrate this case.
> >
> > * **An extremely poor initialization** (admittedly rare and not observed in practice) where the initial RCE lies above the entire singular-value spectrum, preventing meaningful rank selection. We will include an example of this scenario.
> >
> > * **The singular-value spectrum does not decay toward zero**, corresponding to a well-conditioned problem where regularization is unnecessary. We will clarify this case and provide illustrative examples in the revised version.

---

> > > ### Comment · Reviewer_6JRF · 2025-11-28
> > > **Thank you**
> > >
> > > Thanks for the reply. However, the authors make no revision and the response do not solve my conern on the W1 (a fixed cutoff to adaptive cutoff technique) and W2 (simple benchmarks outdate), and the remaning questions unsolved. I also agree with rjuU and yRr7 that more theoretical analysis is needed to strenghth the novelty of this paper. Therefore, I keep my score.

---

### Official Review · Reviewer_rjuU · 2025-10-30

**Soundness:** 3
**Presentation:** 4
**Contribution:** 3
**Rating:** 6
**Confidence:** 4

**Summary:**

The paper proposes AMStraMGRAM, an adaptive multi-cutoff variant of ANaGRAM, which is a natural-gradient-based optimizer for training Physics-Informed Neural Networks (PINNs). ANaGRAM regularizes the pseudo-inverse of the feature matrix by a fixed cutoff in singular values. Based on this, AMStraMGRAM generalizes it by introducing a dynamic, multi-cutoff strategy that tracks the evolution of the reconstruction error (RCE) and singular values during training.

**Strengths:**

(1) The work is well motivated, as the adaptive processing for singular values should benefit the natural gradient method.

(2) The paper is well written.

(3) The empirical results are strong that the proposed method significantly improves the accuracy of training PINNs over other optimizers.

**Weaknesses:**

(1) While the intuition behind adaptive cutoff dynamics is persuasive, the paper lacks a formal convergence analysis. Is the algorithm convergent to the stationary points?

(2) AMStraMGRAM requires SVD at every iteration, similar to ANaGRAM. Although it reuses intermediate quantities, a discussion of computational overhead is missing. I think the computational cost for natural gradient is high if the sample size is large.

(3) The algorithm is deterministic. However, due to the limited GPU memory as well as the large sample size, stochastic optimizers are favourable in training PINNs. Is the algorithm valid with stochastic gradients/mini-batch?

**Questions:**

(1) I found that ANaGRAM exactly resembles classical natural gradient descent. I think the main technique is applying SVD to the matrix and adding the cutoff in the singular values. Besides this, it is exactly the natural gradient descent. Do I understand correctly?

---

> ### Author Response · Authors · 2025-11-25
>
> We thank the reviewer for their encouraging comments.
>
> **Lack of a formal convergence analysis** A fully general convergence theory for the adaptive scheme is in our opinion a very challenging problem, far out of the scope of this work due to the nonlinear and operator-dependent nature of PINNs. However, we will include a clearer discussion of convergence behavior, notably by adding some theoretical and experimental evidences suggesting that the flattening phase corresponds to a lazy training regime (https://proceedings.neurips.cc/paper/2019/hash/ae614c557843b1df326cb29c57225459-Abstract.html).
>
> **Computational overhead discussion** You are absolutely right, the computational cost is dependent on sample size.
> The adaptive cutoff reuses quantities already computed in ANaGRAM, so the additional overhead is negligible. We will clarify this point in the manuscript. In ANaGRAM paper, it is shown that the method scales as $O(\min(PS^2,P^2S))$ and AMStraMGRAM inherits the same complexity. We will also add computational time tables to assert this.
>
> **Stochastic gradient compatibility** This question involves deeper considerations: naive minibatching (resampling points each iteration) disrupts the empirical natural gradient and leads to instabilities. Extending the method to spatially structured minibatches, where each subdomain has its own empirical tangent space and cutoff, is possible but nontrivial. We will clarify this limitation and position stochastic versions as a promising direction for future work rather than part of the current scope.
>
> We also emphasize that designing stable minibatch procedures is a well-known challenge for natural-gradient and second-order methods in general, not a limitation specific to AMStraMGRAM, as mentioned for example in https://arxiv.org/pdf/1605.06049 for L-BFGS.
>
> **Relation to classical natural gradient descent** Yes, ANaGRAM coincides algebraically with a natural-gradient step, but applied within an empirical tangent space (as defined in the ANaGRAM paper),  which is of peculiar interest for designing sampling/quadrature methods with a geometrical perspective. We will make this conceptual distinction clearer, and we will state more explicitly that the adaptive cutoff strategy applies both to empirical and classical natural gradients.

---

> > ### Comment · Reviewer_rjuU · 2025-11-27
> >
> > I agree with reviewer yRr7 that this paper lacks a theoretical analysis, since this paper is targeted as an optimization method. The main incrementation of AMStraMGRAM to ANaGRAM is the adaptive cutoff strategy of the rank, which is too limited. The rebuttal does not address my concern about the theoretical analysis. There are several aspects we can analyze (e.g., standard nonconvex convergence, stochastic objective,  overparameterization/NTK, etc.), but none of them are explored.

---

### Official Review · Reviewer_7H9Y · 2025-10-30

**Soundness:** 4
**Presentation:** 4
**Contribution:** 4
**Rating:** 8
**Confidence:** 5

**Summary:**

The authors analyze and propose an extension of ANaGRAM, a natural gradient method technique for solving PDEs in the physics-informed neural network setting. I have extensively studied the original ANaGRAM work, and the current work represents a non-trivial extension. Notably, the authors are able to obtain convergence "to machine-precision". This is a timely paper, as many are attempting to achieve machine-precision accuracy from PINNs. While this is a trivial achievement for conventional finite element solutions to PDEs, PINNs have struggled to achieve this and this is viewed by many as an important stepping stone to achieving AI/ML models with comparable quality of science/engineering-relevance.

For example, deepmind just launched an effort to develop a numerical counterexample for the millinium prize of finite time blowup for navier-stokes. In contrast, my impression is that this has the desirable feature of being an "optimization only" solution that achieves desired accuracy with no ad hoc modification of the loss or architecture.

I have a few suggestions about how the text could be further improved, but overall this is a strong paper and should be accepted.

**Strengths:**

See summary. The paper provides a notable and timely contribution for what has been an outstanding issue with PINNs. The technique is well-rooted in analysis and provides a principled solution to the problem.

**Weaknesses:**

As noted, the paper could benefit from some minor modifications.

(1) The work should be contextualized in the context of other methods. The only one to my knowledge that has been able to achieve this accuracy is from Ching-Yao Lai's group at Stanford (Wang, Yongji, and Ching-Yao Lai. "Multi-stage neural networks: Function approximator of machine precision." Journal of Computational Physics 504 (2024): 112865.), which is the technique deepmind used for their work (Wang, Yongji, et al. "Discovery of Unstable Singularities." arXiv preprint arXiv:2509.14185 (2025). These are very recent works but would help frame the importance of the fundamental issue and highlights the way in which PINNs can answer ill-posed questions that are challenging for conventional FEM. The authors may also be interested in (Trask, Nathaniel, Amelia Henriksen, Carianne Martinez, and Eric Cyr. "Hierarchical partition of unity networks: fast multilevel training." In Mathematical and Scientific Machine Learning, pp. 271-286. PMLR, 2022.), where authors achieved machine precision but only for regression problems. All of these works achieve accuracy through **hierarchy**, while the current paper achieves it purely through the optimizer, which is interesting.

(2) One more study should be added to the end to disentangle the accuracy of the solution from the sampling error. Formally I would say this achieves a machine precision **residual**, not **error**. The final plot showing that the error is of comparable wavelength to the grid spacing is an interesting one. The authors could easily add another plot showing the residual (log x axis) vs the L2 error (log y axis) for verifying collocation densities (h = sqrt(N_collocationpoints)). Alternatively, the authors could sample residual points from a uniform distribution on the square at each step of training to avoid grid dependence altogether. If the authors include this study I will further increase my score.

**Questions:**

No questions, great paper.

---

> ### Author Response · Authors · 2025-11-25
>
> Thank you very much for the appreciation of our work and the nice comments. We are glad that you find our contribution insightful.
>
> **Missing literature discussion**: We appreciate the suggested references (Lai et al., DeepMind’s “Discovery of Unstable Singularities”, Trask et al.). These works indeed pursue machine-precision accuracy through hierarchical multilevel structures, whereas our approach achieves low residual error through the optimizer alone. We will expand the related-work section to make this distinction explicit and to contextualize the complementary nature of these approaches.
>
> **Disentangling solution accuracy from the sampling error** We agree with the reviewer that our method attains machine precision on the residual and not on the $L^2$ error. This last metric depends crucially on the choice of points.
> We will clarify this point in the revised version by stating that the investigation of a proper sampling/quadrature strategy is the next step, now that limitations of the optimizer are not the main bottleneck anymore. As you suggested, we will include in the revision a short analysis illustrating the relationship between collocation density and $L^2$ error, without entering into the full study of stochastic or moving-grid sampling strategies, which raises additional stability considerations.

---

### Author Response · Authors · 2025-11-25
**General Feedback**

## General Feedback

We thank all reviewers for their careful reading, insightful comments, and constructive critics. As far as we understand, the main concerns can be summarised as follows:
1. a lack of convergence results;
2. compatibility of the method with common sampling techniques beyond fixed datasets;
3. the need for a broader range of PDE benchmarks, including more challenging cases.


## **Concerning the first line of critics (convergence guarantees)**

While we acknowledge that the current version lacks an explicit discussion of convergence, we respectfully disagree with the implication that this undermines the contribution. As we clarify throughout the rebuttal, the convergence analyses cited by reviewers rely on the NTK or lazy-training regime (https://proceedings.neurips.cc/paper/2019/hash/ae614c557843b1df326cb29c57225459-Abstract.html), whereas our method **intentionally operates outside this regime** for most of the training. The adaptive cutoff is effective precisely because it allows the (empirical) tangent space to expand in a feature-learning phase, an essential dynamical behavior that violates the assumptions of existing natural-gradient convergence theorems.

A fully general convergence theory for the adaptive scheme is therefore far beyond the scope of current PINN theory, given the nonlinear and operator-dependent nature of these models. Nonetheless, we will enrich the revised version with a clearer discussion of convergence behavior. In particular, we will provide theoretical arguments and numerical evidence showing that the **flattening phase corresponds to a lazy-training regime**, while the earlier phase fundamentally lies outside the approximation regimes covered by existing analyses. In our view, this mixed behavior—feature learning followed by lazy training—is a key insight offered by our work, and motivates the need for adaptive regularization.

---

## **Concerning the second line of critics (sampling strategies and stochasticity)**

Several reviewers questioned the compatibility of our method with stochastic or adaptive sampling strategies. Here again, our goal is to clarify a crucial point: **sampling and optimization interact through the empirical tangent space**. Changing the collocation points, especially abruptly, as with naive minibatching or dynamic sampling such as RAD, modifies the empirical tangent space and the feature matrix, often as much as regularization itself. Without stabilization, these shifts can destabilize the current adaptive cutoff strategy.

Our method therefore targets a different aspect of the problem: **optimization error**, under sampling conditions that keep the empirical tangent geometry consistent across iterations. We clarify that extending AMStraMGRAM to spatially structured or dynamically updated sampling schemes is in principle feasible, but requires careful control (e.g., damping or continuity constraints). This is an interesting research direction on its own, and lies beyond the present scope. We believe our analysis provides the conceptual tools to understand why naive minibatching fails and how principled sampling–optimizer co-design could be developed.

---

## **Concerning the third line of critics (breadth and difficulty of PDE benchmarks)**


The request for more challenging PDEs and irregular domains highlights an important distinction that we emphasize in our rebuttal: **integration and approximation errors are fundamentally different from optimization error**, and our contribution specifically targets the latter. Highly multiscale or chaotic PDEs (e.g., Navier–Stokes, Kuramoto–Sivashinsky) introduce sampling and approximation difficulties that completely dominate the phenomena we study (rank growth, flattening, RCE dynamics). While it is, in principle, possible to isolate the optimizer’s effect in such settings, doing so is not particularly meaningful: severe overfitting and approximation errors would overshadow any improvement in optimization, and the resulting solutions would remain unsatisfactory regardless of the optimizer.

For this reason, our benchmarks were chosen to control for these confounding factors. Nonetheless, we agree that the scope should be stated more clearly. In the revised version, we will expand the experiments to include additional architectures (Fourier features, PIKANs) to demonstrate applicability beyond MLPs.

---

### Author Response · Authors · 2025-12-03
**Final Rebuttal Comment**

Due to this year’s configuration, this general message is addressed to the area chair to summarise what we believe are the key points of the rebuttal discussion and how they motivated our revised submission.

Minor additions have been made in the color violet.

**Convergence and the role of the Overparametrised NTK regime.**

Several reviewers expressed concerns about the lack of explicit convergence results. We would like to clarify that our earlier wording may have unintentionally suggested that our algorithm lacks convergence guarantees, in contrast to standard optimizers. This is not the case. To make this explicit, we added an appendix (in blue) showing that, for any fixed cutoff rank, the training dynamics fall within a kernel-regression regime under the usual classical assumptions. In particular, for each fixed cutoff, the algorithm converges to the corresponding projected solution.


However, the over-parameterized NTK regime used to establish such guarantees is not representative of the actual training dynamics of neural networks. This is precisely why we did not rely solely on existing convergence analyses from the literature. As noted by reviewers, prior works typically study the over-parameterized NTK regime, in which optimization effectively reduces to kernel regression and feature development is negligible.

In contrast, our method explicitly does not operate in the over-parameterized NTK regime during training. Our algorithm leverages a fundamentally different phenomenon (feature development before flattening) by intentionally delaying flattening as long as possible and then promoting it once the target precision is reached. Because of this feature-learning phase, classical NTK-based convergence analyses are not appropriate during the early and intermediate stages of training.


Our adaptive cutoff strategy is nonetheless theoretically grounded. We carefully characterize the nature of cutoff regularization and show that it corresponds to a new generalized Green’s function. We then analyze the flattening mechanism and demonstrate its direct connection to the sudden drops observed in the training loss.

As a new contribution in the revised version (in light-red), we further show that the end of the flattening phase corresponds precisely to the lazy-training regime (https://arxiv.org/pdf/1812.07956). Consequently, the end of training can be analyzed using the same standard arguments employed in prior works on convergence in the lazy regime, directly addressing the reviewers’ concerns about theoretical foundations.

**Sampling, stochasticity, and batching strategies.**

A second concern raised by reviewers was whether classical stochastic or adaptive sampling strategies could be applied to our method. In response, we have added (in teal) empirical evidence illustrating how sampling interacts with our optimization dynamics. These results show that adapting standard batching schemes requires careful design: different batches can induce different empirical tangent spaces, especially at intermediate cutoff values.

**Optimization vs. approximation error.**

Finally, reviewers asked for a clearer separation between approximation and optimization errors. We now emphasize this distinction explicitly (in red) and provide additional empirical evidence (Fig. 4) showing how the density of the quadrature or sampling scheme can influence the $L^2$ error even when the optimization error is held fixed.

We thank the area chair for considering these clarifications and the corresponding improvements made in the revised version.

---

### Meta-Review · Area_Chair_o2xY · 2026-01-05

**Summary:**

The paper proposes an optimization algorithm for training Physics Informed Neural Networks (PINN) representations and provides an analysis on the different phases of learning based on the reconstruction error of the functional gradient. The optimization algorithm is an extension of a natural-gradient-inspired approach (ANaGRAM) and relies on cutoff regularization techniques for approximating the GRAM matrix in the update equation of the parameters of PINN representation. The main criticism by two reviewers in the lack of theoretical convergence results and the lack of more advance experiments that could include PDEs such the Navier-Stokes equation. I agree with the criticism of the reviewers.

**Reviewer Concerns:**

The main issue that I find with the paper is that the authors do not put an effort to address each reviewer comments in direct way. For this reason three reviewer mentioned in their comments that their concerns are not addressed.  In particular:

Reviewer rjuU feels that its criticism regarding convergence guarantees is not addressed.

Reviewer 6JRF  concerns  on single parameter adaptation, the limited benchmarking and the lack of convergence proofs are not addressed.

Reviewer yRr7 concern on the novelty of the proposed algorithms and the importance of the contribution is not addressed.

**Reviewer Scores:**

None of the reviewers would have changed the scores.  This is notified by at least three out of the four reviewers.

---

### Decision · Program_Chairs · 2026-01-26

Reject